# The E3 ubiquitin ligase mechanism specifying targeted microRNA degradation

Jakob Farnung[1,5], Elena Slobodyanyuk[2,3,4,5], Peter Y. Wang[2,3,4], Lianne W. Blodgett[2,3,4], Daniel H. Lin[2,3,4], Susanne von Gronau[1], Brenda A. Schulman[1✉] & David P. Bartel[2,3,4✉]

MicroRNAs (miRNAs) associate with Argonaute (AGO) proteins to form complexes that down-regulate target RNAs, including messenger RNAs from most human genes[1–3]. Within each complex, the miRNA pairs to target RNAs, and AGO provides effector function while also protecting the miRNA from cellular nucleases[2–5]. Although much is known about miRNA-directed gene regulation, less is known about how miRNAs themselves are regulated. One pathway that regulates miRNAs involves unusual targets called 'trigger' RNAs, which reverse the canonical regulatory logic and instead down-regulate miRNAs[6–9]. This target-directed miRNA degradation (TDMD) is thought to require a cullin–RING E3 ligase because it depends on the cullin protein CUL3 and other ubiquitylation components, including the BC-box protein ZSWIM8 (refs. 10,11). ZSWIM8 is required for murine perinatal viability and for destabilization of most short-lived miRNAs, which suggests biological importance of TDMD[11–13]. Here, biochemical and cellular assays establish AGO binding and polyubiquitylation by the ZSWIM8–CUL3 E3 ligase as the key regulatory steps of TDMD, and thereby define a unique cullin–RING E3 ligase class. Cryogenic electron microscopy analyses show ZSWIM8 recognizing distinct AGO and RNA conformations shaped by pairing of the miRNA to the trigger. Specificity of AGO ubiquitylation is established through generalizable RNA–RNA, RNA–protein and protein–protein interactions. The substrate features recognized by the E3 ligase do not conform to a conventional degron[14,15] but instead establish a two-RNA-factor authentication mechanism for specifying a protein ubiquitylation substrate.

Metazoan microRNAs (miRNAs) recognize their messenger RNA (mRNA) targets primarily through pairing between the miRNA 'seed' region (miRNA nucleotides 2–8) and sites within the mRNA 3' untranslated regions[16]. By contrast, the unusual transcripts that trigger miRNA degradation not only pair to the miRNA seed region but, following an internal loop involving the miRNA central region, also pair extensively to the miRNA 3' region[8,9]. Although these trigger sites presumably afford greater affinity to the Argonaute (AGO)–miRNA complex owing to pairing to the miRNA 3' region[17], this increased affinity is insufficient to explain the selectivity of the target-directed miRNA degradation (TDMD) machinery for the vastly outnumbered trigger-bound complexes. For a miRNA undergoing TDMD, trigger sites in the cellular transcriptome typically number approximately 100 (refs. 11,18), which, when compared with the approximately 100,000 seed-matched sites residing within mRNA 3' untranslated regions[19], present a stoichiometric challenge for molecular recognition.

Understanding of TDMD has come mainly from genetic analyses, which revealed key factors involved in TDMD, and molecular and physiological consequences of losing those factors[10–13,20–22]. This pathway appears to affect many metazoan miRNAs. Current tallies of ZSWIM8-sensitive miRNAs include over 50 in human cell lines, over 50 in mouse tissues, 21 in *Drosophila* embryos and cells and 22 in nematodes[11–13,20,22,23]. Additionally, some herpesviruses express trigger RNAs that direct decay of host miRNAs[7,24–26]. The breadth and magnitude of ZSWIM8-mediated miRNA regulation presumably underlie the lethality of ZSWIM8 knockout in both flies and mice[12,13,20,21], although other functions have also been proposed for ZSWIM8 (refs. 27–32).

In the current model for TDMD, the binding of a trigger RNA to the AGO–miRNA complex causes a conformational change that is recognized by a ZSWIM8–CUL3 cullin–RING E3 ligase (CRL) through an unknown mechanism[10,11,33] (Fig. 1a). This CRL, together with the ARIH1 RBR-type E3 ligase, along with cognate E2 ligases[10], presumably polyubiquitylates AGO[34–36], causing its degradation by the 26S proteasome and ultimately leaving the miRNA susceptible to cellular nucleases[10,11].

The ZSWIM8 CRL belongs to a mix-and-match system of substrate-binding receptors with common ubiquitylation components that assemble in various combinations to form hundreds of different E3 ligases[37]. The CRL that achieves TDMD has some unprecedented features. Genetic data implicate ZSWIM8 and its obligate partner proteins

[1]Department of Molecular Machines and Signaling, Max Planck Institute of Biochemistry, Martinsried, Germany. [2]Howard Hughes Medical Institute, Cambridge, MA, USA. [3]Whitehead Institute for Biomedical Research, Cambridge, MA, USA. [4]Department of Biology, Massachusetts Institute of Technology, Cambridge, MA, USA. [5]These authors contributed equally: Jakob Farnung, Elena Slobodyanyuk. ✉e-mail: schulman@biochem.mpg.de; dbartel@wi.mit.edu

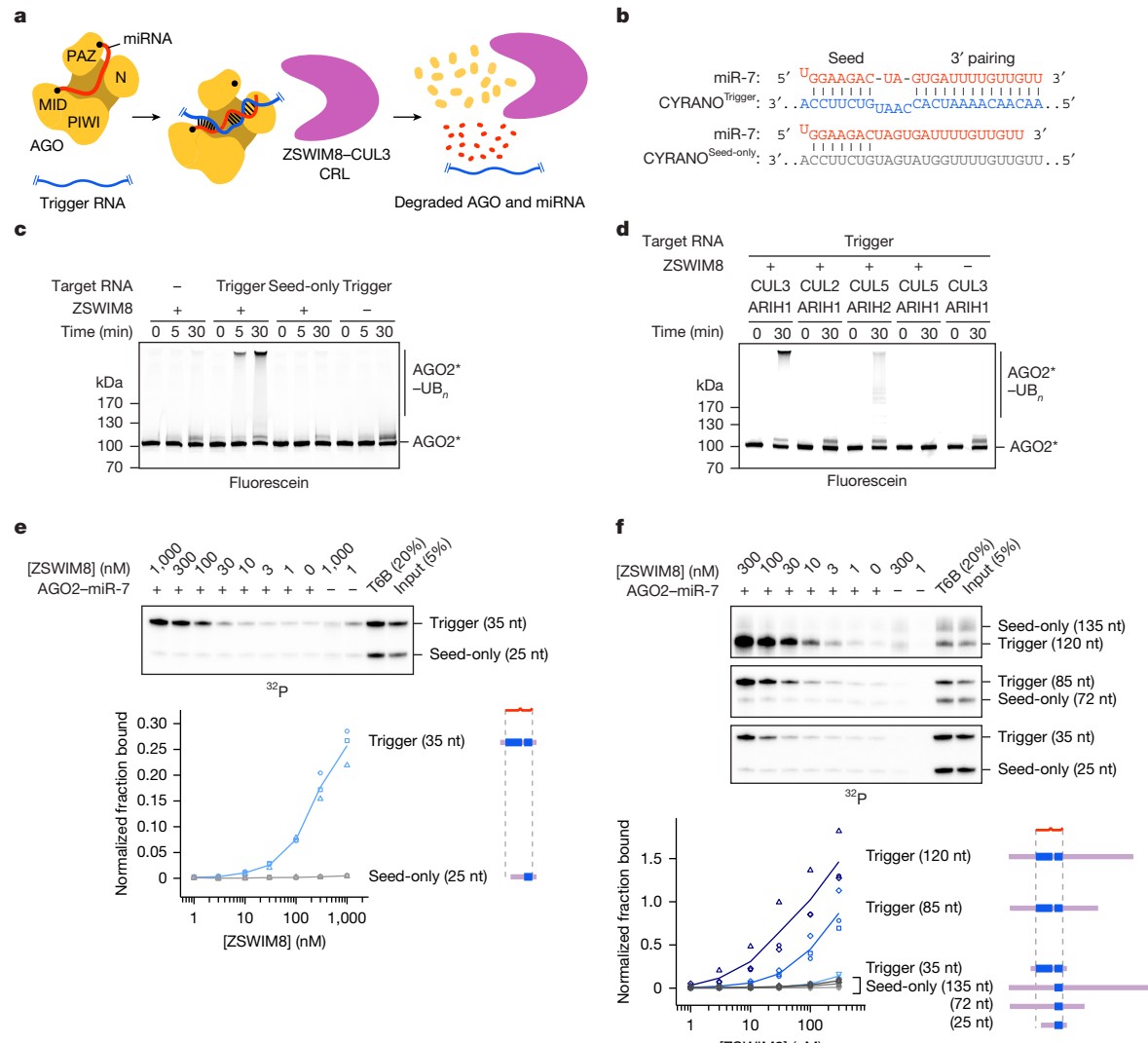

**Fig. 1 | Trigger RNA specifies AGO ubiquitylation by the ZSWIM8–CUL3 E3 ligase through selective ZSWIM8 binding. a**, Schematic of the proposed mechanism for TDMD. **b**, Diagrams showing pairing between miR-7 and CYRANO[Trigger] and CYRANO[Seed-only] target RNAs used for in vitro assays. Vertical lines represent Watson–Crick–Franklin pairing. **c**, Reconstitution of AGO2 polyubiquitylation using the miR-7–CYRANO miRNA–trigger pair. The two target RNAs were CYRANO[Trigger] (120 nt) and CYRANO[Seed-only] (135 nt) (Supplementary Table 1). Polyubiquitylated products (AGO2*–UB$_n$) were resolved by SDS–PAGE and detected by in-gel visualization of fluorescently labelled AGO2 (AGO2*). Shown is a representative experiment; $n = 2$ technical replicates. **d**, In vitro ubiquitylation of AGO2, comparing the activity of different cullin–UB-carrying-enzyme pairs. Otherwise, this panel is as in **c**. $n = 2$ technical replicates. **e**, Reconstitution of ZSWIM8 binding to AGO2–miR-7–CYRANO. Top, in vitro co-IP assay (Extended Data Fig. 1a). AGO2–miR-7 was pre-incubated with

radiolabelled target RNAs CYRANO[Trigger] (35 nt) and CYRANO[Seed-only] (25 nt) (Supplementary Table 1) and then incubated with excess epitope-tagged ZSWIM8. RNAs that co-IPed with ZSWIM8 were resolved on a denaturing gel, and radioactivity was visualized by phosphor imaging. Bottom, quantification of co-IPed RNA. Values were background-subtracted and normalized to those of the T6B sample. Symbols show results from independent measurements; lines pass through mean values. $n = 3$ technical replicates. At the right are schematics of the RNAs, indicating the degree of miRNA pairing (blue) and relative lengths of flanking regions (purple). **f**, Effect of RNA flanking the miRNA-binding site on ZSWIM8 binding to AGO2–miR-7–CYRANO. Top, in vitro co-IP assay, as in **e**, but target RNAs had extra CYRANO sequence flanking the miRNA-binding site, and heparin (1 µg ml$^{-1}$) was included to reduce non-specific binding of longer RNAs to the beads. Target RNAs from **e** were assayed for comparison. Bottom, quantification of co-IPed RNA, as in **e**. $n = 3$ technical replicates.

ELOB and ELOC (hereafter referred to collectively as ZSWIM8, except when referring to the *ZSWIM8* gene) as the substrate-binding receptor of a CUL3-based CRL[10,11,27]. However, how these proteins partner within a CRL is not clear; BC-box proteins such as ZSWIM8 are only known to function with CUL2 or CUL5 (refs. 37,38), whereas CUL3 is thought to partner exclusively with BTB-type substrate-binding receptors[37]. Moreover, for TDMD, this mix-and-match system of proteins is elaborated with a mix-and-match system of RNAs—the miRNAs and trigger RNAs—that promote polyubiquitylation of their common protein partner. Here we show how these RNAs provide a unique two-RNA-factor authentication mechanism specifying a ubiquitylation substrate.

## Trigger RNA specifies AGO ubiquitylation

Despite genetic and molecular evidence supporting an E3 model for TDMD[10,11] (Fig. 1a), ubiquitylation of TDMD-competent AGO had not been reported. We purified the genetically defined factors implicated in TDMD and assayed their collective ability to ubiquitylate AGO2 (one of four human AGO paralogues). Fluorescent AGO2 preloaded with miR-7 was polyubiquitylated on multiple surface-exposed lysines in a ZSWIM8-, CUL3- and ARIH1-dependent manner only if also incubated with a 120-nucleotide (nt) fragment of CYRANO, the trigger RNA of human miR-7 (ref. 18) (Fig. 1b,c, Supplementary Table 1 and Supplementary Fig. 3). A mutant ('seed-only') version of the CYRANO fragment, in

which pairing to the 3′ region of miR-7 was disrupted, failed to specify polyubiquitylation. The ZSWIM8–CUL3–ARIH1 E3–E3 assembly was required, as polyubiquitylation was not achieved with other canonical CRL–RBR E3–E3 partnerings (ZSWIM8–CUL2–ARIH1 or ZSWIM8–CUL5–ARIH2)[35,36,39] (Fig. 1d). Thus, AGO2 associated with a miRNA is a direct substrate for ZSWIM8–CUL3-mediated polyubiquitylation, but essentially only when bound to a trigger RNA.

## ZSWIM8 binds AGO–miRNA–trigger complex

Our in vitro ubiquitylation assay showed that components present in our purified system were sufficient for substrate selectivity. One mechanism for achieving this selectivity would be through preferential binding of ZSWIM8 to the AGO–miRNA–trigger ternary complex. We investigated this possibility with an in vitro co-immunoprecipitation (co-IP) assay designed to detect binding of the ternary complex. Excess purified epitope-tagged ZSWIM8 was incubated with AGO2–miR-7 that had been mixed with two radiolabelled target-RNA species—one harbouring the native CYRANO site, the other, a seed-only site. These RNAs were different sizes (35 and 25 nt, respectively), enabling RNA species co-purifying with ZSWIM8 to be resolved on a denaturing gel (Extended Data Fig. 1a). In parallel, co-IPs were performed with a peptide bait derived from TNRC6, a downstream effector of AGO[2]. This peptide (called 'T6B') binds AGO–miRNA complexes irrespective of their bound target RNAs[40]; it was used to normalize for differences in AGO2–miR-7 binding to different target RNAs (Extended Data Fig. 1a).

ZSWIM8 preferentially co-IPed with the AGO2–miR-7–trigger complex, with up to 70-fold enrichment of the trigger-bound complex over its seed-only counterpart (Fig. 1e). As another specificity control, we tested a target designed to represent '3′-supplementary' pairing, a type of pairing that increases affinity to the miRNA but is insufficient to direct miRNA degradation[3,8,9]. As expected, this target with 7 nt of 3′-supplementary pairing was unable to promote ZSWIM8 co-IP (Extended Data Fig. 1b,c). Together, these results indicated that preferential ZSWIM8 binding to the AGO2–miRNA–trigger complex underpins much of the selective activity observed in the in vitro ubiquitylation assay, and presumably also in cells. Beyond imparting preferential ZSWIM8 binding, the trigger RNA could also enhance catalytic efficiency through additional means, such as by orienting AGO lysine residues for more rapid ubiquitylation, or by increasing the processivity of polyubiquitylation.

We next examined whether these binding and polyubiquitylation activities were generalizable to other TDMD-competent AGO–miRNA complexes. ZSWIM8 displayed binding and polyubiquitylation activity towards another TDMD-competent AGO paralogue, human AGO1 (ref. 11) (Extended Data Fig. 1d,e), and toward another miRNA–trigger pair, miR-27a–HSUR1 (ref. 7) (Extended Data Fig. 1f–h). Importantly, as in regulation observed in vivo, cognate miRNA–trigger pairing was required in our biochemical reconstitution; HSUR1 failed to elicit polyubiquitylation of AGO2–miR-7, and CYRANO failed to elicit polyubiquitylation of AGO2–miR-27a (Extended Data Fig. 1i). Taken together, these results demonstrate that diverse but specific miRNA–trigger pairs direct the recognition of their common AGO protein partner by an uncharacterized class of CRL.

## Key role of RNA flanking trigger pairing

Previous studies on triggers had largely focused on the miRNA-binding site[8,9,23,33,41,42]. Our in vitro binding assay allowed assessment of whether parts of the trigger flanking the miRNA-binding site might also influence ZSWIM8 recognition. Including extra CYRANO sequence flanking the miR-7-binding site increased the efficiency of AGO2–miR-7–trigger co-IP by 100-fold. For instance, 300 nM ZSWIM8 was required to achieve 15% pull-down in the absence of extra flanking sequence, whereas only 3 nM was required in the presence of 85 nt of extra flanking sequence (Fig. 1f). Selectivity remained high when using the longer

trigger sequences, with a greater than 100-fold preference observed for CYRANO over its seed-only counterpart (Fig. 1f and Extended Data Fig. 1c). Increased co-IP efficiency was similarly observed for finer-grained extensions of the CYRANO fragment (Extended Data Fig. 2a), as well as with AGO2 paralogue AGO1 (Extended Data Fig. 2b). One explanation for the increased co-IP efficiency would be that trigger RNA flanking regions interact directly with ZSWIM8. Interestingly, the efficiency of co-IP remained high when CYRANO sequences flanking the miR-7 binding site were scrambled (Extended Data Fig. 2c,d), which suggested that ZSWIM8 has some sequence-independent affinity to RNA. Indeed, we detected weak ZSWIM8 binding to RNA alone in filter-binding experiments, with affinity increasing as RNA lengths increased from 28 to 120 nt (Extended Data Fig. 2e).

## ZSWIM8 dimer clamps AGO2–miR-7–CYRANO

To gain structural insights into TDMD substrate recognition, we obtained cryogenic electron microscopy (cryo-EM) data for a human ZSWIM8–CUL3 complex bound to an AGO2–miR-7–CYRANO complex. The CUL3 construct corresponded to the N-terminal domain (NTD) of CUL3, which binds the substrate receptor, and lacked the C-terminal domain, which binds RBX1. The final reconstruction has an overall resolution of 3.1 Å. A composite of focused refined maps showed extensive protein and RNA interactions (Fig. 2, Extended Data Table 1, Supplementary Figs. 4–6 and Supplementary Video 1).

ZSWIM8 forms a dimer, with each protomer projecting an alpha-helical solenoid. Using these solenoids, the two ZSWIM8 protomers form an asymmetric clamp around a single AGO–miR-7–CYRANO complex. One protomer, which we call ZSWIM8[NPAZ], interacts with the N and PAZ domains of AGO2, whereas the other one, which we call ZSWIM8[MID], interacts with the MID domain of AGO2. As described below, the structure rationalizes the unexpected finding that ZSWIM8 partners with CUL3 and explains how the trigger RNA orchestrates E3–substrate interactions enabled by its distinct pairing architecture with the miRNA. The high-resolution reconstruction showed how the signature miRNA–trigger pairing reshapes the conformation of AGO2 and the miRNA–trigger duplex to promote binding to ZSWIM8. miRNA–trigger pairing also renders a pocket within the AGO2 PAZ domain—which is otherwise occupied by the miRNA 3′ end—available to bind ZSWIM8. In addition, flanking regions of the trigger RNA, visualized as described below using low-pass filtering and three-dimensional (3D) variability analysis, wrap around the ZSWIM8 clamp and bind positively charged surfaces on ZSWIM8. In this way, disparate parts of the AGO2–miR-7–CYRANO complex present a constellation of ZSWIM8-interacting elements that dictate the selectivity of binding—in a manner that appears generalizable across miRNA–trigger pairings.

## A distinct class of cullin–RING ligases

ZSWIM8 differs from structurally characterized CRL substrate-binding receptors in having a dimeric E3 superdomain, which coordinates ZSWIM8 dimerization, CUL3 ubiquitin (UB) ligase assembly and substrate binding (Extended Data Fig. 3a). The dimeric E3 superdomain is a singular interconnected ZSWIM8 unit comprising four regions. First, a zinc-binding SWIM domain (residues 172–208) is the central organizer of the dimeric E3 superdomain (Extended Data Fig. 3a–c). It is essential for target-directed degradation of miR-7 in cells, as measured using a reporter assay that provides a fluorescent readout of cellular miR-7 activity[10] after rescue of *ZSWIM8*-knockout cells with a wild-type (WT) or variant *ZSWIM8* transgene of interest (Extended Data Fig. 3d–g, Supplementary Figs. 7 and 8 and Supplementary Table 2).

Second, the dimerizing D-domain (residues 225–268) is composed of intertwined helices from each ZSWIM8 protomer, as also observed for the D-domain of β-TRCP (refs. 43,44) (Extended Data Fig. 3h–j). The ZSWIM8 D-domain forms an intermolecular knot, presumably

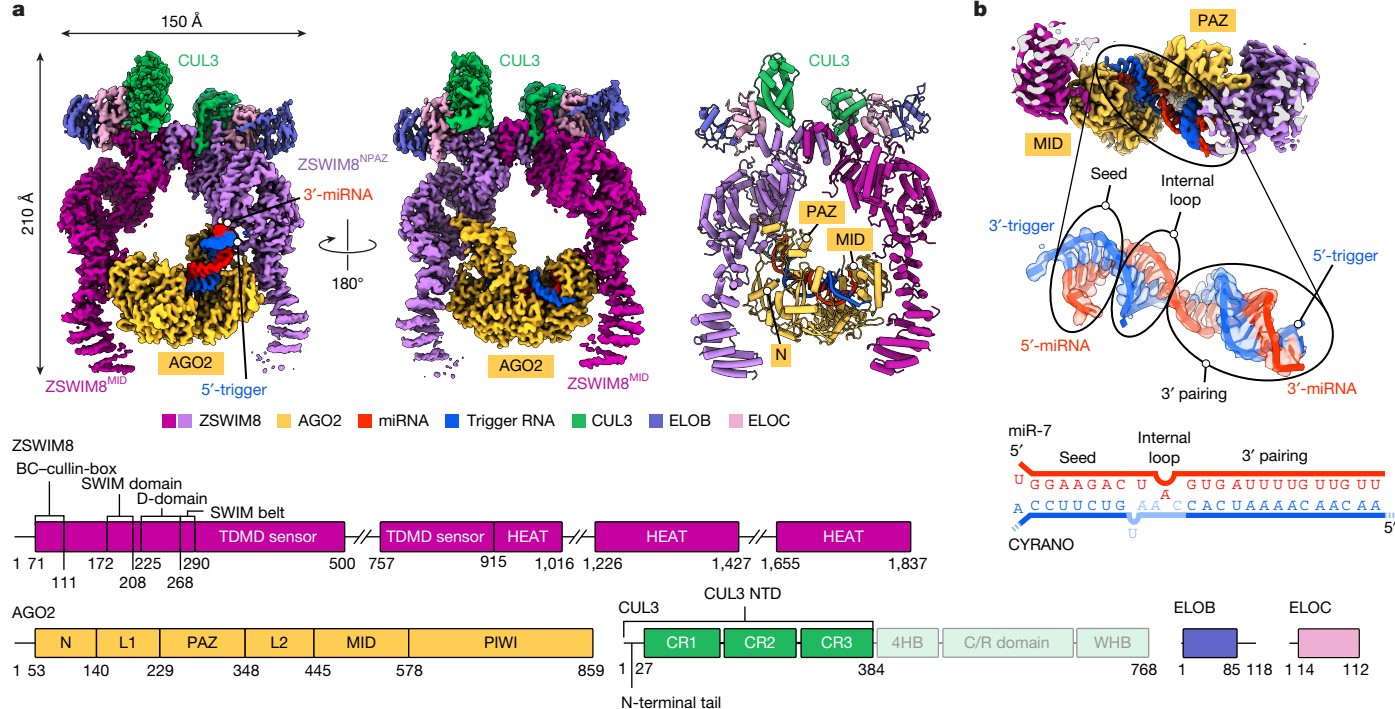

**Fig. 2 | ZSWIM8 dimer clamps around the AGO2–miR-7–CYRANO complex.**
**a**, Structure of ZSWIM8–CUL3 bound to AGO2–miR-7–CYRANO. Top left and top middle, cryo-EM map of the complex. The composite map based on multiple focused refined maps is shown. Top right, cartoon representation of the model. Large unstructured regions in ZSWIM8 (500–757, 1016–1226, 1427–1655) are not modelled. For the CYRANO trigger RNA, nucleotides 27–52 could be modelled (Supplementary Table 1). Bottom, domain architecture of ZSWIM8, AGO2, CUL3, ELOB and ELOC. Numbers indicate residues at domain boundaries. For CUL3, the NTD (1–384) used in the structural study is shown in bold green. Owing to structural flexibility, CUL3 NTD was modelled up to residue 188. **b**, Top, cut-out cryo-EM map showing AGO2 bound to the miR-7–CYRANO duplex, between ZSWIM8 protomers. Middle, map and model of the miR-7–CYRANO duplex. Bottom, schematic representation of the miR-7–CYRANO duplex, as observed in the density.

rendering dimerization irreversible when the adjacent domains fold and thereby prevent unthreading. To test the role of ZSWIM8 dimerization, we designed a monomeric version of ZSWIM8 (ZSWIM8^mono^) that retained CUL3 recruitment and intrinsic E3 ligase activity (Extended Data Fig. 3k,l). ZSWIM8^mono^ was defective at AGO2 polyubiquitylation in vitro and was correspondingly unable to rescue miR-7 degradation in ZSWIM8-deficient cells (Extended Data Fig. 3m,n).

Third, an extended region (residues 269–290) that we term the 'SWIM belt' emanates from the D-domain and connects the dimeric E3 superdomain to the downstream substrate-binding domain. The SWIM belt secures the BC-box and SWIM domain of the same protomer, the D-domain of the opposite protomer and part of the N-terminal tail of CUL3 (Extended Data Fig. 3a–c).

Fourth, ZSWIM8 displays a unique BC–cullin-box (residues 71–111)[38] (Extended Data Fig. 4a–c) with a ZSWIM-family-specific CUL3-box that interacts with CUL3 and excludes other cullins (Extended Data Fig. 4d–f). Furthermore, ELOC binds CUL3 cullin repeat 1 in a manner resembling how BTB domains bind CUL3 (ref. 45) and how ELOC otherwise binds CUL2 (ref. 46) (Extended Data Fig. 4c). Finally, the distinctive N-terminal tail of CUL3 is anchored into a unique groove formed by the BC–cullin-box, D-domain and SWIM belt of ZSWIM8 (refs. 47,48) (Extended Data Figs. 3h and 4g), which enables rotation of CUL3 around the D-domain, as shown by 3D variability analysis (Supplementary Video 2). Removal or mutation of the CUL3 N-terminal tail reduced affinity of CUL3 for ZSWIM8 and diminished AGO2 polyubiquitylation in vitro (Extended Data Fig. 4h,i). Reciprocally, mutation of ZSWIM8 residues (E91, E94) contacting the CUL3 N-terminal tail diminished AGO2 polyubiquitylation in vitro and abolished TDMD in cells (Extended Data Fig. 4i,j). Considering that the key CUL3-binding elements are maintained in other ZSWIM-family BC-box proteins (Extended Data Fig. 4d), these results define a distinct class of E3 ligases.

## Trigger shapes AGO–miRNA for E3 binding

The substrate-binding regions emanating from the ZSWIM8 dimeric E3 superdomain form a clamp that engulfs the AGO2–miR-7–CYRANO complex. This clamp makes multivalent interactions with the PAZ, N and MID domains of AGO2 and with the trigger RNA.

A key question was the extent to which the AGO2–miRNA–trigger complex bound by ZSWIM8 resembles the previously determined AGO2–miRNA–target structures lacking ZSWIM8. In the absence of ZSWIM8, AGO2–miRNA–target complexes populate distinct conformations depending on the miRNA–target pairing[33,49–53] (Fig. 3a and Extended Data Fig. 5a). Most prominently, the position of the AGO2 PAZ domain reports on the extent of pairing between the miRNA and the target site, with its position progressively shifting as pairing transitions from 3'-supplementary pairing to TDMD pairing (which involves more extended pairing to the miRNA 3' region) to full pairing[33,50,51] (Fig. 3a, Extended Data Fig. 5a and Supplementary Video 3). Transitioning from 3'-supplementary pairing to TDMD pairing also extracts the miRNA 3' end from the PAZ domain[33,50] (Fig. 3a, Extended Data Fig. 5a, Supplementary Fig. 10a and Supplementary Video 3).

The AGO2–miRNA–trigger conformation bound by ZSWIM8 in our structure resembles the TDMD conformation previously observed for AGO2–miRNA–trigger complexes in the absence of ZSWIM8 (ref. 33) (Fig. 3a). This resemblance indicates that ZSWIM8 largely recognizes AGO2 in the conformation intrinsically induced by miRNA–trigger pairing. Nonetheless, some differences observed in our E3-bound structure, including a shift of the PAZ domain (Fig. 3a and Supplementary Video 4), suggest that some dynamically populated conformational features of a trigger-bound AGO2–miRNA complex are captured after binding the E3. Additionally, the internal loop of the RNA duplex is fully

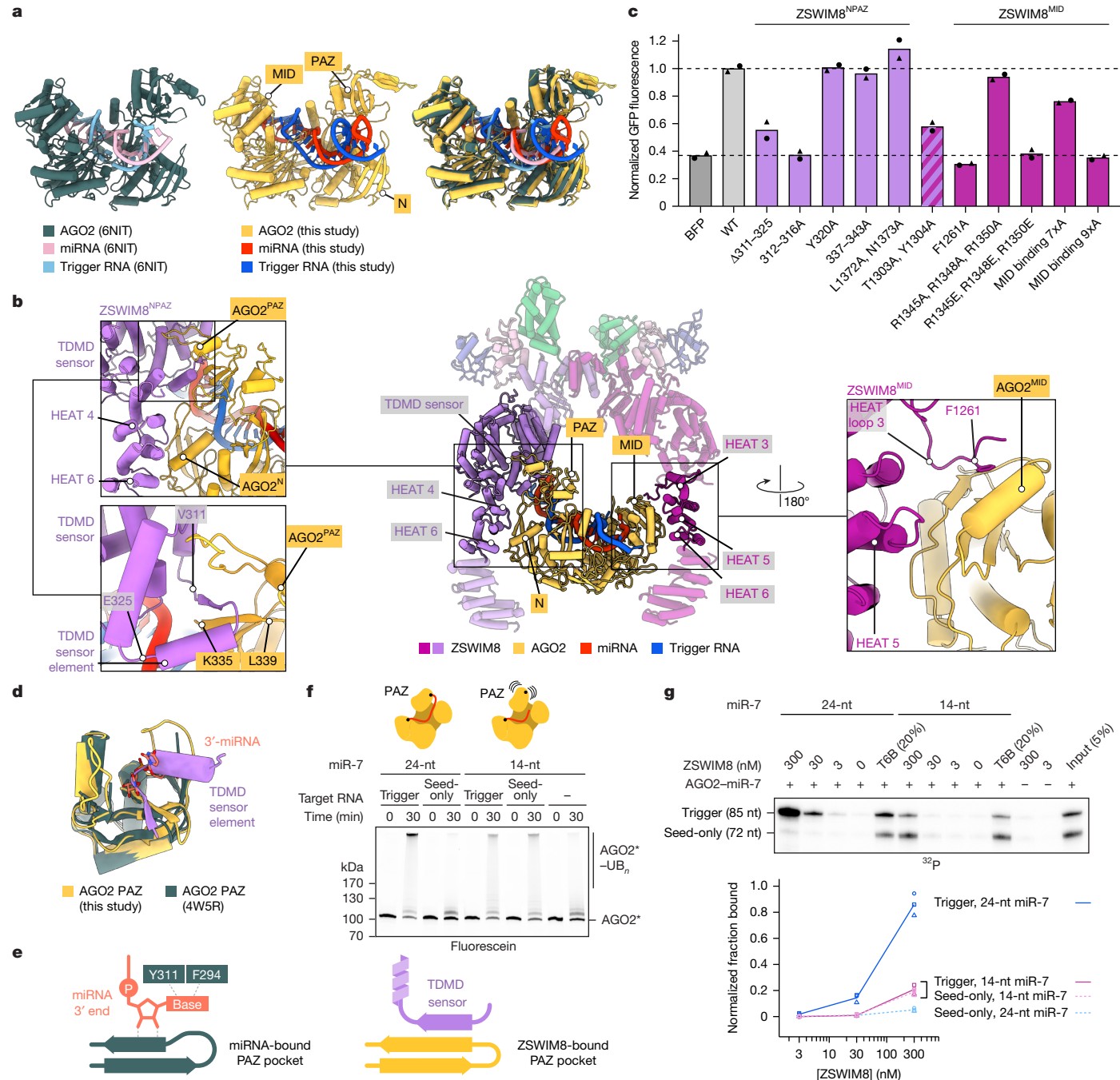

**Fig. 3 | Multivalent interactions between the ZSWIM8 dimeric clamp and AGO2 specify TDMD. a**, Structural comparison of AGO2–miR-7–CYRANO in association with ZSWIM8–CUL3 with a previously determined structure of AGO2–miRNA–trigger (but without ZSWIM8–CUL3) (PDB 6NIT). **b**, Cartoons illustrating ZSWIM8–CUL3 bound to AGO2–miR-7–CYRANO, highlighting interactions between ZSWIM8 and AGO2. Middle, overview of highlighted interactions. Top left, multivalent interactions between ZSWIM8^NPAZ and the AGO2 PAZ and N domains. Bottom left, close-up of ZSWIM8 TDMD sensor element interacting with the AGO2 PAZ-domain pocket. Right, multivalent interactions between ZSWIM8^MID and the AGO2 MID domain. **c**, Importance of ZSWIM8 clamp interactions for CYRANO-directed miR-7 degradation in cells. Plotted are results of the intracellular TDMD assay testing the ability of ZSWIM8 variants to rescue TDMD in *ZSWIM8*-knockout cells (Extended Data Fig. 3d–g), with a focus on mutations of residues that contact AGO2. Bars for

mutant variants are coloured by the relevant protomer that uses those residues to make the interaction (purple, ZSWIM8^NPAZ; magenta, ZSWIM8^MID; stripes, both). For Δ311–325, residues 311–325 were replaced with (GGGS)₃; for MID-binding variants 7xA and 9xA, either seven or nine residues were substituted with alanines, respectively (Supplementary Table 2). *n* = 2 biological replicates. **d**, Structural comparison of an isolated PAZ domain illustrating binding of either the ZSWIM8 TDMD sensor element (this study) or the miRNA 3′ terminus (PDB 4W5R). **e**, Schematic of mutually exclusive interactions between the PAZ-domain pocket of AGO2 and either the miRNA 3′ end (left) or the ZSWIM8 TDMD sensor element (right). **f**, Effect of miR-7 truncation on AGO2 polyubiquitylation. This panel is as in Fig. 1c, except it also examines AGO2 complexed with a truncated, 14-nt miR-7. *n* = 2 technical replicates. **g**, Effect of miR-7 truncation on ZSWIM8 binding to AGO2–miR-7–CYRANO. This panel is as in Fig. 1f, except the AGO2–miR-7 complexes are those of **f**. *n* = 3 technical replicates.

resolved and includes an A:U pair and intercalating nucleobases in a configuration that might support the AGO2 conformation recognized by ZSWIM8 (Fig. 2b).

Importantly, ZSWIM8 binding appears incompatible with conformations observed for AGO2–miRNA complexes not engaged with a TDMD trigger, as the PAZ and N domains would either clash with or

fail to contact ZSWIM8 (refs. 49–51) (Supplementary Fig. 10b). Thus, pairing to the trigger RNA reshapes the AGO2–miRNA complex into a conformation recognized by ZSWIM8.

## Multivalent binding of ZSWIM8 and AGO2

Following the ZSWIM8 dimeric E3 superdomain is a helix- and loop-rich region (residues 290–915) we term the 'TDMD sensor' because it detects most features imparted by the trigger RNA to the AGO2–miR-7 complex. Following this sensor is a stretch of HEAT repeats that also bind the substrate (Figs. 2a and 3b, Extended Data Fig. 5b,c and Supplementary Video 5). HEAT repeats form curved shapes that undergo accordion-like twisting and turning[54], enabling the two ZSWIM8 protomers to adopt different structures and asymmetrically capture the AGO2 substrate between them (Extended Data Fig. 5d and Supplementary Video 6).

In the ZSWIM8[NPAZ] protomer, the TDMD sensor makes three-way interactions with the miRNA–trigger duplex and the RNA-oriented AGO2 PAZ and N domains (Fig. 3b and Extended Data Fig. 5e,f). A loop-helix-turn-helix-loop element (ZSWIM8 residues 309–345) intercalates between a quartet of hairpins in the AGO2 PAZ domain. It contains a 'TDMD sensor element' (ZSWIM8 residues 311–325) that is visible only in ZSWIM8[NPAZ], in which it forms a β-sheet with one of the PAZ-domain hairpins (ZSWIM8 residues 311–314; AGO2 residues 335–337) (Supplementary Video 5). Following this β-strand of ZSWIM8, downstream residues traverse across the interior of the PAZ domain, contact the N domain, then reverse orientation to wrap around the distal edge of the PAZ domain (ZSWIM8 residues 337–343; AGO2 residues 297–303) (Fig. 3b). In addition, loops between HEAT repeats 3–4 and 5–6 form a surface that binds the edge of the AGO2 N domain (ZSWIM8 residues 1302–1306 and 1369–1375; AGO2 residues 116–127 and 81–85, respectively) (Fig. 3b and Extended Data Fig. 5e).

The opposite protomer, ZSWIM8[MID], uses the concave interior of HEAT repeats 3–6 to instead recognize the edge of the AGO2 MID domain (Fig. 3b and Extended Data Fig. 5e). In addition, an extended ZSWIM8 loop within HEAT-repeat 3 anchors the MID domain, primarily through a phenylalanine (F1261) inserted into an exposed hydrophobic cavity of the MID domain. This loop is resolved only in the ZSWIM8[MID] protomer, further illustrating the asymmetric recognition of AGO2 by the two protomers. As the MID domain undergoes little change in the transition to the TDMD conformation, we propose that these interactions serve as anchor points to help position the AGO–miRNA–trigger complex in an orientation suitable for the sensor domain of the opposite protomer to recognize trigger-induced conformational changes in the N and PAZ domains, and the trigger itself.

We tested these interactions by introducing mutations at the corresponding interfaces. These results, described below, demonstrated the role of the ZSWIM8 dimeric clamp in forming asymmetric, multivalent interactions that recognize TDMD-competent AGO.

### TDMD sensor interactions

Replacing the TDMD sensor element with a linker (Δ311–325) or mutating residues 312–316 impaired the ability of ZSWIM8 to rescue miR-7 degradation in ZSWIM8-deficient cells (Fig. 3c). This loss of TDMD was attributable to reduced polyubiquitylation, as indicated by biochemical assays (Extended Data Fig. 5g).

We also examined the effects of mutating AGO2 residues. In a cell-based co-IP assay, epitope-tagged AGO2 variants were expressed in WT and Δ*Zswim8* mouse embryonic fibroblasts, and the levels of both miR-7 and control miRNAs that co-IPed with each variant were quantified on northern blots[11] (Extended Data Fig. 6a). Although we were able to examine mutations of the MID domain, as described below, results of PAZ mutations were difficult to interpret in this assay, presumably because the mutations also caused defects in PAZ–miRNA association that impaired complex formation or stability in cells, which reduced

co-IP of not only miR-7 but also the control miRNAs—regardless of the presence of ZSWIM8 (Extended Data Fig. 6b,c). Therefore, for such PAZ variants, we turned to our in vitro ZSWIM8 co-IP assay, as we were able to generate the recombinant AGO2–miR-7 complexes to perform this assay. Replacement of the PAZ loops at AGO2 residues 296–305 or 332–336 reduced trigger-dependent ZSWIM8 binding (Extended Data Fig. 6d,e), reinforcing the importance of the PAZ domain as a binding platform for ZSWIM8.

### Anchoring interactions with AGO2 MID

ZSWIM8 residue F1261 docks at the centre of the MID-domain interface, and its mutation to alanine reduced AGO2 polyubiquitylation in vitro and abrogated TDMD in cells (Fig. 3c and Extended Data Fig. 5g). Reciprocally, mutation of AGO2 residues (F491 and K493) that bind ZSWIM8 F1261 resulted in accumulation of associated miR-7 in cells, as well as reduced trigger-dependent ZSWIM8 co-IP in vitro (Extended Data Fig. 6b–e).

### HEAT-repeat interactions

Substitutions at ZSWIM8 HEAT-repeat residues 1303 and 1304 were also detrimental (Fig. 3c and Extended Data Fig. 5g). However, because these residues appear to contact AGO2 in both protomers, interacting with the N and MID domains in the cases of ZSWIM8[NPAZ] and ZSWIM8[MID], respectively, the contributions of these potentially bifunctional residues could not be assigned to a single protomer.

AGO2 residues that interact with ZSWIM8 are conserved across AGO homologues in humans and other bilaterian species—especially those thought to be TDMD-sensitive (Supplementary Figs. 12 and 13). Likewise, ZSWIM8 residues that interact with AGO2 are conserved across bilaterian species (Supplementary Fig. 14) and also contribute to polyubiquitylation of AGO1 (Extended Data Fig. 5h), suggesting a general recognition mode of TDMD substrates by ZSWIM8.

## ZSWIM8 recognizes unoccupied PAZ pocket

The ZSWIM8 TDMD sensor element interacts with the AGO2 PAZ domain near the pocket that usually binds the miRNA 3′ terminus[55] but is vacated when the miRNA 3′ region pairs to the trigger (Fig. 3d,e). Thus, the ZSWIM8 TDMD sensor element might specifically recognize not only the position of the PAZ domain but also an unoccupied PAZ pocket as a feature of trigger-bound AGO–miRNA complexes. To test this hypothesis, we performed assays using AGO2–miR-7 variants designed to display a more constitutively accessible PAZ pocket, as described below.

### Mutation of AGO2 PAZ pocket

To impair miRNA 3′-end binding to the PAZ pocket, PAZ residues that interact with the terminal miRNA nucleotide (F294A and Y311A) were mutated[33,52] (Fig. 3e). These substitutions caused increased ZSWIM8 binding to the seed-only-bound AGO2–miR-7 (Extended Data Fig. 6d,e), consistent with an unoccupied PAZ pocket comprising a feature selectively recognized by ZSWIM8.

### miRNA truncation

We further tested sufficiency of an unoccupied PAZ pocket for mediating ZSWIM8 binding using another approach. AGO2 was loaded with a shortened miR-7, truncated after nucleotide 14, which was designed to be long enough to achieve seed-based target binding but too short to engage with the PAZ domain[56,57]. This truncation renders the 3′-end-binding pocket within PAZ unoccupied, regardless of whether the miRNA is bound to CYRANO or to its seed-only counterpart. With truncated miR-7, little difference was observed between the trigger and seed-only versions of CYRANO for both ZSWIM8 binding and AGO2 polyubiquitylation (Fig. 3f,g). Nonetheless, trigger-induced binding and polyubiquitylation were both reduced with truncated

miR-7 compared with full-length miR-7. Moreover, the residual poly-ubiquitylation required the presence of a target RNA, even though the truncated guide lacked the nucleotides required to form extensive 3′ pairing. These results suggested that an unoccupied PAZ pocket is important but not wholly sufficient to mediate binding to ZSWIM8, and that the purpose of extensive 3′ pairing to the trigger is not simply to vacate the PAZ pocket.

The observation of some binding and polyubiquitylation of AGO2 complexes containing truncated miRNAs suggested that similar complexes containing naturally truncated miRNAs (presumably resulting from extensive 3′-exonucleolytic trimming) might also be susceptible to ZSWIM8-dependent degradation in cells. To test this, we performed small-RNA sequencing of AGO-associated RNAs in WT and *Zswim8*-knockout cells to examine whether cells containing ZSWIM8 accumulate fewer miRNAs shorter than 19 nt. A statistically significant, albeit weak, signal for ZSWIM8-dependent reduction of such shortened miRNAs was observed, with a more substantial effect in *Drosophila* cells than in mammalian cells (Extended Data Fig. 7 and Supplementary Fig. 15). These results suggested that in addition to its role in TDMD, ZSWIM8 might act more broadly to destabilize AGO–miRNA complexes that contain extensively trimmed miRNAs.

## Flanking trigger RNA embraces ZSWIM8

Given the importance of RNA flanking the trigger site in directing AGO2–miR-7 binding to ZSWIM8 (Fig. 1f and Extended Data Fig. 2), we considered the trajectory of the miRNA–trigger duplex and how it might position the RNA flanking the trigger site. The internal loop of the miRNA–trigger duplex enables a bend between the seed and distal RNA helices[33] (Supplementary Fig. 10a). This bend causes the trajectory of the distal RNA helix to differ from that of other AGO2–miRNA–target conformations (Fig. 4a), facilitating contacts between the end of the distal helix and ZSWIM8[NPAZ]. These interactions were observed at high resolution for a loop within the TDMD sensor domain that we term RNA-binding element 1 (RBE1; residues 395–408). RBE1 wedges between the TDMD sensor element, the miRNA–trigger duplex and the HEAT-repeat subdomain and, together with the AGO2 PAZ domain, clasps the final turn of miR-7 (Fig. 4b). Accordingly, a ZSWIM8 variant harbouring charge-reversal substitutions in RBE1 had reduced ability to rescue TDMD in ZSWIM8-deficient cells and reduced polyubiquitylation activity in vitro (Fig. 4c and Extended Data Fig. 8a).

Although not visible at high resolution, additional density emanating from the 5′ and 3′ ends of the trigger site and embracing both ZSWIM8 protomers was visible at low resolution (Fig. 4d, Extended Data Fig. 8b and Supplementary Video 7). The low-resolution maps showed the 5′ flanking region of the trigger RNA surrounding RBE1, as well as two other positively charged ZSWIM8[NPAZ] loops designated RBE2 and RBE3 (residues 460–470 and 803–823, respectively) (Fig. 4d and Extended Data Fig. 8c). Low-resolution maps also suggested a path for the 3′ flanking region of the trigger RNA; weak density was visible exiting AGO2 adjacent to a C-terminal ZSWIM8[MID] HEAT-repeat and extending to RBEs 1–3 of this protomer (Fig. 4d and Supplementary Video 8). Thus, the two trigger regions that flank each end of the miRNA-binding site interact with the RBEs of both ZSWIM8 protomers to form a cross-brace securing the AGO2–miRNA complex inside the E3 ligase clamp. We further tested the effects of altering the RBEs and trigger RNA flanking regions.

### RNA-binding elements

Although charge-reversal mutations within RBE2 and RBE3 did not individually impact rescue of miR-7 degradation in ZSWIM8-deficient cells, combined mutation of all three RBEs exacerbated the defect caused by mutating RBE1, leading to severe loss of both TDMD in cells and polyubiquitylation in vitro (Fig. 4c and Extended Data Fig. 8a). Thus, the interactions between the trigger RNA and the ZSWIM8 RBEs

shown in our structure can rationalize the biochemical roles of flanking trigger RNA (Fig. 1f and Extended Data Fig. 2).

### Trigger flanking regions

The structure suggested that 25 nt of trigger RNA flanking the miRNA-binding site would be sufficient for both the 5′ and 3′ flanks to access the three RBEs on ZSWIM8[NPAZ] and ZSWIM8[MID], respectively. Indeed, lengthening the flanking sequences beyond 25 nt on either side had only a weak effect on the interaction with ZSWIM8 (Extended Data Fig. 8d), whereas eliminating the RNA immediately flanking either end of the trigger site reduced ZSWIM8 binding and AGO2 polyubiquitylation (Fig. 4e and Extended Data Fig. 8e). Deleting the 5′ flank was threefold more consequential to binding compared with deleting its 3′ counterpart (Fig. 4e). This directional hierarchy was maintained with two sets of scrambled flanking sequences, which indicated that the contributions of the trigger 5′ and 3′ flanks did not require sequence-specific contacts (Extended Data Fig. 8f,g). The stronger effect of the 5′ flanking sequence depended on pairing to the 3′ region of the miRNA (Extended Data Fig. 8h). These results supported the idea that trigger pairing specifies a trajectory of the 5′ flank of the trigger RNA, which favours interaction with the ZSWIM8[NPAZ] RBEs.

### Trigger loop length

To further investigate whether the miRNA–trigger conformation promotes ZSWIM8 recognition, we tested the effects of trigger variants expected to alter the RNA-duplex trajectory. Changing the length of the trigger internal loop (that is, the length of the target fragment bridging the two paired regions of the target site), and thereby presumably altering the distal miRNA–trigger duplex trajectory, substantially reduced ZSWIM8 binding and polyubiquitylation (Extended Data Fig. 9a–d). Thus, the configuration of the internal loop can influence TDMD, perhaps through its effect on accommodating the preferred duplex trajectory, as the internal loop does not make contacts with ZSWIM8. After mutation of the PAZ RNA-binding pocket (F294A, Y311A), trigger variants with longer internal loops induced binding and polyubiquitylation no better than did the seed-only variant, supporting the idea that together, trigger-specific RNA trajectory and binding of the vacated PAZ pocket substantially contribute to ZSWIM8 selectivity for TDMD substrates.

### Fully complementary target

Another way to perturb the RNA trajectory is to eliminate the internal loop altogether by replacing the trigger with a fully complementary target. As with TDMD pairing, fully complementary pairing extracts the miRNA 3′ end from its pocket in the PAZ domain[51–53]. However, in contrast to TDMD pairing, which accommodates a 40° bend in the duplex, the fully paired duplex adopts a relatively straight conformation and thus exits AGO2 along a different trajectory (Fig. 4a). Assays performed using a fully complementary target RNA and an AGO2 active-site variant (D669A, to prevent slicing of fully complementary target RNA) showed weak binding to ZSWIM8 and polyubiquitylation activity intermediate between that of the seed-only and trigger RNAs (Extended Data Fig. 9e–g). These results reinforce the concept that trigger-specific RNA trajectory, similar to miRNA 3′-end release, is an emergent feature of miRNA–trigger pairing that drives AGO recognition by ZSWIM8.

## Conclusions

Our results show how the exquisite selectivity of TDMD is achieved through numerous features orchestrated by the trigger RNA. In particular, pairing of the trigger RNA to an AGO2–miRNA complex induces (1) a structural remodelling of the AGO2 protein that arranges the MID, N and PAZ domains for simultaneous engagement by two interlocked ZSWIM8 protomers; (2) removal of the miRNA 3′ end from its binding pocket in the AGO2 PAZ domain, which exposes the binding pocket

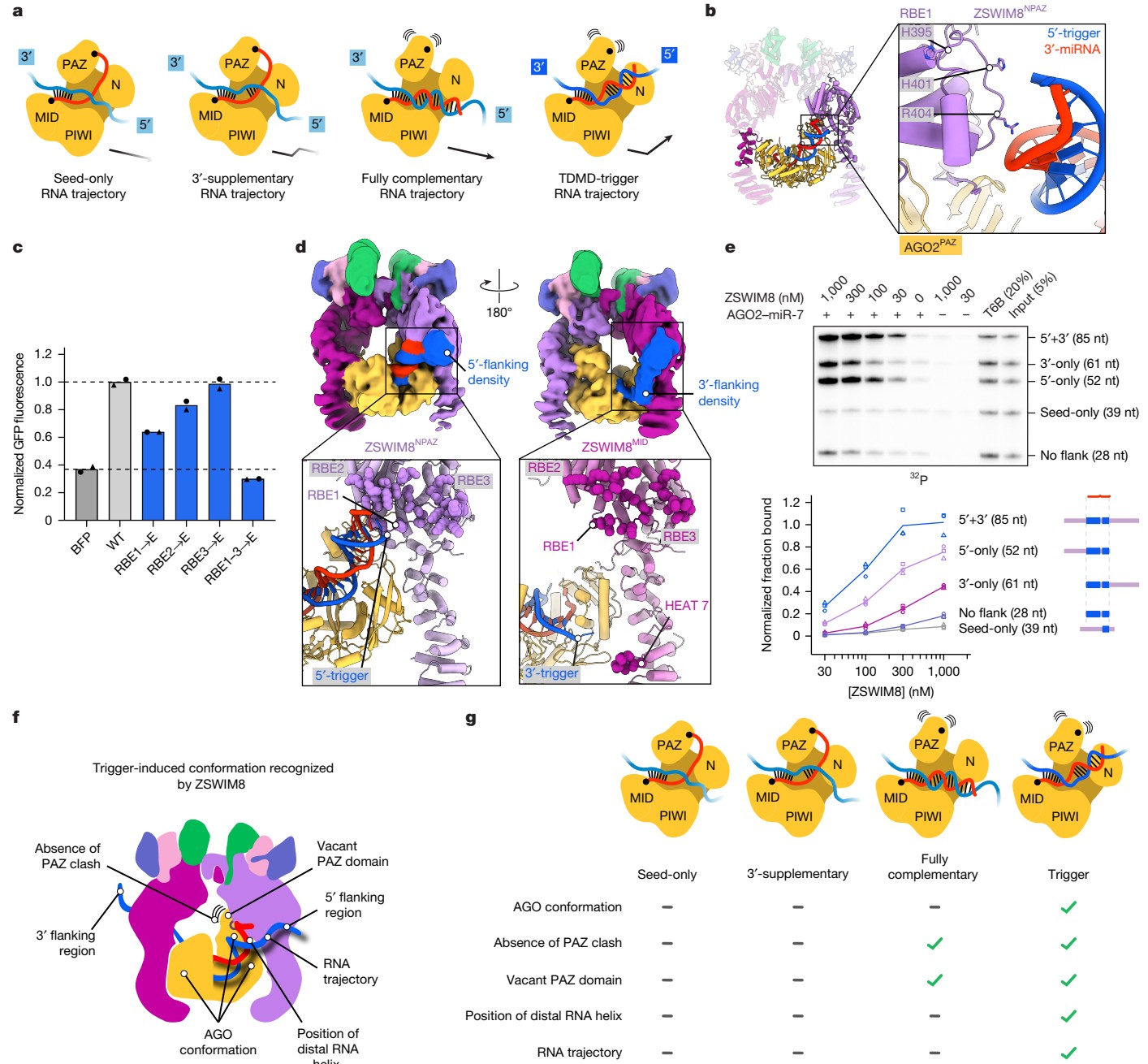

**Fig. 4 | Flanking trigger RNA embraces ZSWIM8. a**, Comparison of trajectories of target-RNA 5′ regions as they emerge from AGO2–miRNA–target structures with seed-only, 3′-supplementary, fully complementary or trigger pairing (based on PDBs 4W5R, 6N4O and 9CMP, and this study, respectively). Black arrows depict simplified trajectories of each target RNA, with opacity roughly indicating the degree of conformational constraint. **b**, Cartoon representation illustrating ZSWIM8^NPAZ interactions with the distal portion of the miRNA–target helix involving the 3′ region of the miRNA. RBE1 and its putative RNA-interacting residues are shown. **c**, Importance of ZSWIM8 RBEs for CYRANO-directed miR-7 degradation in cells. Plotted are results of the intracellular TDMD assay, as in Fig. 3c. Residues in RBEs were mutated to glutamate (E) residues. *n* = 2 biological replicates. **d**, Cryo-EM evidence for RNA in the vicinity of the RBEs of both ZSWIM8^NPAZ and ZSWIM8^MID. Top, cryo-EM map low-pass filtered to 10 Å. Density corresponding to the modelled complex is coloured as in Fig. 2.

Additional densities assigned as flanking RNA are in blue. Top-left panel shows extra 5′-trigger density interacting with ZSWIM8^NPAZ. Top-right panel shows extra 3′-trigger density interacting with ZSWIM8^MID. Bottom-left and bottom-right panels show cartoon representations with residues of RBEs 1–3 highlighted as either purple or magenta spheres, respectively. **e**, Contributions of RNA flanking the 5′ and 3′ ends of the trigger site to ZSWIM8 binding to AGO2– miR-7–CYRANO. Shown are results of the in vitro co-IP assay using CYRANO fragments with the indicated site-flanking regions and lengths; otherwise, as in Fig. 1f. *n* = 3 technical replicates. **f**, Model figure depicting structural changes in the AGO–miRNA complex induced by trigger binding, and the interactions leveraged by ZSWIM8 to detect trigger-bound AGO–miRNA. **g**, Extent to which various AGO–miRNA–target complexes contain the key features for ZSWIM8 recognition.

for recognition by the ZSWIM8 TDMD sensor domain; and (3) a unique trajectory of the miRNA–trigger duplex, which might favour direct contacts to the ZSWIM8 sensor domain and guide flanking regions of the trigger to engage with positively charged ZSWIM8 elements

(Fig. 4f,g). This multi-factorial selectivity ensures that miRNA degradation is tightly regulated such that AGO–miRNA complexes paired to non-trigger transcripts remain active as needed for biological regulation. We suspect that the reduced recognition of fully complementary

targets by ZSWIM8 also helps explain the high efficacy and long duration of small interfering RNA therapies[58].

The myriad trigger-RNA-induced interactions specifying substrate recognition by the ZSWIM8 E3 ligase do not correspond to a traditional degron motif[14,15]. Rather than interacting with either a linear peptide or a single extended surface, ZSWIM8 uses its dimeric architecture to interact with two extended surfaces of the substrate, and with the RNA. Thus, the structure of a ZSWIM8–CUL3-bound AGO2–miR-7–CYRANO complex illuminated a two-RNA-factor authentication mechanism determining E3 ligase substrate specificity. The effects of matching two RNA factors—the miRNA and the trigger RNA—propagate allosterically across both RNAs and the AGO protein to drive binding to ZSWIM8. These conformational changes seem to be conserved across many miRNA–trigger pairs, converging on a general trigger-bound AGO–miRNA conformation distinct from any other AGO–miRNA conformation. Although the possibility that some triggers might use sequence-specific factors or interactions to compensate for suboptimal miRNA–trigger pairing cannot be excluded[42,59,60], this generalized binding mode is supported by several observations. These include the charge-driven, sequence-independent interaction of ZSWIM8 with the trigger RNA within the complex that we examined, and the high sequence conservation specifically among TDMD-sensitive AGO homologues. Thus, overall, our work provides a structural and mechanistic framework for how RNA–RNA base pairing can induce generalizable ubiquitin-dependent protein degradation to ultimately induce degradation of a particular RNA.

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

## Methods

### Plasmids

Cloning was performed either by Gibson assembly with the NEBuilder HiFi DNA Assembly Master Mix (New England Biolabs (NEB), E2621) or by site-directed mutagenesis by direct PCR with the KAPA HiFi HotStart Ready Mix (Roche, 07958935001) or Q5 Master Mix (NEB, M0492S). Plasmids were purified using the Plasmid Plus Midi Kit (QIAGEN, 12945) and verified by Sanger or nanopore sequencing. Constructs for insect cell expression were cloned into pFLN to facilitate bacmid generation in DH10EmBacY[61].

### Cell culture

All mammalian cells were cultured at 37 °C with 5% $CO_2$. Expi293F cells (Gibco, A14527) were cultured in Expi293 expression media (Gibco, A1435102) and passaged 1:10 every 3 d to maintain a maximal cell density of 3 million cells per ml. K562 cells harbouring a miR-7-sensitive GFP reporter were a gift from J. Mendell[10]. K562 cells were cultured in RPMI 1640 (Gibco, 11875119) supplemented with 10% (v/v) fetal bovine serum (FBS) (Takara Bio, 631367) and passaged 1:4 every 3 d to maintain a maximal cell density of 800,000 cells per ml. Mouse embryonic fibroblast cells (MEFs) and HEK293FT cells were cultured in DMEM (Corning, 17-207-CV) with 10% FBS. Except in the case of contact inhibition for AGO2 intracellular co-IP and small-RNA-sequencing (sRNA-seq) experiments, described below, MEFs were passaged 1:4 or 1:8 as needed to maintain a maximal confluency of approximately 70%. HEK293FT cells were passaged 1:4 or 1:8 as needed to maintain a maximal confluency of approximately 70%. For passaging, adherent cells were dissociated using 0.25% trypsin–EDTA (Gibco, 25200114). For Expi293F cells, transfections were performed as described in the section 'AGO–miRNA complexes for in vitro co-IP assays'. For K562 and HEK293FT cells, transfections were performed with Lipofectamine 2000 (Invitrogen, 11668019) and Opti-MEM (Gibco, 31985062) according to the manufacturer's instructions. *Drosophila* S2 cells were cultured at 26 °C in Schneider's Drosophila Medium (Gibco, 21720001) supplemented with 10% heat-inactivated FBS (Gibco, 16140071). S2 cells were passaged every 3–6 d to maintain an approximate concentration between 1 and 10 million cells per ml. For insect cell expressions, baculoviruses were generated from bacmids in *Spodoptera frugiperda* Sf9 insect cells (Thermo Fisher Scientific, 11496015) and used to infect *Trichoplusia ni* High-Five insect cells (Thermo Fisher Scientific, B85502) for expression. Cell lines were not authenticated. All cell lines tested negative for mycoplasma contamination upon arrival to the lab.

### Proteins and RNAs

**Proteins for in vitro ubiquitylation assays and for cryo-EM.** Unless otherwise stated, all proteins are of human origin. Sequences of proteins and associated tags are listed in Supplementary Table 3. Genes were synthesized by TwistBiosciences or obtained as stated. Ubiquitin was expressed untagged and purified as described[62]. Cys-UB, in which ubiquitin carries an N-terminal cysteine for protein labelling, the lysine-less UB version (Cys-UB K0), NEDD8, UBE2L3, UBE2R2, UBE2M, ARIH1 and ARIH2 were expressed as GST-TEV or GST-3C fusions in BL21(DE3) RIL bacterial cells, as described[35,39,63,64]. Following glutathione-affinity pull-down (Cytiva, 17075605) and TEV protease cleavage, the protein was further purified by ion-exchange and size-exclusion chromatography. SrtA-5M, CUL3(NTD), CUL3(NTD, 1–384, I342R, L346D)-Avi and CUL3(ΔN, NTD, 25–384, I342R, L346D)-Avi were expressed in BL21(DE3) RIL cells and purified by immobilized metal-affinity chromatography (Ni-INDIGO, Cube Biotech, 75105), followed by ion-exchange chromatography and size-exclusion chromatography, as described[65,66]. CUL2, CUL3, CUL5, GST-TEV-RBX1(5–108), GST-TEV(5–113) and CUL2(NTD, 1–380, L344K, V3776D)-Avi were expressed in insect cells using two baculoviruses for assembly of CUL–RBX complexes and purified following

reported procedures[35,63]. CUL2–RBX1, CUL3–RBX1 and CUL5–RBX2 were neddylated and purified as described[35,63].

ZSWIM8-Twin-Strep was cloned into pFLN. ELOB and ELOC were cloned into pFLN and assembled into pBig1a using biGBac Assembly[61]. ZSWIM8–ELOB–ELOC complex (hereafter referred to as ZSWIM8–ELOB/C) was expressed in insect cells using two baculoviruses for 3 d at 27 °C. Cells were resuspended in lysis buffer (50 mM HEPES, 300 mM NaCl and 10% (v/v) glycerol, pH 7.5) supplemented with protease inhibitors and 1 mM tris(2-carboxyethyl)phosphine (TCEP). Cells were disrupted by sonication and cell debris was removed by centrifugation. ZSWIM8–ELOB/C was enriched using Twin-Strep affinity purification (Strep-Tactin, IBA Lifesciences, 2-1201-025) and eluted with lysis buffer supplemented with 3 mM desthiobiotin, followed by size-exclusion chromatography (HiLoad Superdex 200, Cytiva).

ZSWIM8$^{mono}$ was designed on the basis of our cryo-EM structure. We hypothesized that CUL3 binding to ZSWIM8 requires the presence of a folded D-domain. To functionally separate the requirement of ZSWIM8 dimerization for AGO2 binding and ubiquitylation from its impact on CUL3 binding, we designed a ZSWIM8 variant that retained a folded D-domain and retained binding to CUL3. We expected that transplanting a second copy of the D-domain into the native sequence would allow the D-domain to fold in *cis* and resemble the D-domain that is natively folded in *trans*. We inserted D-domain residues E222–P273 with two flanking GGSGGS linkers in between D-domain residues Q228 and R229 (Supplementary Table 2). AlphaFold3 structure predictions supported the expected assembly of the D-domain.

For structural studies and stepwise ubiquitin transfer assays, His$_6$-TEV-AGO2 was cloned into pFLN and expressed in insect cells. AGO was purified following an adapted purification protocol[67]. In brief, insect cells were resuspended in 50 mM Tris, 100 mM KCl and 5 mM dithiothreitol (DTT), pH 8.0. Following sonication, polyethylenimine was added to a concentration of 0.2% (v/v), mixed by inversion and immediately centrifuged to remove cell debris. AGO was enriched by immobilized metal-affinity chromatography (Ni-INDIGO, Cube Biotech, 75105). Following affinity chromatography, a nucleic-acid-free AGO fraction (as judged by 260/280 ratio) was isolated by cation-exchange chromatography (Source 15S or MonoS, Cytiva) with a gradient of 0–500 mM KCl in 20 mM Tris, pH 8.0. Nucleic-acid-free AGO was incubated with synthetic single-stranded miRNA (Metabion, IDT; Supplementary Table 4) at room temperature for 30 min. Excess miRNA was removed by size-exclusion chromatography in 25 mM HEPES, 150 mM NaCl and 1 mM TCEP, pH 7.8 (Superdex 200, Cytiva).

To label AGO for in vitro ubiquitylation assays, we used a split-intein-mediated labelling approach because of its high efficiency and site selectivity. We used the split intein gp41-1, which consists of an N-terminal part, gp41-1-N, and a C-terminal part, gp41-1-C[68,69]. These two parts excise themselves after *trans*-splicing, leaving a minimal ligation scar on the protein consisting of two short extein motifs. gp41-1-N was cloned into pGEX downstream of a TEV cleavage site. Following the TEV cleavage site, an N-terminal GGG motif was incorporated for sortase labelling, along with the native extein sequence SGY. GST-gp41-1-N was expressed in Rosetta *Escherichia coli* by induction with IPTG. Following glutathione-affinity purification and TEV protease cleavage, the protein was further purified by size-exclusion chromatography. gp41-1-C was cloned downstream of a Twin-Strep site into pFLN to create Twin-Strep-gp41-1-C-AGO2 or Twin-Strep-gp41-1-C-AGO1. These constructs contained the native extein linker, SSS, inserted between the split intein and the AGO sequence. AGO was purified following the procedure described above.

**Labelling and biotinylation of proteins.** Peptides for sortase labelling were obtained from the Peptide Synthesis Facility (Max-Planck Institute of Biochemistry Core Facility). The sortase peptide had the following sequence: GSGGLPETGG, with an N-terminally linked

carboxyfluorescein. Peptides were more than 90% pure as judged by HPLC and mass spectrometry.

gp41-1-N was fluorescently labelled at its N-terminal GGG using sortylation with a fluorescein-labelled acceptor peptide (fluorescein-GSGGLPETGG) (Supplementary Table 3). gp41-1-N (100 µM) was mixed with peptide (350 µM) in 25 mM HEPES, 150 mM NaCl and 10 mM CaCl$_2$, pH 7.5. The reaction was initiated by addition of SrtA-5M (10 µM)[70] and allowed to proceed for 1 h at room temperature. Ni-INDIGO beads (20 µl) (Cube Biotech, 75105) were added and excess peptide was removed using a PD MidiTRAP G-25 (Cytiva) followed by size-exclusion chromatography (Superdex 75) in 25 mM HEPES, 150 mM NaCl and 1 mM TCEP, pH 7.5.

For labelling AGO proteins with fluorescein-gp41-1-N, the nucleic-acid-free fraction containing Twin-Strep-gp41-1-C-AGO2 or Twin-Strep-gp41-1-C-AGO1, isolated through ion-exchange chromatography, was incubated with miRNA and fluorescein-gp41-1-N at 1.2× molar excess over AGO. The reaction was allowed to proceed for 1 h at room temperature. Excess miRNA and unreacted fluorescein-gp41-1-N were removed by size-exclusion chromatography in 25 mM HEPES, 150 mM NaCl and 1 mM TCEP, pH 7.5. Following labelling, AGO is modified at its N terminus with fluorescein-GGGSGYSSS (Supplementary Table 3). Fluorescently labelled AGO was used in all in vitro ubiquitylation assays, as described below.

For biotinylation of CUL-NTDs, Avi-tagged CUL-NTDs were incubated with 50 mM ATP, 25 mM MgCl$_2$, 5 mM biotin and 0.5 mg ml$^{-1}$ BirA at 4 °C for 18 h following ion-exchange chromatography. Biotinylated NTDs were purified by size-exclusion chromatography.

For fluorescent labelling of UB, Cys-UB and Cys-UB K0 (1 mM) were buffer-exchanged using a PD MidiTrap G-25 column (Cytiva, 28918008) into 25 mM HEPES, 150 mM NaCl and 0.5 mM TCEP and incubated with 5-(Iodoacetamido)fluorescein (5 mM, Sigma-Aldrich, I9271) for 18 h at 4 °C. Excess 5-(Iodoacetamido)fluorescein was removed using a PD MidiTrap G-25 column followed by size-exclusion chromatography (Superdex 75) in 25 mM HEPES and 150 mM NaCl to yield fluorescently labelled UB and UB K0 (UB* and UB K0*, respectively). UB* was used in stepwise ubiquitin transfer assays, and UB K0* was used in ubiquitylation site mapping, as described below.

**AGO–miRNA complexes for in vitro co-IP assays.** Plasmids expressing AGO2 variants were generated from the pcDNA3.3-3xFLAG-SUMO$^{EuI}$-HsAGO2 plasmid (Addgene, 231372). The plasmid expressing AGO1 was subcloned into the same backbone from an existing AGO1-expressing plasmid. Sequences of proteins and associated tags are listed in Supplementary Table 3. AGO–miRNA complexes were prepared largely as described[51,71]. Guide and passenger RNAs were each chemically synthesized (IDT) and gel-purified on a urea–polyacrylamide gel, then annealed into duplexes at 5 µM final concentration in annealing buffer (30 mM Tris-HCl pH 7.5, 100 mM NaCl and 1 mM EDTA). Annealing reactions were heated to 90 °C and slowly cooled to 30 °C over 1.5 h, before chilling on ice and storage at −80 °C. To generate cell lysate overexpressing the appropriate AGO variant, 200 ml of Expi293F cells at about 2 million per ml cell density were transfected with 190 µg of the AGO expression plasmid, 10 µg of pMaxGFP (Lonza) and 600 µg of poly-ethylenimine (Polysciences, 23966) incubated in 10 ml of Opti-MEM at room temperature for 20 min. After culture for 40 h, cells were collected and lysed in 8 ml of hypotonic buffer (10 mM HEPES pH 8.0, 10 mM KOAc, 1.5 mM Mg(OAc)$_2$, 2% (v/v) glycerol, 5 mM NDSB-256 and 0.5 mM TCEP, with 1 tablet of cOmplete Mini EDTA-free Protease Inhibitor Cocktail (Roche, 11873580001) per 10 ml of buffer) by a Dounce homogenizer. Cell lysate was clarified by centrifugation at 3,000g for 15 min at 4 °C, then again for 30 min. The supernatant was re-equilibrated by adding 25% volume of re-equilibration buffer (150 mM HEPES pH 8.0, 700 mM KOAc, 15 mM Mg(OAc)$_2$, 42% (v/v) glycerol, 5 mM NDSB-256 and 0.5 mM TCEP, with 1 tablet of cOmplete Mini EDTA-free Protease Inhibitor Cocktail per 10 ml of buffer), then further clarified by centrifugation at 60,000g for

20 min at 10 °C. Lysate was either used immediately or flash-frozen as aliquots in liquid nitrogen for storage at −150 °C.

To assemble each AGO–miRNA complex, 5–10 ml of cell lysate was first incubated with 600 µl of 5 µM annealed guide–passenger duplex for 1–2 h at 25 °C, to allow the in-lysate assembly of complexes. Meanwhile, 1 ml of slurry of Dynabeads MyOne Streptavidin C1 (Invitrogen, 65002) was washed according to the manufacturer's instructions, then pre-bound to 1 nmol of gel-purified biotinylated, 2′O-methylated capture oligonucleotide containing an 8mer seed-match site with complementarity to the loaded miRNA (Supplementary Table 4), at 25 °C for 30 min with end-over-end rotation, followed by four washes with ice-cold equilibration buffer (18 mM HEPES pH 7.4, 100 mM NaCl, 1 mM MgCl$_2$, 10% (v/v) glycerol, 0.003% (v/v) IGEPAL CA-630, 0.033 mg ml$^{-1}$ recombinant albumin (NEB, B9200) and 0.005 mg ml$^{-1}$ yeast transfer RNA (tRNA) (Life Technologies, AM7119)), then set aside at 4 °C until use. The incubated lysate was centrifuged at 6,000g for 10 min at 4 °C, filtered through a 5-µm PVDF filter (Millipore, SLSV025LS) and then added to the prepared beads and incubated at 25 °C for 1 h with end-over-end rotation. Beads were then washed twice with 5 ml of equilibration buffer and twice with 5 ml of capture-wash buffer (18 mM HEPES pH 7.4, 2 M NaCl, 1 mM MgCl$_2$, 10% (v/v) glycerol, 0.003% (v/v) IGEPAL CA-630, 0.033 mg ml$^{-1}$ recombinant albumin and 0.005 mg ml$^{-1}$ yeast tRNA), then incubated with 1 nmol 3′-biotinylated DNA competitor oligonucleotide (IDT) complementary to the capture oligonucleotide (Supplementary Table 4) in 18 mM HEPES pH 7.4, 1 M NaCl, 1 mM MgCl$_2$, 10% (v/v) glycerol, 0.003% (v/v) IGEPAL CA-630, 0.033 mg ml$^{-1}$ recombinant albumin and 0.005 mg ml$^{-1}$ yeast tRNA, at 25 °C for 2 h with end-over-end rotation. The eluate was then incubated with 50 µl of anti-FLAG M2 magnetic bead slurry (Millipore, M8823) pre-equilibrated in equilibration buffer according to manufacturer's instructions, at 25 °C for 2 h with end-over-end rotation. The beads were washed twice with 500 µl of equilibration buffer and twice with 500 µl of storage buffer (18 mM HEPES pH 7.4, 100 mM NaCl, 1 mM MgCl$_2$, 5% (v/v) glycerol and 0.003% (v/v) IGEPAL CA-630), before incubation in 100 µl of SENP$^{EuB}$ elution solution (500 nM of in-house purified SENP$^{EuB}$ protease[51] in storage buffer) at 4 °C with gentle rotation for 1 h. The eluate was combined with an extra 100-µl wash with SENP$^{EuB}$ elution solution, then supplemented with β-mercaptoethanol (BME) (Sigma-Aldrich, M6250) to a final concentration of 0.5 mM. Aliquots were flash-frozen in liquid nitrogen for storage at −80 °C. AGO2 loaded with a 14-nt miR-7 was purified using the same method as for purifying AGO–miRNA complexes for in vitro ubiquitylation assays, described above, but without fluorescent labelling.

Purified AGO–miRNA complexes for use in in vitro co-IP assays were quantified by filter-binding titration. A serial dilution series of limiting concentrations of complexes were incubated with 2 nM of a radiolabelled target RNA, in 18 mM HEPES pH 7.4, 100 mM NaCl, 1 mM MgCl$_2$, 2.5% (v/v) glycerol, 0.003% (v/v) IGEPAL CA-630, 0.5 mM BME, 0.017 mg ml$^{-1}$ recombinant albumin and 0.003 mg ml$^{-1}$ yeast tRNA, at 37 °C for 1 h. Filter binding was conducted as described above, except membranes were pre-equilibrated in 18 mM HEPES pH 7.4, 100 mM NaCl and 1 mM MgCl$_2$ and washed with 18 mM HEPES pH 7.4, 100 mM NaCl, 1 mM MgCl$_2$ and 5 mM DTT. Quantified fractions of target RNA bound across AGO–miRNA dilutions were fit to a quadratic equation by nonlinear least-squares regression in R using the Levenberg–Marquardt algorithm (nlsLM from the R package minpack.lm):

$$F_{bound} = \frac{DF[\text{stock}] + [\text{target}_T] + K_D - \sqrt{(DF[\text{stock}] + [\text{target}_T] + K_D)^2 - 4DF[\text{stock}][\text{target}_T]}}{2[\text{target}_T]} F_{max}$$

where $F_{bound}$ represents fraction of target bound, [target$_T$] represents the concentration of total target oligonucleotide, [stock] represents stock concentration of RISC, $DF$ represents the dilution factor, $K_D$ represents the dissociation constant for the affinity between RISC and the target

and $F_{max}$ represents the maximal fraction of target bound at the plateau. A range of bounds and initiation values were tested for [stock], $K_D$ and $F_{max}$ to ensure robust estimation of [stock]. $F_{max}$ was always limited to the range (0, 1). $K_D$ was fit in log-transformed space.

**ZSWIM8 for in vitro co-IP assays.** 3xFLAG-SUMO[EuI]-ZSWIM8-3xHA was cloned into pDARMO_CMVT (a gift from K. Rogala)[72]. ELOB and ELOC were cloned into pRK5. Sequences of proteins and associated tags are listed in Supplementary Table 3. For 200 ml of culture, Expi293F cells at about 3 million per ml cell density were transfected with 95 µg of the ZSWIM8 expression plasmid, 47.5 µg of the ELOB expression plasmid, 47.5 µg of the ELOC expression plasmid, 10 µg of pMaxGFP and 600 µg of polyethylenimine incubated in 24 ml of Opti-MEM at room temperature for 20 min. After culture for 48 h, cells were pelleted, washed with PBS and resuspended in double the cell pellet volume of lysis buffer (18 mM HEPES pH 7.4, 150 mM KOAc, 10 µM Zn(OAc)$_2$, 5% (v/v) glycerol, 0.5% (v/v) IGEPAL CA-630 and 0.5 mM TCEP, with 1 tablet of cOmplete Mini EDTA-free Protease Inhibitor Cocktail per 10 ml of buffer). The resuspension was passed through a 23-G1 M1.5-inch needle and incubated at 4 °C for 15 min with end-over-end rotation. Lysate was clarified by centrifugation at 1,000$g$ for 2 min at 4 °C, followed by centrifugation at 40,000$g$ for 30 min at 4 °C. Meanwhile, 1.2 ml of anti-FLAG magnetic beads (Genscript, L00835) were washed twice with wash buffer (18 mM HEPES pH 7.4, 150 mM KOAc, 5% (v/v) glycerol, 0.01% (v/v) IGEPAL CA-630 and 0.5 mM TCEP). Clarified lysate was filtered through a 5-µm PVDF filter, then incubated with the prepared beads at 4 °C for 1.5 h with end-over-end rotation. After incubation with lysate, beads were washed four times with 3 ml of wash buffer and eluted with 1 ml of elution buffer (5 µM of in-house purified SENP[EuB] protease[51] in wash buffer) at 4 °C for 1.5 h with end-over-end rotation. The eluate was centrifuged at 17,000$g$ for 5 min at 4 °C, and the supernatant was collected and quantified by nanodrop. Aliquots were flash-frozen in liquid nitrogen for storage at −80 °C.

**T6B peptide.** 3xHA-tagged T6B peptide (6xHis-SUMO-TNRC6B (599–683)-3xHA[73], otherwise referred to as T6B-3xHA peptide) was expressed and purified as described previously[74], with some modifications. 3xFLAG-tagged T6B peptide (6xHis-SUMO-TNRC6B(599–683)-3xFLAG, otherwise referred to as T6B-3xFLAG peptide) used in sRNA-seq experiments, described below, was purified by an analogous method. Sequences of proteins and associated tags are listed in Supplementary Table 3.

6xHis-SUMO-TNRC6B(599–683)-3xHA was cloned into pET28a. The expression construct was transformed into BL21 (DE3)-competent *E. coli* (NEB, C2527H). A single transformant was used to seed an overnight culture in Luria broth supplemented with 50 µg ml$^{-1}$ kanamycin (GoldBio, K-120-5) and 35 µg ml$^{-1}$ chloramphenicol (GoldBio, C-105-25) and grown at 30 °C overnight with shaking. Each 1.5-l batch of Luria broth was supplemented with 50 µg ml$^{-1}$ kanamycin and 35 µg ml$^{-1}$ chloramphenicol and seeded with 15 ml of overnight culture. This culture was grown at 37 °C with shaking. Once the culture reached an optical density (OD)$_{600}$ of around 0.2, it was transferred to 18 °C with shaking, and, once it reached an OD$_{600}$ of around 0.6, 0.5 mM of IPTG (GoldBio, I2481C50) was added. Induced cells were grown further overnight at 18 °C with shaking.

Cells were collected by centrifuging at 8,980$g$ for 6 min at 4 °C. Pellets were resuspended in lysis buffer (50 mM Tris pH 8.0, 500 mM NaCl, 5% (v/v) glycerol, 15 mM imidazole, 4 mM BME and 1 tablet of cOmplete EDTA-free Protease Inhibitor Cocktail (Roche, 11873580001)), with 1 ml of lysis buffer per 240 ml of overnight culture. Cells were lysed by sonication, and lysate was clarified by centrifuging at 41,400$g$ for 30 min at 4 °C. Clarified lysate was added to Ni-NTA agarose beads (QIAGEN, 30210) that had been washed twice with lysis buffer (280 µl of slurry per 1 l of culture), and the slurry was incubated at 4 °C for 2 h with end-over-end rotation. The beads were collected by centrifugation at

100$g$ for 2–5 min at 4 °C and washed three times with 40 bead volumes of lysis buffer. Following the last wash, beads were resuspended in 5 ml of lysis buffer and transferred to a gravity-flow column. Captured protein was eluted with elution buffer (50 mM Tris pH 8.0, 500 mM NaCl, 5% (v/v) glycerol, 250 mM imidazole and 4 mM BME).

The eluate was dialysed at 4 °C overnight in dialysis buffer (50 mM Tris pH 8.0, 100 mM NaCl, 5% (v/v) glycerol, 15 mM imidazole and 4 mM BME) using a 10-kDa molecular weight cutoff dialysis cassette (Thermo Scientific, 69570). On the following day, the dialysed eluate was purified by anion-exchange chromatography on either a Resource Q column (Cytiva, 17-1179-01) or a HiTrap Q XL column (Cytiva, 17515901) pre-equilibrated in buffer A (50 mM Tris pH 8.0, 100 mM NaCl, 5% (v/v) glycerol and 4 mM BME). T6B-3xHA peptide was eluted off the column with a linear gradient of buffer B (50 mM Tris pH 8.0, 2 M NaCl, 5% (v/v) glycerol and 4 mM BME). Fractions containing T6B-3xHA peptide were pooled and concentrated by centrifugal filtration using a concentrator pre-equilibrated with buffer A (Amicon Ultra 10-kDa cutoff, Millipore, UFC8010). Concentrated T6B-3xHA peptide was further purified using size-exclusion chromatography with a Superdex 200 Increase column (Cytiva, 28990944) in buffer A. Fractions containing T6B-3xHA peptide were pooled, then centrifuged at 17,000$g$ for 5 min at 4 °C, and the supernatant was collected and quantified by nanodrop. Aliquots were flash-frozen in liquid nitrogen for storage at −80 °C.

**Target RNAs.** Shorter target RNAs were chemically synthesized (IDT; Supplementary Tables 1 and 4) with 5′ hydroxyl chemistry, and gel-purified on a urea-polyacrylamide gel. Longer target RNAs were in vitro transcribed using chemically synthesized and gel-purified single-stranded DNA templates annealed to an oligonucleotide containing the T7 promoter sequence (IDT; Supplementary Tables 1 and 4). Transcription reactions using in-house purified T7 RNA polymerase were conducted in 5 mM ATP, 2 mM UTP, 5 mM CTP, 8 mM GTP, 5 mM DTT, 40 mM Tris pH 7.9, 2.5 mM spermidine, 26 mM MgCl$_2$, 0.01% (v/v) Triton X-100, 5 mM DTT, SUPERase•In (1 U µl$^{-1}$; Invitrogen, AM2694) and Thermostable Inorganic Pyrophosphatase (0.0083 U µl$^{-1}$; NEB, M0296). After incubation at 37 °C for 3–4 h, DNA templates were digested with RQ1 DNase (Promega, M6101) at 37 °C for 30 min. RNA products were then gel-purified on a urea-polyacrylamide gel.

Gel-purified chemically synthesized RNAs were radiolabelled directly using T4 Polynucleotide Kinase (PNK) (NEB, M0201) and [γ-$^{32}$P] ATP (Revvity, BLU035C005MC) in T4 PNK reaction buffer (NEB, B0201S) at 37 °C for 1–2 h, then desalted using Micro Bio-Spin P-6 columns (Bio-Rad, 7326221) and gel-purified on a urea-polyacrylamide gel. In vitro transcription products were first dephosphorylated using Quick CIP (NEB, M0525) at 37 °C for 15 min, followed by heat-inactivation at 80 °C for 3 min. Dephosphorylated RNAs were then immediately radiolabelled using T4 PNK and [γ-$^{32}$P] ATP in rCutSmart buffer (NEB, B6004S) supplemented with 5 mM DTT (Invitrogen, 15508013) at 37 °C for 1–2 h, then desalted using Micro Bio-Spin P-30 columns (Bio-Rad, 7326250), Micro Bio-Spin P-6 columns or Amersham MicroSpin G-25 columns (Cytiva, 27532501) and gel-purified on a urea-polyacrylamide gel.

### Biochemical assays

**In vitro ubiquitylation assays.** The identification of cullin–RING ligase subunits and the ARIH1 RBR-type E3 as TDMD effectors[10] suggested that ZSWIM8 performs ubiquitylation by the previously defined E3–E3 mechanism, which is a variation on the canonical E1–E2–E3 pathway[35,36]. In the E3–E3 mechanism, ubiquitin is transferred from the E2 UBE2L3 to the active site of ARIH1 only after ARIH1 activation through binding a neddylated CRL. ARIH1 then ubiquitylates the substrate recruited to the CRL substrate receptor, and generates short ubiquitin chains. Optimal polyubiquitylation is achieved in collaboration with a UBE2R-family E2, also activated by the neddylated CRL. The design of our in vitro ubiquitylation assays was informed by this mechanism.

All concentrations stated refer to the final concentrations used. Fluorescently labelled AGO (0.2 μM) loaded with the indicated miRNA was incubated with a target RNA (0.22 μM) for 15 min at room temperature. In the meantime, CUL-NEDD8–RBX1 (0.5 μM), ZSWIM8–ELOB/C (0.5 μM), ARIH1 (0.4 μM), UBE2L3 (1.5 μM), UBE2R2 (1.5 μM), UB (50 μM) and BSA (0.5 mg ml⁻¹) were mixed in 50 mM HEPES, 50 mM NaCl and 1 mM TCEP, pH 7.5. AGO–miRNA–target-RNA complex was added to the remaining components and incubated for 1 h at room temperature. ATP (5 mM) and MgCl$_2$ (7.5 mM) were added, and the reaction was initiated by addition of UBA1 (0.1 μM). At indicated time points, samples were removed and quenched with an equal volume of 2 × SDS–PAGE sample buffer (100 mM Tris-HCl, 20% (v/v) glycerol, 30 mM EDTA, 4% (v/v) SDS and 4% BME) and incubated at 95 °C for 5 min. Samples were resolved on 4–22% SDS–PAGE gels and visualized on a Typhoon 9410 Imager by in-gel fluorescence, followed by gel staining with Coomassie blue. Assays were performed with $n \geq 2$, and representative gels are shown. For reactions containing CUL5 and ARIH2, CUL-NEDD8–RBX1 was substituted for CUL5-NEDD8–RBX2, and ARIH1 was substituted with ARIH2. The assay setup as described above was used to generate the data shown in Figs. 1c,d and 3f and Extended Data Figs. 1e,h,i, 3m, 4i, 5g,h, 8a,e and 9b,g. Uncropped scans of fluorescent gels and gels stained with Coomassie blue are provided in Supplementary Figs. 1 and 2, respectively.

**Stepwise ubiquitin transfer assays.** Stepwise transfer assays were performed in a two-step process consisting of a loading reaction and a discharge reaction. In the loading reaction, the UB-carrying enzyme (UBE2L3) is charged by UBA1, and quenched by apyrase, which prevents further ubiquitin activation once UBE2L3 is discharged. The discharge reaction is initiated by addition of the remaining reaction components (CRL, substrate receptor, substrate and other E3s). Stepwise ubiquitin transfer assays follow the transfer of ubiquitin not just to the substrate but also to and from other proteins, such as E2 and E3 components. Substrate receptors that interact with CRLs are commonly autoubiquitylated and thus autoubiquitylation provides a functional readout of the ability of a substrate receptor to interact with a catalytically competent CRL.

All concentrations stated refer to the final concentrations used. AGO2 (0.2 μM) loaded with the indicated miRNA was incubated with a target RNA (0.22 μM) at room temperature for 15 min. In the meantime, CUL-NEDD8–RBX1 (0.5 μM), ZSWIM8–ELOB/C (0.5 μM), ARIH1 (0.4 μM) and BSA (0.5 mg ml⁻¹) were mixed in 50 mM HEPES, 50 mM NaCl and 1 mM TCEP, pH 7.5, and incubated at room temperature for 15 min. In the meantime, UBE2L3 (10 μM) was mixed with fluorescein-UB (15 μM) in 25 mM HEPES, 100 mM NaCl, 2.5 mM MgCl$_2$ and 1 mM ATP at room temperature. E2 charging was initiated by addition of UBA1 (0.4 μM) and allowed to proceed for 15 min. The reaction was quenched by addition of apyrase (NEB, 0.25 units) and incubated at 4 °C for 5 min. Assuming complete charging of UBE2L3, UBE2L3–UB* was further diluted to 2 μM in 25 mM HEPES and 150 mM NaCl, pH 7.5. Ubiquitylation was initiated by addition of UBE2L3–UB to the other components at a final concentration of 0.4 μM. For the sample at $t_0$, an aliquot of UBE2L3–UB* was diluted to 0.4 μM in 25 mM HEPES and 150 mM NaCl, pH 7.5, and a sample was immediately removed and quenched with 2 × SDS–PAGE sample buffer (100 mM Tris-HCl, 20% glycerol, 30 mM EDTA and 4% SDS). Samples were removed and quenched at indicated time points. Samples were resolved on 4–22% SDS–PAGE gels and visualized on a Typhoon 9410 Imager by in-gel fluorescence, followed by gel staining with Coomassie blue. Assays were performed with $n \geq 2$, and representative gels are shown. The assay setup as described above was used to generate the data shown in Extended Data Figs. 3l and 4f. Uncropped scans of fluorescent gels and gels stained with Coomassie blue are provided in Supplementary Figs. 1 and 2, respectively.

**Ubiquitylation site mapping.** To identify ubiquitylated lysines on AGO2, we adapted the stepwise ubiquitin transfer assay described

above. Final concentrations were increased to CUL3-NEDD8–RBX1 (1.6 μM), ZSWIM8–ELOB/C (1.6 μM), ARIH1 (1.2 μM), AGO2 (2 μM) and UBE2L3-UB K0* (0.8 μM), and the reaction volume was increased to 50 μl. Instead of fluorescently labelled WT UB, we used fluorescently labelled UB K0, in which all lysines have been mutated to arginine to prevent unwanted ubiquitin chain formation. The reaction was allowed to proceed for 5 min at room temperature and was quenched by addition of 20 mM DTT.

Hereafter, 50 μl of SDC buffer (1% sodium deoxycholate, 40 mM 2-chloroacetamide and 10 mM TCEP in 100 mM Tris, pH 8.0) was added. Samples were incubated for 20 min at 37 °C and subsequently diluted 1:1 with water. Proteins were digested overnight at 37 °C using 0.5 μg of Lys-C and 1 μg of trypsin. Following digestion, peptides were acidified with trifluoroacetic acid to a final concentration of 1%. Approximately 200 ng of peptide material was loaded onto Evotips (Evotip Pure, Evosep) for liquid chromatography–tandem mass spectrometry analysis.

Evotips were eluted onto a 15-cm C18 analytical column (PepSep C18, 15 cm × 150 μm, 1.5 μm; Bruker Daltonics) using the Evosep One HPLC system. The column temperature was maintained at 50 °C, and peptides were separated using the 30 SPD method. Eluting peptides were directly introduced into a timsTOF HT mass spectrometer (Bruker Daltonics) via a nanoelectrospray ionization source. The mass spectrometer was operated in data-dependent PASEF (parallel accumulation serial fragmentation) mode, comprising one survey TIMS (trapped ion mobility spectrometry) mass spectrometry scan followed by ten PASEF tandem mass spectrometry scans per acquisition cycle. Data were acquired over a mass range of $m/z$ 100–1,700 and an ion mobility range of $1/K_0 = 0.85$–1.35 Vs cm⁻², using equal ion accumulation and ramp times of 100 ms in the dual TIMS analyser, corresponding to a spectral rate of 9.43 Hz. Precursor ions were isolated using a 2-Th window for $m/z < 700$ and a 3-Th window for $m/z > 700$ by synchronized quadrupole switching during TIMS elution. Collision energy was applied as a function of ion mobility, ranging from 45 eV at $1/K_0 = 1.3$ Vs cm⁻² to 27 eV at 0.85 Vs cm⁻², with linear interpolation between these values. Singly charged precursor ions were excluded using a polygon filter mask, and further $m/z$ and ion mobility criteria were applied for dynamic exclusion.

Raw mass spectrometry data were processed using FragPipe (v.23.0) with default parameters from the LFQ-UB workflow. Spectra were searched against a database containing common contaminants and the proteins of interest. Carbamidomethylation of cysteine was set as a fixed modification, whereas methionine oxidation, lysine ubiquitylation, and protein N-terminal acetylation were included as variable modifications. StrictTrypsin was specified as the protease. The identified ubiquitylation sites are shown in Supplementary Fig. 3.

**In vitro co-IP assays.** All concentrations stated refer to the final concentrations used. Pierce anti-HA magnetic bead slurry (Thermo Scientific, 88837, 15 μl per reaction) was washed twice with blocking buffer (18 mM HEPES pH 7.4, 150 mM KOAc, 1 mM Mg(OAc)$_2$, 5% (v/v) glycerol, 0.01% (v/v) IGEPAL CA-630, 0.1 mg ml⁻¹ recombinant albumin and 0.01 mg ml⁻¹ yeast tRNA) and incubated in blocking buffer at 4 °C for 2 h with end-over-end rotation. Blocking buffer was then removed immediately before use. Radiolabelled target RNAs were prepared as described above and pre-mixed. Pre-mixed RNA concentrations were calculated on the basis of a conservative estimate of 0–50% loss during radiolabelling. AGO–miRNA complex was pre-diluted in storage buffer (18 mM HEPES pH 7.4, 100 mM KOAc, 1 mM Mg(OAc)$_2$, 20% (v/v) glycerol, 0.003% (v/v) IGEPAL CA-630, 0.5 mM BME, 0.033 mg ml⁻¹ recombinant albumin and 0.005 mg ml⁻¹ yeast tRNA). Unless other concentrations are specified, 0.75 nM of pre-diluted AGO–miRNA complex was pre-incubated with approximately 0.25 nM target RNAs in co-IP buffer (18 mM HEPES pH 7.4, 150 mM KOAc, 2 mM Mg(OAc)$_2$, 5% (v/v) glycerol, 0.01% (v/v) IGEPAL CA-630 and 0.5 mM BME) supplemented with blocking reagents (0.1 mg ml⁻¹ recombinant albumin and 0.01 mg ml⁻¹ yeast tRNA) at 37 °C for 30–60 min. Where specified,

heparin (Sigma-Aldrich, H3393), which reduces non-specific binding of RNA (especially longer RNA) to the beads, was also included in the blocking buffer and co-IP buffer. Heparin at 1 µg ml$^{-1}$ was included in assays shown in Figs. 1f, 3g and 4e and Extended Data Figs. 6d and 9c,f. Heparin at 10 µg ml$^{-1}$ was included in assays shown in Extended Data Figs. 1c and 8d. Heparin was not included in assays shown in Fig. 1e and Extended Data Figs. 1d,g, 2a–d and 8f–h. In the meantime, bait proteins ZSWIM8-3xHA and T6B-3xHA were pre-diluted in ZSWIM8 buffer (18 mM HEPES pH 7.4, 150 mM KOAc, 5% (v/v) glycerol, 0.01% (v/v) IGEPAL CA-630 and 0.5 mM TCEP).

Pre-incubated AGO–miRNA–target complex and pre-diluted bait proteins (or ZSWIM8 buffer) were mixed at a 1:1 ratio and incubated at 25 °C for 1 h, at a final volume of 30 µl per reaction. To reduce non-specific binding during the subsequent immunoprecipitation, any aggregation-prone species were removed by centrifugation at 17,000–21,000g for 5 min at 4 °C. Then, 5% of the supernatant was taken from the no-bait control to serve as input. Supernatants were otherwise resuspended with pre-blocked beads and incubated at 4 °C for 1.5 h, with intermittent shaking at 1,200 rpm for 15 s every 2 min. Following incubation, reactions were transferred to fresh tubes, and beads were washed four times with 125 µl of 18 mM HEPES pH 7.4, 150 mM KOAc, 1 mM Mg(OAc)$_2$, 5% (v/v) glycerol, 0.01% (v/v) IGEPAL CA-630 and 0.5 mM BME, with a tube change during the fourth wash. Immunoprecipitated species were eluted by resuspension of beads with gel loading buffer (8 M urea, 25 mM EDTA, 0.025% (w/v) xylene cyanol, 0.025% (w/v) bromophenol blue) and shaking at 1,000 rpm at 65 °C for 10 min. Eluates were run on a urea-polyacrylamide gel. The gel was frozen at −20 °C while exposing a phosphorimager plate, and radioactivity was subsequently imaged on a phosphorimager. Band intensities were quantified using ImageQuant TL (v.10.2); image background was estimated either by subtracting the mean signal of equal-sized rectangles drawn at regions with no bands, or, where indicated, by additionally examining the lane profile tool drawn across each lane. Signal from each band was then background-subtracted using the 0 nM ZSWIM8 sample and normalized to that of the T6B sample. The assay setup as described above was used to generate the data shown in Figs. 1e,f, 3g and 4e and Extended Data Figs. 1c,d,g, 2a–d, 6d,e, 8d,f–h and 9c,d,f. Uncropped scans of gels are provided in Supplementary Fig. 1.

**Filter-binding assays.** To measure intrinsic affinity of ZSWIM8 for RNAs, a dilution series of purified ZSWIM8-3xHA was incubated in excess with radiolabelled RNAs in 18 mM HEPES pH 7.4, 150 mM KOAc, 1 mM Mg(OAc)$_2$, 5% (v/v) glycerol, 0.01% (v/v) IGEPAL CA-630, 0.25 mM BME, 0.25 mM TCEP (Supelco, 646547), 0.1 mg ml$^{-1}$ recombinant albumin and 0.01 mg ml$^{-1}$ yeast tRNA, at 25 °C for 2 h.

Nitrocellulose (Amersham Protran, 0.45-µm pores; Cytiva, 10600062) and nylon (Amersham Hybond-XL; Fisher Scientific, 45001147) membrane filters were cut with a circle punch into discs of 0.5-inch diameter each and pre-equilibrated at room temperature for at least 20 min in 18 mM HEPES pH 7.4, 100 mM KOAc and 1 mM Mg(OAc)$_2$. For each sample, a nitrocellulose disc was stacked on top of a nylon disc, atop a circular pedestal mounted on a Visiprep SPE Vacuum Manifold (Supelco, 57250-U), set at approximately −20 kPa. Then, 10 µl of reaction was promptly applied to the stacked discs, followed by 100 µl of ice-cold wash buffer (18 mM HEPES pH 7.4, 100 mM KOAc, 1 mM Mg(OAc)$_2$ and 5 mM DTT). Filter membrane discs were separated with tweezers, air-dried, then exposed on a phosphorimager plate. Radioactivity was imaged on a Typhoon phosphorimager (Cytiva), and spot intensities were quantified using ImageQuant TL (v.10.2). The assay setup as described above was used to generate the data shown in Extended Data Fig. 2e.

**Analytical size-exclusion chromatography.** ZSWIM8–ELOB/C, ZSWIM8$^{mono}$–ELOB/C and ZSWIM8$^{\Delta dimer}$–ELOB/C (10 µM) were each incubated with CUL3–RBX1 (12 µM) in 25 mM HEPES, 150 mM NaCl and

1 mM TCEP, pH 7.5, at 4 °C for 30 min before analysis by size-exclusion chromatography (Superose 6, Cytiva). Samples were resolved on 4–22% SDS–PAGE gels, and gels were stained with Sypro Ruby (Sigma-Aldrich, S4942) before analysis by in-gel fluorescence. Analytical size-exclusion chromatograms and corresponding SDS–PAGE analyses are shown in Extended Data Fig. 3k. Uncropped scans of gels are provided in Supplementary Fig. 1.

**Bio-layer interferometry.** Bio-layer interferometry measurements were performed on an Octet K8 system (Sartorius) at 30 °C with shaking at 1,000 rpm. Concentrated proteins were diluted into BLI buffer (25 mM HEPES, 150 mM NaCl, 0.5 mg ml$^{-1}$ BSA and 1 mM TCEP, pH 7.5). Biotinylated cullin-NTDs were immobilized on streptavidin sensors (Sartorius, 18-5019). ZSWIM8 served as the analyte and was measured in seven dilutions from 650 nM to 0.9 nM. Raw data were processed in Octet Data Analysis HT software (v.13.0.1). As no dissociation was detected, the $K_D$ was estimated through fitting $R_{max}$ of association. Processed data were plotted and fit using a single-site binding model in GraphPad Prism (v.10.4.0).

**Sequence alignments of protein homologues.** Protein sequences were downloaded from UniProt[75]. Multiple-sequence alignment was conducted with the MUSCLE algorithm[76]. Maximum-likelihood phylogeny was calculated using FastTree 2.2 (ref. 77).

### Cryo-EM

**Sample preparation.** All concentrations stated refer to the final concentrations used. AGO2–miR-7 (7 µM) was mixed with a 120-nt fragment of the CYRANO trigger (8 µM) (Supplementary Table 1) and incubated at room temperature for 15 min. In the meantime, ZSWIM8–ELOB/C (6 µM) was mixed with CUL3(NTD) (6 µM) in 25 mM HEPES, 50 mM NaCl and 1 mM TCEP, pH 7.8. AGO2–miR-7–CYRANO was added and incubated on ice until plunge-freezing. Cu holey-carbon grids (R1.2/1.3, 200 mesh, Quantfoil, Q250-CR1.3) were freshly glow-discharged. Then, 3.5 µl of complex was applied, incubated for 5 s and blotted with a blot force of 2 and blot time of 2 s using a Vitrobot Mark IV (4 °C, 95% humidity) and plunge-frozen in liquid ethane.

**Electron microscopy.** A screening dataset was recorded on a Glacios cryo-transmission electron microscope at 200 kV using a Gatan Vista Alpine detector in counting mode and SerialEM software (v.4.1). The dataset consisted of 3,998 videos with a pixel size of 1.871 Å. Total exposure was set to 60 electrons per Å$^2$ (40 frames), and with a defocus range between −0.5 and −2.5 µm. A high-resolution dataset was recorded on a Titan Krios transmission electron microscope at 300 kV using a Gatan K3 detector in counting mode and Serial EM software. The dataset (26,355 micrographs) was recorded at a magnification of 105,000, pixel size 0.8512 Å, total exposure 58 electrons per Å$^2$ and a defocus range between −0.5 and −2.0 µm. A representative micrograph can be found in Supplementary Fig. 4.

**Data processing.** All datasets were processed in CryoSparc (v.4.6.2)[78] (Extended Data Table 1). Raw video files were motion-corrected and dose-weighted, followed by estimation of contrast-transfer functions. Exposures were manually curated to remove micrographs with damage or ice contamination. Particles were picked in CryoSparc without a template and with a template generated from the screening dataset. Heterogeneous refinements, 3D classification, global and local refinements, particle subtraction and post-processing were performed in CryoSparc. For all heterogeneous refinements, one decoy class was included. All final maps were post-processed using manual sharpening or DeepEMhancer. The processing scheme is shown in Supplementary Figs. 4 and 5. 3D variability analysis was performed in CryoSparc with six modes and a filter resolution of 10 Å. Examples of map-to-model fits are shown in Supplementary Fig. 6.

**Model building.** The high-resolution map of CUL3(NTD)–ZSWIM8–ELOB/C–AGO2–miR-7–CYRANO (map A, sharpened with a B factor of 89.8) was further refined using three focused refined maps showing high-resolution features (maps B, C and D). A composite map was constructed on the basis of focused refined and DeepEMhancer-sharpened maps (map E). Each focused map has been deposited, along with a composite map. Maps A, B, C and E were used to build ZSWIM8[NPAZ] and ZSWIM8[MID] interactions with AGO2, RNA and C-terminal HEAT repeats. Maps A, C, D and E were used to build CUL3(NTD), ELOB, ELOC and the ZSWIM8 E3 superdomain. Structurally flexible RBEs were modelled using a combination of unsharpened maps (map F) and low-pass-filtered maps (map G). An initial model was generated by docking de novo-folded ZSWIM8 (AlphaFold3), CUL3 (PDB 5NLB), ELOB/C (PDB 1LM8) and AGO2 (PDB 6NIT) into the composite map. Iterative manual model building in COOT and real-space refinement using Phenix.refine were performed until a satisfactory map-to-model correlation was achieved[79,80]. Owing to the structural heterogeneity of CUL3, only the first 150 residues were modelled. For ZSWIM8, side chains were built for residues with interpretable density. For CUL3 and ELOB/C, side-chain residues were included on the basis of input models. Three long unstructured regions in ZSWIM8 were not modelled owing to the absence of any interpretable density. The TDMD sensor domain of ZSWIM8[NPAZ] includes a disconnected density that was modelled to contain ZSWIM8 residues Q39–R45 on the basis of AlphaFold3 predictions. RBEs 1–3 on ZSWIM8[NPAZ] were modelled with side chains on the basis of sharpened and unsharpened maps. RBEs 1–3 on ZSWIM8[MID] were modelled without side chains owing to inferior densities. C-terminal HEAT repeats were modelled on the basis of AlphaFold3 predictions without side chains owing to a lack of interpretable side-chain densities. Trigger RNA base U44 was poorly resolved in sharpened maps; its flipped-out position was modelled on the basis of unsharpened maps. The density for trigger RNA base A43 indicates a syn conformation and was modelled as such. All figures were generated using ChimeraX (v.1.8–v.1.9)[81]. All densities shown in figures were prepared using the composite map, unless otherwise stated.

## Cellular assays

**Cell lines.** For generating clonal *ZSWIM8*-knockout K562s, parental K562 cells harbouring a miR-7-sensitive GFP reporter[10] were transiently transfected with a plasmid derived from PX458 (Addgene, 48138) expressing pSpCas9-2a-BFP and either one or two single guide RNAs (sgRNAs) targeting the *ZSWIM8* gene (Supplementary Table 4), followed by single-cell sorting of BFP-positive cells. Clones were screened for disruption of all *ZSWIM8* alleles by amplification of the locus followed by nanopore sequencing of the PCR amplicon.

When generating clonal *Zswim8*-knockout MEFs, we observed substantial heterogeneity in clonal MEF populations derived from polyclonal parental populations, which was consistent with previous observations[82]. To minimize this clone-to-clone variability, we first generated a clonal parental population of WT MEFs by single-cell sorting of GFP-positive cells from MEFs[83] that had been transiently transfected with a plasmid expressing pSpCas9-2a-GFP and a non-targeting sgRNA (Addgene, 48138; Supplementary Table 4). After isolating clonal lines, these parental cells were transiently transfected with a plasmid expressing pSpCas9-2a-GFP and an sgRNA targeting the *Zswim8* gene (Addgene, 48138; Supplementary Table 4), followed by single-cell sorting of GFP-positive cells. Clones were screened for disruption of all *Zswim8* alleles by amplification of the locus followed by Sanger sequencing of the PCR amplicon. Clonal WT MEFs were generated in parallel from the same parental clonal lines, except using a plasmid expressing pSpCas9-2a-GFP and a non-targeting sgRNA (Supplementary Table 4).

Polyclonal knock-in K562 lines stably expressing doxycycline-inducible *ZSWIM8* transgenes were generated by PiggyBac transposition into clonal *ZSWIM8*-knockout K562 cells containing a miR-7-sensitive GFP reporter, described above. The WT ZSWIM8 CDS was cloned into PB-TetON (Addgene, 97421), and ZSWIM8 variants were generated

from this WT plasmid. For each well in a six-well plate, approximately 600,000 cells were transfected with 1 μg of the PiggyBac transposon plasmid and 0.4 μg of the Super PiggyBac transposase plasmid (a gift from A. Hansen) using Lipofectamine 2000 and Opti-MEM according to the manufacturer's instructions. After 48 h, cells were passaged 1:2 into fresh media containing 3 μg ml⁻¹ puromycin (Gibco, A1113803). Subsequently, selection medium was refreshed every 3 d as cells were expanded up to 10-cm plates.

**ZSWIM8 intracellular rescue assay.** Following generation of *ZSWIM8*-knockout reporter K562 cells stably expressing doxycycline-inducible *ZSWIM8* transgenes, as described above, about 400,000 cells were plated per well in a six-well plate with medium containing 3 μg ml⁻¹ puromycin and 33.3 ng ml⁻¹ doxycycline (Takara Bio, 631311). After 48 h, cells were expanded to 10-cm plates and medium containing 3 μg ml⁻¹ puromycin and 33.3 ng ml⁻¹ doxycycline was refreshed. After another 48 h, about 75% of the culture was set aside for northern analysis and about 12.5% was set aside for immunoblotting. The remaining 12.5% was concentrated to 2 million cells per ml in media and subjected to flow cytometry. For each sample, approximately 20,000 live cells were analysed using a BD LSR Fortessa instrument (BD Biosciences) and BD FACSDiva software (v.9.0). Subsequent analysis was performed using FlowJo (v.10.10.0). Gating strategies for flow cytometry analysis are shown in Supplementary Fig. 7. Quantified results of the ZSWIM8 intracellular rescue assay are shown in Figs. 3c and 4c and Extended Data Figs. 3f,n and 4b,j. Immunoblots of ZSWIM8 protein variants are shown in Supplementary Fig. 8.

**RNA isolation.** Total RNA was extracted with TRI Reagent (Invitrogen, AM9738) according to the manufacturer's instructions. K562 cells were pelleted by centrifugation, washed with PBS and resuspended in 1 ml of TRI Reagent. GlycoBlue (Invitrogen, AM9516) was added to a final 25 μg ml⁻¹ to facilitate precipitation and visualization of RNA pellets. After isopropanol precipitation and one 75% ethanol wash, pellets were resuspended in water and quantified by nanodrop.

**Northern blots.** Unless otherwise specified, 10 μg of total RNA was resolved on a denaturing 15% polyacrylamide gel and transferred onto a Hybond-N+ membrane (Cytiva, RPN303B) using a semi-dry transfer apparatus (Bio-Rad). The membrane was then incubated at 65 °C for 1–2 h with EDC (N-(3-dimethylaminopropyl)-N′-ethylcarbodiimide; Thermo Scientific, 22980) diluted in 1-methylimidazole, which chemically crosslinked 5′ phosphates to the membrane. Blots were hybridized to radiolabelled oligonucleotide probes (DNA or LNA) (Supplementary Table 4). Stripping of probes before re-probing was performed by incubation in boiling 0.04% (v/v) SDS with shaking. A detailed protocol for northern-blot analysis of small RNAs is available at http://bartel-lab.wi.mit.edu/protocols.html. Results were analysed on a Typhoon phosphorimager, using ImageQuant TL (v.10.2) for quantification of band intensities.

**Immunoblots.** K562 cells were pelleted by centrifugation, washed with PBS and lysed using Pierce IP Lysis Buffer containing 1 tablet of cOmplete Mini EDTA-free Protease Inhibitor Cocktail per 10 ml of buffer. Following 15 min of incubation in lysis buffer at 4 °C, lysates were clarified by centrifugation at 17,000g for 15 min at 4 °C and quantified by Bradford assay. Clarified lysates were incubated at 65 °C for 10 min in the presence of 1 × NuPAGE LDS sample buffer (Invitrogen, NP0007) and 2.5% (v/v) BME. Next, 8–12 μg of lysate was resolved on NuPAGE 4–12% Bis-Tris protein gels (Invitrogen, WG1403) with 1 × MES SDS running buffer (Invitrogen, NP0002) at 180 V for 50 min, and then transferred onto 0.45-μm PVDF membranes (Invitrogen, STM2006) at 25 V for 1 h on ice using a Midi Gel Tank (Invitrogen, STM1001) and Midi Blot Module (Invitrogen, STM2001). Membranes were then blocked with 5% (w/v) BSA (RPI, A30075-100.0) dissolved in PBS containing 0.1% (v/v) Tween

20 (Sigma-Aldrich, P1379) (PBS-T) for 1 h at room temperature, and then incubated with primary antibodies overnight at 4 °C. The following day, membranes were washed with 0.1% PBS-T at room temperature and incubated with near-infrared fluorescent secondary antibodies (LI-COR) for 1 h at room temperature, followed by washes and detection on an Odyssey CLx imager (LI-COR). Primary antibodies and their dilutions included: rabbit polyclonal anti-ZSWIM8 (1:400; Invitrogen, PA5-59492), rabbit monoclonal anti-HA (1:5,000; Cell Signaling Technology, C29F4, 3724) and mouse monoclonal anti-GAPDH (1:2,000; Invitrogen, GA1R, MA5-15738). Near-infrared fluorescent secondary antibodies and their dilutions included: IRDye 680RD goat anti-rabbit (1:10,000, LI-COR, 926-68071) and IRDye 800CW goat anti-mouse (1:10,000, LI-COR, 926-32210). Protein levels were quantified in ImageeStudio (LI-COR, v.6.1.0.79).

**Lentiviral transduction.** Lentiviral production and transduction were performed as described[11]. In brief, transfer plasmids expressing WT AGO2, as well as R40 and $R_{C5+6}$ variants (Supplementary Table 2), were prepared as described[11]. Transfer plasmids expressing new AGO2 variants were generated from the WT pLJM1-3xHA-AGO2 plasmid and prepared from NEB Stable Competent *E. coli* (NEB, C3040). For each well in a six-well plate, about 170,000 HEK293FT cells per cm$^2$ were reverse-transfected with 1.4 µg of transfer plasmid, 0.94 µg of pCMV-dR8.91 packaging plasmid (a gift from J. Weissmann) and 0.47 µg of pMD2.G envelope plasmid (Addgene, 12259) using Lipofectamine 2000 and Opti-MEM. After 16 h, medium was refreshed. After another 48 h, medium was collected and centrifuged at 500$g$ for 10 min to pellet debris. To infect MEFs, 500 µl of virus-containing supernatant (approximately 40% of the total) was combined with 1 ml of media and 12 µg ml$^{-1}$ polybrene (Santa Cruz Biotechnology, SC-134220), and then added to one well of a six-well plate containing MEFs that were plated 24 h before at around 15,600 cells per cm$^2$, for a final 8 µg ml$^{-1}$ polybrene. Plates were centrifuged at 1,200$g$ for 1.5 h at about 30 °C, and then returned to the 37 °C incubator. The following day, cells were expanded to 10-cm plates with media containing 4 µg ml$^{-1}$ puromycin. Puromycin selection continued for another 6 d, refreshing selection media every 2 d. Cells typically reached confluency after 3 d of selection and were left confluent for another 3 d.

**AGO2 intracellular co-IP assays.** Experiments testing the effect of AGO2 mutations on intracellular levels of associated miRNAs were performed as described[11]. In brief, lentiviral production and transduction of MEFs with pLJM1-3xHA-AGO2 constructs were performed as described above. Following 6 d of puromycin selection (which included 3 d of contact inhibition), approximately 5 million cells per sample were washed with PBS, dissociated using 1 × TrypLE Express Enzyme (Gibco, 12604039), washed with PBS again and lysed with 600 µl of Pierce IP Lysis Buffer (Thermo Scientific, 87788) containing 1 tablet of cOmplete Mini EDTA-free Protease Inhibitor Cocktail per 10 ml of buffer. Following 15 min of end-over-end rotation in lysis buffer at 4 °C, lysate was clarified by centrifugation at 17,000$g$ for 15 min at 4 °C. After setting aside a portion for immunoblotting, each clarified lysate was incubated with 50 µl of Pierce anti-HA magnetic beads with end-over-end rotation at 4 °C for 2 h. Beads were then washed three times with 250 µl of lysis buffer, and co-immunoprecipitating RNA was extracted with TRI Reagent added directly to the beads. Following RNA isolation, one-third of the total volume was used for northern analysis. Representative northern blots are shown in Extended Data Fig. 6b, and the corresponding quantification is shown in Extended Data Fig. 6c. Immunoblots of AGO2 protein variants are shown in Supplementary Fig. 11.

**sRNA-seq.** AGO protein, along with bound miRNAs, was isolated from MEFs and S2 cells using a FLAG-tagged T6B peptide[73], as described[74], except that a 3xFLAG-SUMO-tagged peptide (hereafter referred to as T6B-3xFLAG peptide) was used instead of the original 1xFLAG-GST. Its purification is described above. *Zswim8*-knockout and control MEF cell lines, generated as described above, were allowed to reach confluency, at which point medium was refreshed and cells were left confluent for 5 d before collection. *Zswim8*-knockout and control S2 cells[11] were collected once they reached confluency. Both MEFs and S2 cells were flash-frozen as cell pellets before lysis. Anti-FLAG M2 magnetic beads were pre-coupled to T6B-3xFLAG peptide, washed to remove unbound peptide and then incubated with clarified lysate from MEFs or S2 cells that were lysed in 50 mM Tris pH 8.0, 150 mM NaCl, 5 mM EDTA, 10% (v/v) glycerol and 0.5% (v/v) NP-40, with 1 tablet of cOmplete Mini EDTA-free Protease Inhibitor Cocktail per 10 ml of buffer. Beads were washed three times with lysis buffer and once with PBS, and bound RNA was extracted directly from beads with TRI Reagent, as described above.

sRNA-seq libraries were prepared as described (http://bartellab.wi.mit.edu/protocols.html), with the following changes. sRNA-seq libraries were generated using 40% of the RNA isolated from a T6B pull-down. Instead of an 18-nt size marker, a 12-nt size marker (Supplementary Table 4) was used in all size-selection steps to capture shorter miRNA isoforms. For libraries generated from S2 cells, 2S ribosomal RNA was depleted from samples after the initial size selection using subtractive hybridization with a biotinylated antisense oligo, as described[74]. Libraries were sequenced on the Illumina NovaSeq 6000 platform with 100-nt single-end reads.

Processing of sequencing reads was done by trimming adaptors with cutadapt (v.4.8)[84], filtering out reads with a quality score of 30 or below with FASTX Toolkit v.0.0.14 and then string-matching the first 13 nt of each read against a dictionary of miRNA names and sequences derived mainly from TargetScanFly7 and TargetScanMouse7 (ref. 85). For miRNAs whose sequences in the TargetScan annotations differed from those in miRGeneDB 3.0 annotations[86], which was most common for lowly expressed and/or non-conserved miRNAs, sequences from miRGeneDB were added to the mapping dictionaries as processing variants (for example, mmu-let-7g-3p.1 and mmu-let-7g-3p.2). miRNAs sharing the same first 13 nt were collapsed into a single entry, and the full names of the collapsed species are provided in Supplementary Table 5. The length of each mapped miRNA was recorded, generating a table of read counts for each miRNA at each size, ranging from 13 to 31 nt.

To identify miRNAs accumulating in *Zswim8*-knockout lines compared with control lines, reads associated with all isoforms of a given miRNA were summed, filtered on the basis of a read cutoff of greater than five reads per library for at least two of the knockout lines and 30 reads across all six lines (three knockouts and three controls) and normalized for sequencing depth using the estimateSizeFactors function of DESeq2 (v.1.38.3)[87]. Fold-changes between *Zswim8*-knockout and control lines were then estimated using summed count values with DESeq2 v.1.38.3 without use of the lfcShrink function[87]. The mean of normalized abundances in the control lines was taken as the expression in WT cells. To identify miRNAs that accumulated significantly after loss of ZSWIM8, we applied a modified BBUM model (false discovery rate-adjusted $P < 0.05$)[88].

Three approaches were used to identify miRNAs that were not classified as significantly up-regulated but nonetheless were potentially ZSWIM8 sensitive, with the goal of stringently removing from subsequent analyses any miRNAs whose dominant length isoform (typically 22 nt) could be subject to TDMD. First, using the fold-changes calculated by summing the counts associated with all isoforms of a given miRNA, miRNAs with $P_{adj}$ less than 0.05, in which the $P$ values were adjusted by the Benjamini–Hochberg method by DESeq2, and fold-changes (log$_2$) greater than zero were considered potentially ZSWIM8-sensitive. Second, the fold-change of each miRNA was compared with that of the co-produced passenger miRNA derived from the same precursor hairpin, as described[12]. miRNAs were classified as potentially ZSWIM8-sensitive if their log$_2$ fold-changes and those of their passenger strands were separated beyond their respective standard errors, as calculated by DESeq2, as would be expected if ZSWIM8 altered the degradation rate of the miRNA and not the production rate.

For miRNAs with more than one potential passenger strand, read counts attributed to each passenger were summed, and the $\log_2$ fold-change of the summed counts was used in the comparison. For this analysis, miRNAs were not subject to the read cutoff described previously, as many passenger miRNAs were lowly expressed. Third, miRNAs in MEFs were annotated as potentially ZSWIM8-sensitive if they were previously called ZSWIM8-sensitive in mouse tissues[12], and miRNAs in S2 cells were annotated as potentially ZSWIM8-sensitive if they were called sensitive in S2 cells[11] or *Drosophila* embryos[20]. The isoforms of miRNAs meeting any of these three conditions were excluded from the analyses in Extended Data Fig. 7. Analogous analyses for these miRNAs whose levels are significantly or potentially affected by ZSWIM8 are shown in Supplementary Fig. 15.

To investigate the relationship between miRNA length and ZSWIM8-dependent degradation, fold-changes observed after loss of ZSWIM8 were calculated for individual length isoforms of the remaining miRNAs that had not been excluded owing to potential ZSWIM8 sensitivity of their major isoforms. To this end, read counts corresponding to each isoform were first normalized using the size factors previously computed on the basis of the summed counts, and fold-changes ($\log_2$) were calculated with DESeq2. Finally, the median $\log_2$ fold-change of all 22-nt species was calculated and subtracted from the $\log_2$ fold-changes of all species. This normalization centres the distribution of $\log_2$ fold-changes for the 22-nt isoforms at zero, as would be expected if they were not sensitive to loss of ZSWIM8.

### Reporting summary

Further information on research design is available in the Nature Portfolio Reporting Summary linked to this article.

### Data availability

Extended Data Figs. 1–9, Extended Data Table 1 and Supplementary Information are available for this paper. The atomic coordinates have been deposited in the PDB with accession code 9RWZ, and electron microscopy maps have been deposited with the Electron Microscopy Data Bank with accession codes EMD-54348, EMD-54349, EMD-54350, EMD-54351 and EMD-54352. Raw and processed sRNA-seq data have been deposited in the Gene Expression Omnibus with accession code GSE303177. Uncropped gel images are included as Supplementary Fig. 1. Source data are provided with this paper.

### Code availability

Original code for the analysis of sRNA-seq data is available publicly at https://github.com/lwblodgett/ZSWIM8_sensitivity_of_miRNA_iso-forms.git and is also available at Zenodo (https://doi.org/10.5281/zenodo.18265217)[89].

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

**Acknowledgements** We thank the Max-Planck Institute of Biochemistry Protein Production Facility (RRID:SCR_025741) for protein expression, the Biochemistry Core Facility (RRID:SCR_025743) for peptide synthesis and access to BLI instruments, the Mass Spectrometry Facility (RRID:SCR_025745) and the Cryo-EM facility (RRID:SCR_025744) with D. Bollschweiler and T. Schäfer for their help with cryo-EM analysis. We thank the Whitehead Institute Flow Cytometry Core for assistance with flow cytometry and the Whitehead Institute Genome Technology Core for high-throughput sequencing. We thank J. Mendell for sharing cell lines. We thank J. Weissman, K. Rogala and A. Hansen for sharing plasmids. This study was supported by grants from the European Union (ERC, UPSmeetMet, grant no. 101098161), Deutsche Forschungsgemeinschaft (DFG, German Research Foundation) and the NIH (grant no. GM118135). J.F. was supported by a Postdoctoral Fellowship of the Peter & Traudl Engelhorn Stiftung. E.S. was supported by the National Science Foundation Graduate Research Fellowship Program. P.Y.W. was supported by the MIT Office of Graduate Education Fellowship. D.H.L. was a Howard Hughes Medical Institute fellow of the Damon Runyon Cancer Research Foundation (grant no. DRG-2345-18). D.P.B. is an investigator of the Howard Hughes Medical Institute. We thank J. Kellerman and F. Çivril for their support. We thank L. Kiss, J. R. Prabu, S. Maiwald, J. Liwocha, L. Henneberg, C. Shi, B. Wierbowski, L. Elcavage, M. Frank, A. Latifkar and other members of the Schulman and Bartel labs for fruitful discussions.

**Author contributions** J.F. performed in vitro ubiquitylation assays. S.V.G. performed insect cell expression. E.S. and P.Y.W. performed in vitro co-IP assays. E.S. performed cellular reporter assays. E.S. and D.H.L. performed cellular co-IP assays. J.F. collected, processed, refined and performed in-depth analyses of the cryo-EM structure. E.S., P.Y.W., D.H.L., B.A.S. and D.P.B. also analysed the cryo-EM structure. L.W.B. performed sRNA sequencing and associated analyses. J.F., E.S., B.A.S. and D.P.B. prepared the manuscript with input from all other authors. The study was supervised by B.A.S. and D.P.B.

**Funding** Open access funding provided by Max Planck Society.

**Competing interests** D.P.B. has equity in Alnylam Pharmaceuticals, where he is a co-founder and an advisor. B.A.S. is a member of the scientific advisory boards of Proxygen and Lyterian. J.F. provides consultancy to Serac Biosciences. The other authors declare no competing interests.

**Additional information**
**Correspondence and requests for materials** should be addressed to Brenda A. Schulman or David P. Bartel.

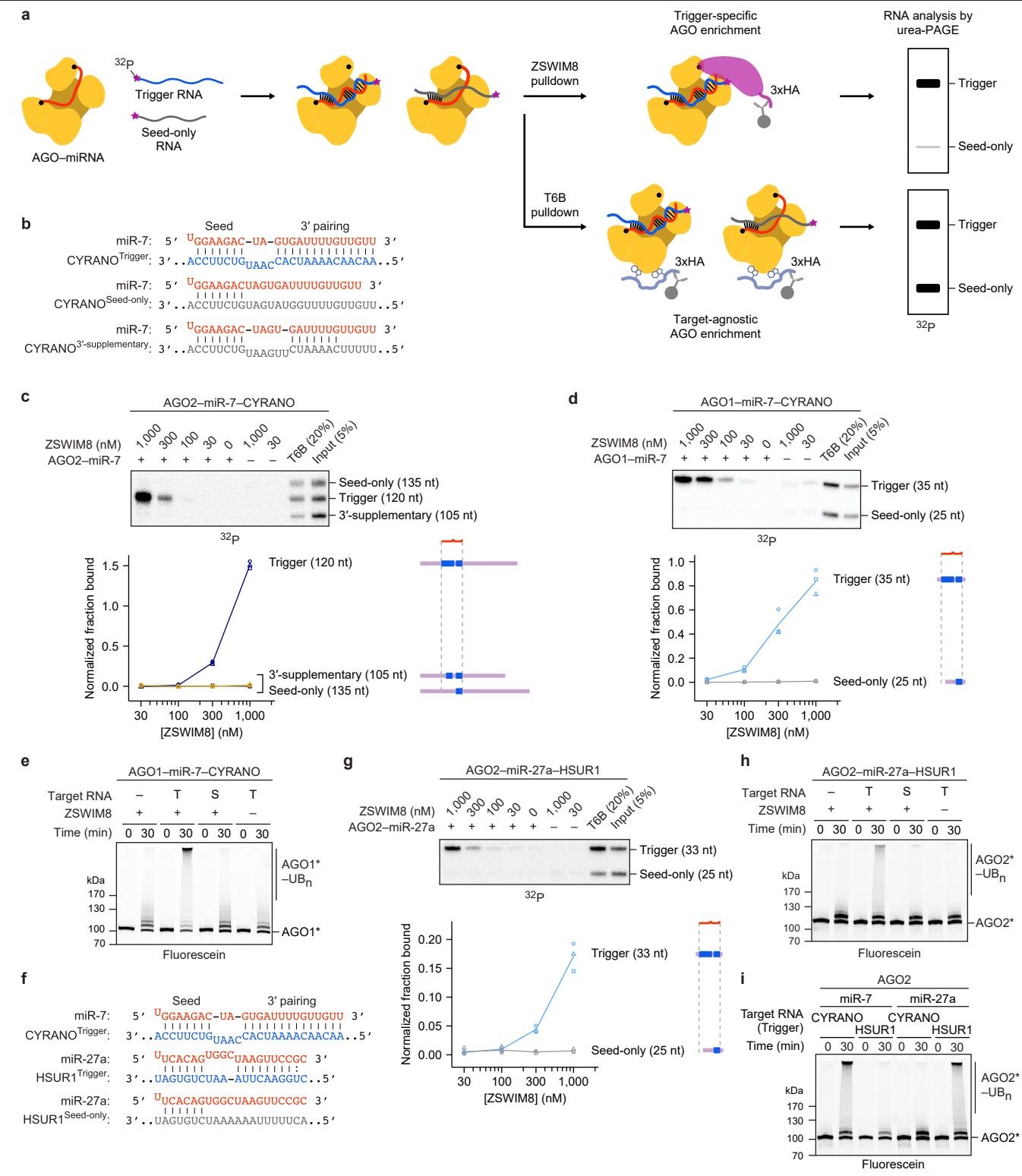

**Extended Data Fig. 1** | See next page for caption.

**Extended Data Fig. 1 | ZSWIM8 binds to AGO–miRNA–trigger complexes to direct the polyubiquitylation of AGO. a**, Schematic of the in vitro co-IP assay for detecting binding of ZSWIM8 to AGO–miRNA–target complexes. AGO–miRNA complexes are incubated with a mixture of 5′-radiolabelled target RNAs of different lengths. These RNAs are in large excess over the $K_D$ values observed between AGO–miRNA and target RNA, and the AGO–miRNA complex is in slight excess over the total concentration of the target RNAs. The target RNAs are designed to differ with respect to their pairing to the miRNA, such as seed-only or trigger pairing (shown), or with respect to the amount of RNA flanking the site (not shown). The AGO–miRNA–target ternary complexes are then incubated with an excess of either 3xHA-tagged ZSWIM8 or a 3xHA-tagged T6B peptide (derived from TNRC6B). Complexes associating with ZSWIM8 or T6B are enriched by 3xHA IP. Following washes, IPs are eluted and the co-IPed target RNAs are resolved on a denaturing gel. Radiolabelled species are visualized by phosphor imaging. The T6B IP is designed to detect any differences in AGO–miRNA binding to different target RNAs, because T6B is expected to bind AGO–miRNA complexes irrespective of their bound target RNA. Thus, for quantification, signal from each band is background-subtracted using the 0 nM ZSWIM8 sample and then normalized to that of the T6B sample. **b**, Diagrams showing pairing between miR-7 and the miRNA-binding regions of CYRANO^Trigger, CYRANO^Seed-only, and CYRANO^3′-supplementary target RNAs with their cognate miRNAs used for in vitro co-IP assays. Vertical lines represent W–C–F base pairing. **c**, The relative effects of trigger pairing, 3′-supplementary pairing, and seed-only pairing on ZSWIM8 binding to AGO2–miR-7 complexes. Top: Shown are results from the in vitro co-IP assay with ZSWIM8 and AGO2–miR-7 mixed with radiolabelled target RNAs derived from the CYRANO trigger containing either trigger, seed-only, or 3′-supplementary pairing to miR-7 (Supplementary Table 1). This panel is as in Fig. 1f, except 10 nM of AGO2–miR-7 and ~1 nM of target RNAs were used (instead of 0.75 nM of AGO2–miR-7 and ~0.25 nM of target RNAs), and 10 μg/mL of heparin was included (instead of 1 μg/mL of heparin). Bottom: Quantification of radiolabelled co-IPed RNA in the assay shown above. The lane-profile tool was used to better quantify the weak signal representing binding to seed-only and 3′-supplementary targets; otherwise, as in Fig. 1f. $n = 3$ technical replicates. This quantification revealed a 100-fold

preference for co-IP of target RNAs with trigger pairing over 3′-supplementary pairing, and a larger preference for trigger pairing over seed-only pairing, although this larger preference was difficult to quantify due to the very low signal for co-IP of the seed-only target. **d**, Biochemical reconstitution of selective ZSWIM8 binding to AGO1–miR-7–CYRANO^Trigger complexes. This panel is as in Fig. 1e, except human AGO2 was replaced by human AGO1. In addition, 6 nM of AGO1–miR-7 and ~1 nM of target RNAs were used instead of 0.75 nM of AGO2–miR-7 and ~0.25 nM of target RNAs. $n = 3$ technical replicates. **e**, Biochemical reconstitution of trigger- and ZSWIM8-dependent polyubiquitylation of AGO1 using the miR-7–CYRANO miRNA–trigger pair. This panel is as in Fig. 1c, except fluorescent AGO2 was replaced by fluorescent AGO1. Trigger and seed-only variants of CYRANO are indicated by T and S, respectively. Shown is a representative experiment; $n = 2$ technical replicates. **f**, Diagrams showing pairing of miRNA-binding regions of CYRANO^Trigger, HSUR1^Trigger, and HSUR1^Seed-only target RNAs with their cognate miRNAs used for in vitro ubiquitylation and co-IP assays. Dots represent G:U wobble; otherwise as in **b**. The miR-27a–HSUR1 miRNA–trigger pair represents a founding example of TDMD, in which a viral noncoding RNA directs the degradation of a host miRNA[7]. **g**, Biochemical reconstitution of selective ZSWIM8 binding to AGO2–miR-27a–HSUR1^Trigger complexes. This panel is as in Fig. 1e, except miR-7 was replaced by miR-27a, and CYRANO was replaced by HSUR1 (panel **f**). In addition, 10 nM of AGO2–miR-7 and ~1 nM of target RNAs were used instead of 0.75 nM of AGO2–miR-7 and ~0.25 nM of target RNAs. $n = 3$ technical replicates. **h**, Biochemical reconstitution of trigger- and ZSWIM8-dependent polyubiquitylation of AGO2 using the miR-27a–HSUR1 miRNA–trigger pair. This panel is as in Fig. 1c, except miR-7 was replaced by miR-27a, and CYRANO was replaced by HSUR1 (panel **f**). Trigger and seed-only variants of HSUR1 are indicated by T and S, respectively. Shown is a representative experiment; $n = 2$ technical replicates. **i**, Requirement for cognate miRNA–trigger pairing for AGO2 polyubiquitylation. Shown is in vitro ubiquitylation of AGO2 comparing activity with different combinations of miRNA–trigger pairs using miRNAs miR-7 and miR-27a, and trigger RNAs CYRANO and HSUR1. Otherwise, this panel is as in Fig. 1c. CYRANO trigger (120 nt) and HSUR1 trigger (144 nt) were used as target RNAs (Supplementary Table 1). Shown is a representative experiment; $n = 2$ technical replicates.

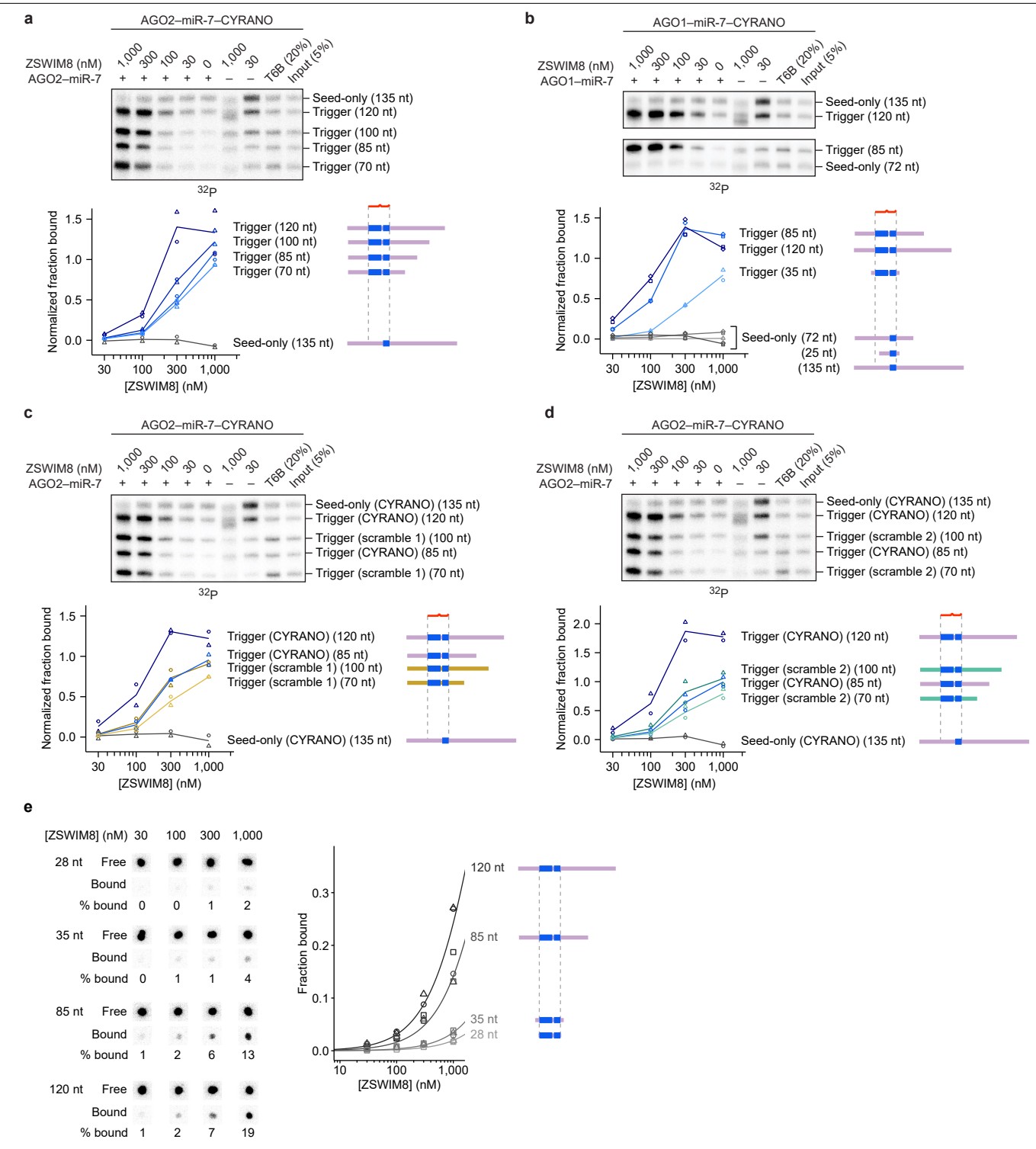

**Extended Data Fig. 2** | See next page for caption.

**Extended Data Fig. 2 | RNA flanking the trigger site contributes to ZSWIM8 affinity. a**, Effect of finer increments of trigger RNA flanking the 3′ end of the miRNA-binding site on ZSWIM8 binding to AGO2–miR-7–CYRANO. Shown is an in vitro co-IP assay like that in Fig. 1e, except target RNAs containing trigger pairing had variable amounts CYRANO sequence flanking the 3′ end of the miRNA-binding site. In addition, 10 nM of AGO2–miR-7 and -1 nM of target RNAs were used instead of 0.75 nM of AGO2–miR-7 and -0.25 nM of target RNAs. $n = 2$ technical replicates. **b**, Effect of trigger RNA flanking the miRNA-binding site on ZSWIM8 binding to AGO1–miR-7–CYRANO. Shown is an in vitro co-IP assay like that in **a**, except that 6 nM of AGO1–miR-7 was used instead of 10 nM of AGO2–miR-7. $n = 2$ technical replicates. Results from Extended Data Fig. 1d are replotted for comparison. **c**, Sequence specificity of the contribution of trigger flanking sequences to ZSWIM8 binding to AGO2–miR-7–CYRANO. Shown is an in vitro co-IP assay like that in **a**, except the flanking sequences were either the native CYRANO sequence, or a scrambled sequence maintaining the nucleotide composition of the original CYRANO sequence (scramble 1). $n = 2$ technical replicates. **d**, Sequence specificity of the contribution of trigger flanking sequences to ZSWIM8 binding to AGO2–miR-7–CYRANO. This panel is as in **c**, except a different scramble combination was used (scramble 2). $n = 2$ technical replicates. **e**, Intrinsic ZSWIM8 affinity for RNA alone. Left: Measurements of RNA of different lengths binding to ZSWIM8 in the absence of AGO–miRNA, as detected by filter binding. Filters were stacked such that the binding reaction passed first through the nitrocellulose filter (which binds proteins and protein-associated RNA) and then through the nylon filter (which binds free RNA). Shown are phosphorimager scans of the filters used to determine the fraction of RNA bound for each concentration of ZSWIM8. For each RNA, the top row shows the amount of radiolabelled RNA designated as free RNA because it passed through the nitrocellulose filter and bound to the nylon filter. The bottom row shows the amount of radiolabelled RNA designated as bound to ZSWIM8 because it was retained with ZSWIM8 on the nitrocellulose filter. Right: Fraction of RNA bound as a function of ZSWIM8 concentration, as detected by filter binding on the left. Spot intensities were background-subtracted, and the bound RNA signal was normalized to the sum of the free and bound RNA signal. The symbols show data from independent measurements. $n = 3$ technical replicates. Curves were fit to a standard fraction-bound curve for visualization, with the plateau constrained to 1.

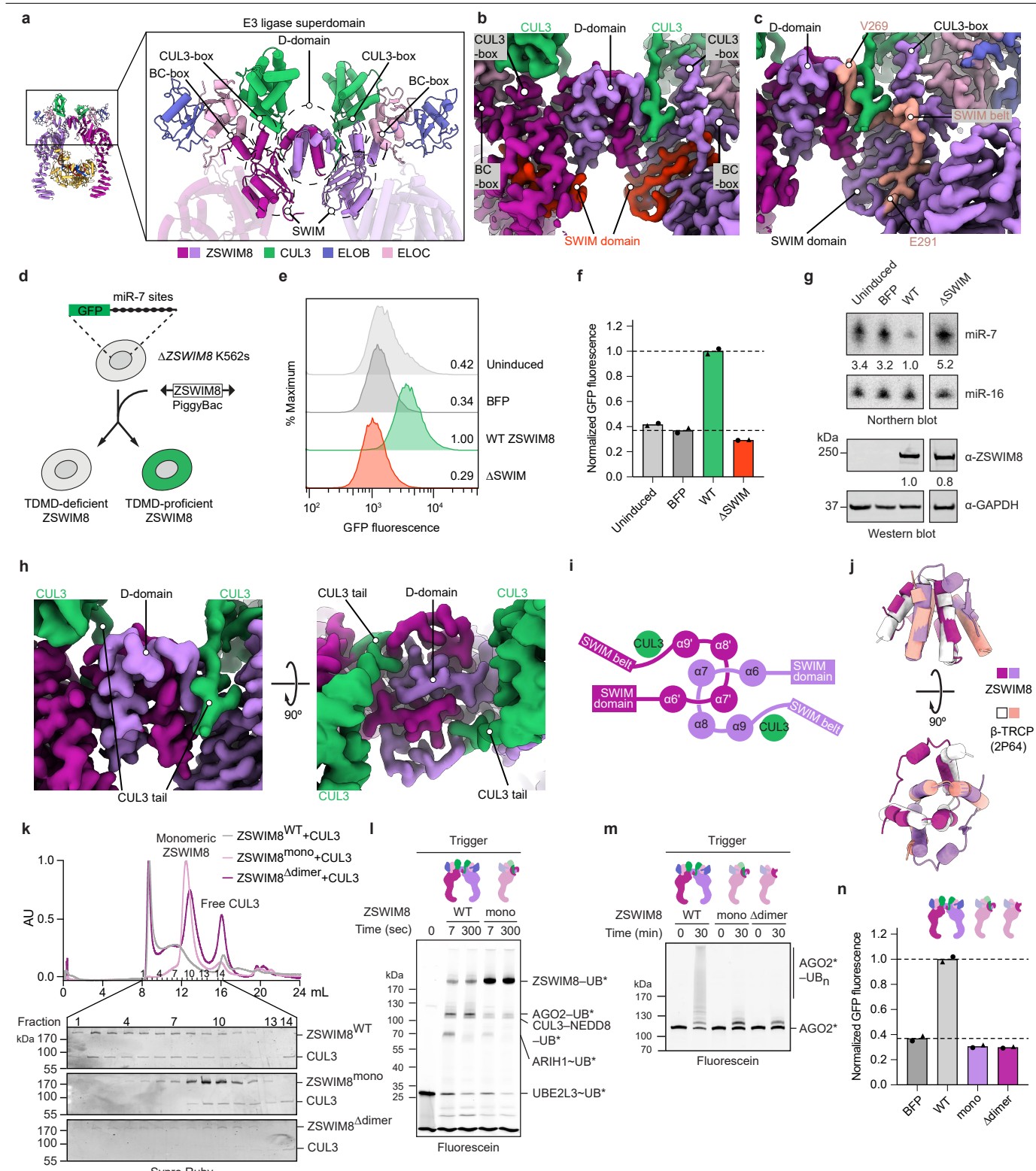

**Extended Data Fig. 3** | See next page for caption.

**Extended Data Fig. 3 | ZSWIM8 contains a dimeric E3 superdomain. a**, Cartoon representation showing the ZSWIM8 dimeric E3 superdomain and the individual domains associated with it. The E3 superdomain consists of the D-domain, CUL3-box, BC-box, and SWIM domain. **b**, Density showing the SWIM domain and its organization of the E3 superdomain. The SWIM domain is highlighted in red. Density was derived from the composite map. **c**, Density showing the SWIM belt, which connects the D-domain and SWIM domain to the TDMD sensor domain. Density showing the SWIM belt is highlighted in salmon. Density was derived from the composite map. **d**, Schematic of the intracellular TDMD reporter assay. The assay is performed using a clonal *ZSWIM8*-knockout K562 cell line that expresses a GFP reporter containing multiple copies of a miR-7 slicing site[10]. Lack of TDMD in these cells results in elevated levels of endogenous miR-7, which leads to increased repression of *GFP* reporter mRNA, causing reduced GFP fluorescence. To assay the activity of a ZSWIM8 variant, it is introduced using the PiggyBac system and expressed under a doxycycline-inducible promoter, and then its effect on GFP levels is measured by flow cytometry. ZSWIM8 variants proficient at TDMD rescue miR-7 degradation, resulting in increased GFP levels, whereas ZSWIM8 variants deficient in TDMD fail to decrease miR-7 levels, resulting in a lack of increase in GFP levels. **e**, Validation of the intracellular TDMD reporter assay. GFP levels in cells containing WT *ZSWIM8* without doxycycline induction (uninduced, light gray) were determined, as were those in cells expressing BFP (dark gray), WT ZSWIM8 (green), and a variant of ZSWIM8 with alanine substitutions in the SWIM domain (red). To the right of each histogram are median GFP values, each normalized to that of WT ZSWIM8. ZSWIM8 expression levels were detected by western blotting (Supplementary Fig. 8). **f**, Results of two biological replicates of the intracellular TDMD reporter assay in **e**. Each point represents the median GFP fluorescence, and each bar height represents the mean of these measurements from two independently derived PiggyBac lines. The top dashed line is drawn at the normalized mean GFP level (1.00) in cells expressing WT ZSWIM8, and the bottom dashed line is drawn at the normalized mean GFP level (0.37) in cells expressing BFP. **g**, Levels of miR-7 and ZSWIM8 in reporter cells used to generate the measurements in **e**. For each cell population, northern blots measured the levels of miR-7 and miR-16 (a control miRNA), and western blots detected expression of ZSWIM8 protein. Numbers in northern blots show the ratio of miR-7:miR-16, normalized to that of cells expressing WT ZSWIM8. Numbers in western blots show the ratio of ZSWIM8:GAPDH, normalized to that of cells expressing WT ZSWIM8. **h**, Density showing the ZSWIM8 D-domain and the associated CUL3 N-terminal tail. Density was derived from the composite map. **i**, Cartoon representation illustrating the knot-like assembly of the ZSWIM8 D-domain, along with the associated CUL3 N-terminal tail. Numbers indicate the order of alpha helices as counted from the ZSWIM8 N terminus. **j**, Superimposition of the ZSWIM8 D-domain with the D-domain of β-TrCP (PDB: 2P64). ZSWIM8 residues 220–270 were used for alignment. The D-domains align with a root-mean-square deviation of 1.1 Å. Remaining ZSWIM8 residues are not shown. **k**, Size-exclusion chromatograms of ZSWIM8–CUL3, ZSWIM8^mono–CUL3, and ZSWIM8^Δdimer–CUL3 complexes (top) and corresponding SDS-PAGE analysis (bottom). ZSWIM8^mono contained a second copy of the D-domain transplanted into its D-domain, creating a constitutively monomeric ZSWIM8 with a folded D-domain. ZSWIM8^Δdimer contained a deletion of its D-domain, which was replaced by a G/S-rich linker. Both ZSWIM8^mono and ZSWIM8^Δdimer eluted at later fractions, indicating lower molecular weight. ZSWIM8^WT and ZSWIM8^mono associated with CUL3, whereas ZSWIM8^Δdimer did not. Note that fraction 14 was not contiguous with the other fractions. **l**, Requirement of ZSWIM8 dimerization for AGO2 ubiquitylation. Shown is a stepwise ubiquitin transfer assay using fluorescently labelled ubiquitin testing the activity of ZSWIM8^mono. AGO2–miR-7 complexes were bound to a target RNA derived from CYRANO, which harboured trigger pairing to miR-7. CYRANO trigger (120 nt) was used as the target RNA (Supplementary Table 1). UBE2L3 was charged with ubiquitin, quenched by addition of apyrase, and mixed with remaining components. ARIH1 - UB* indicates the formation of a thioester-linked complex competent in transferring ubiquitin. Ubiquitylation of ZSWIM8^mono indicates that ZSWIM8^mono is still capable of interacting with CUL3. Ubiquitylation was detected by SDS-PAGE, followed by in-gel fluorescence to visualize fluorescently labelled ubiquitin. Shown is a representative experiment; *n* = 2 technical replicates. **m**, Requirement of ZSWIM8 dimerization for AGO2 ubiquitylation. Shown is an in vitro ubiquitylation assay like that in Fig. 1c, but testing the activity of monomeric ZSWIM8 mutants described in **k**. Shown is a representative experiment; *n* = 2 technical replicates. **n**, Requirement of ZSWIM8 dimerization for TDMD in cells. Shown are results of the intracellular TDMD assay diagrammed and piloted in **d**–**g**, testing mutations affecting ZSWIM8 dimerization, described in **k**. The results are plotted as in **f**. *n* = 2 biological replicates.

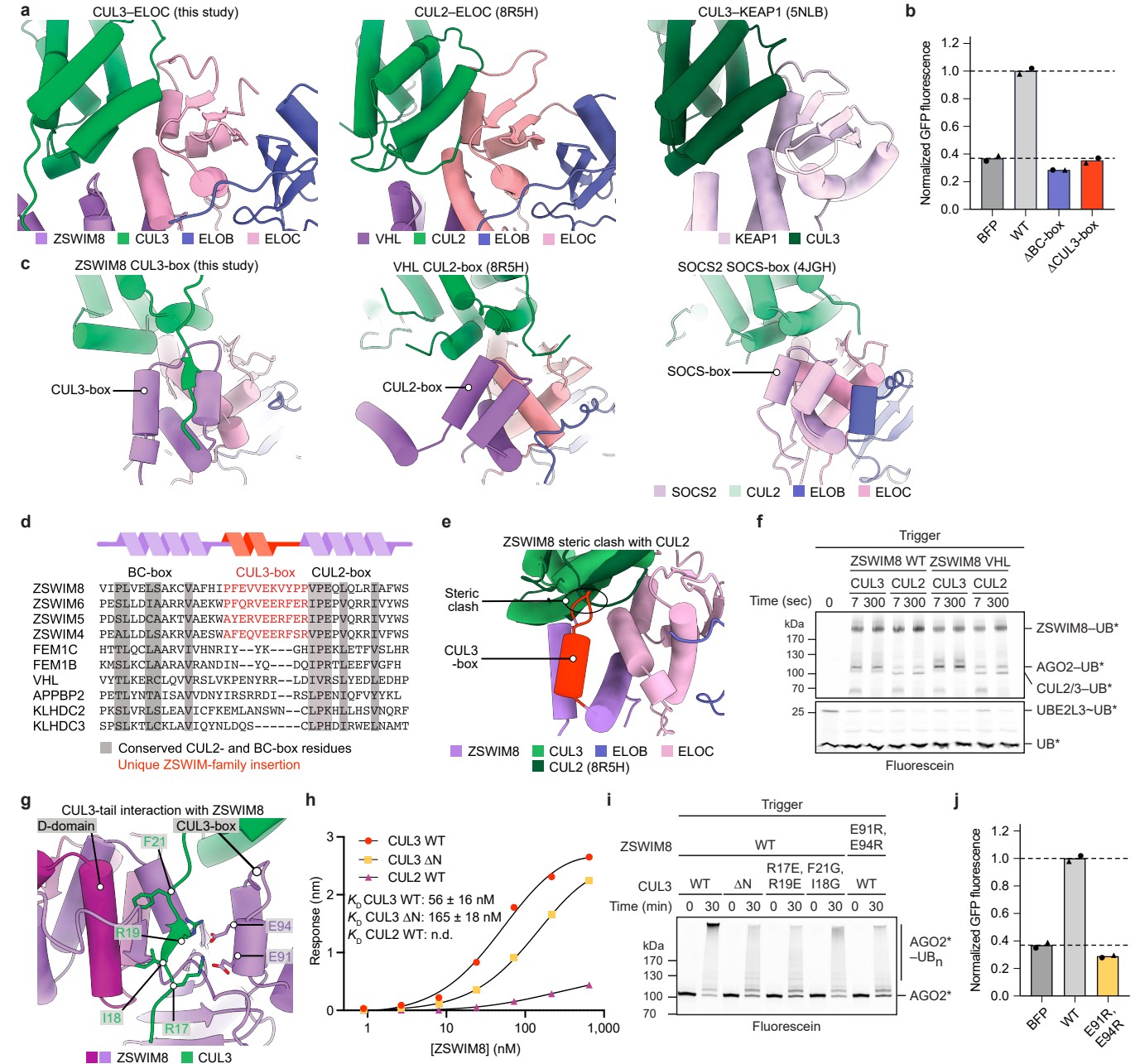

**Extended Data Fig. 4 |** See next page for caption.

**Extended Data Fig. 4 | ZSWIM8–CUL3 defines a distinct class of cullin–RING ligases. a**, Cartoon representations comparing CUL3 binding to ELOC in a ZSWIM8–ELOB/C complex (our structure) (left) with CUL2 binding to ELOC in a VHL–ELOB/C complex (PDB: 8R5H) (middle) and CUL3 binding to KEAP1 (PDB: 5NLB) (right). Complexes were aligned to the first 100 residues of CUL3 in the CUL3–ZSWIM8–AGO2–miR-7–trigger complex. **b**, Importance of ZSWIM8 interactions with ELOB/C and CUL3 for TDMD in cells. Shown is quantification of the intracellular TDMD reporter assay testing mutations in the ZSWIM8 BC-box and CUL3-box, plotted as in Extended Data Fig. 3f. $n$ = 2 biological replicates. **c**, Cartoon representations comparing ZSWIM8 CUL3-box binding to CUL3 (our structure) (left) with VHL CUL2-box binding to CUL2 (PDB: 8R5H) (middle) and SOCS2 SOCS-box binding to CUL5 (PDB: 4JGH) (right). Complexes were aligned to the first 100 residues of CUL3 in the CUL3–ZSWIM8–AGO2–miR-7–trigger complex. **d**, Sequence alignment of ZSWIM8 with other human members of the ZSWIM family and with CUL2 substrate receptors. BC- and CUL2-boxes are shown with conserved regions highlighted. A ZSWIM-family-specific insertion between the BC-box and CUL2-box is shown in red. Hence, we annotate ZSWIM family members as containing a unique CUL3-box. **e**, Overlay of the ZSWIM8–CUL3 interaction with CUL2 (PDB: 8R5H), showing a steric clash between the ZSWIM8 CUL3-box and CUL2. The ZSWIM-family-specific CUL3-box is shown in red. The circle highlights a clash of ZSWIM8 residues and CUL2 N-terminal residues. **f**, Effect of ZSWIM8 cullin-box–cullin pairing on AGO2 ubiquitylation. Shown are results of a stepwise ubiquitin transfer assay using fluorescently labelled ubiquitin testing the activity of a ZSWIM8 variant in which the CUL3-box residues (96–102) were replaced with the VHL CUL2-box residues (177–183) (ZSWIM8 VHL). Otherwise, this panel is as in Extended Data Fig. 3l. This substitution improved the ability of CUL2 to function with ZSWIM8. Shown is a representative experiment; $n$ = 2 technical replicates. **g**, Cartoon representation illustrating the interaction of the CUL3 N-terminal tail with ZSWIM8. The CUL3 N terminus forms an intermolecular amphipathic beta sheet with part of the ZSWIM8 D-domain. In addition, the ZSWIM8 D-domain binds to CUL3 via hydrophobic interactions, and ZSWIM8 residues in the CUL3-box form a negatively charged surface. **h**, ZSWIM8 binding to CUL3 and other cullin variants. Shown is biolayer interferometry data comparing binding of ZSWIM8 to CUL3, CUL3 with a truncation of residues 1–24 (CUL3 ΔN), and CUL2. CUL2 and CUL3 N-terminal domains were biotinylated and immobilized; ZSWIM8 dilutions were used as the analyte. The maximal response is plotted against the concentration of ZSWIM8–ELOB/C complex. Binding affinity was determined by fitting to a single-site binding model. Concentrations of ZSWIM8 were too low to estimate a binding affinity for CUL2. $n$ = 2 technical replicates. Sensorgrams are shown in Supplementary Fig. 9. **i**, Importance of ZSWIM8 interactions with the CUL3 N-terminal tail for AGO2 polyubiquitylation. Shown is an in vitro ubiquitylation assay like that in Fig. 1c but testing the activity of CUL3 variants with mutations in the N-terminal tail (ΔN, truncation of residues 1–24) and a ZSWIM8 variant with mutations in two CUL3-interacting residues. Shown is a representative experiment; $n$ = 2 technical replicates. **j**, Importance of ZSWIM8 interactions with the CUL3 N-terminal tail for TDMD in cells. Shown is quantification of the intracellular TDMD reporter assay testing a ZSWIM8 variant that impairs CUL3 recruitment, plotted as in Extended Data Fig. 3f. $n$ = 2 biological replicates.

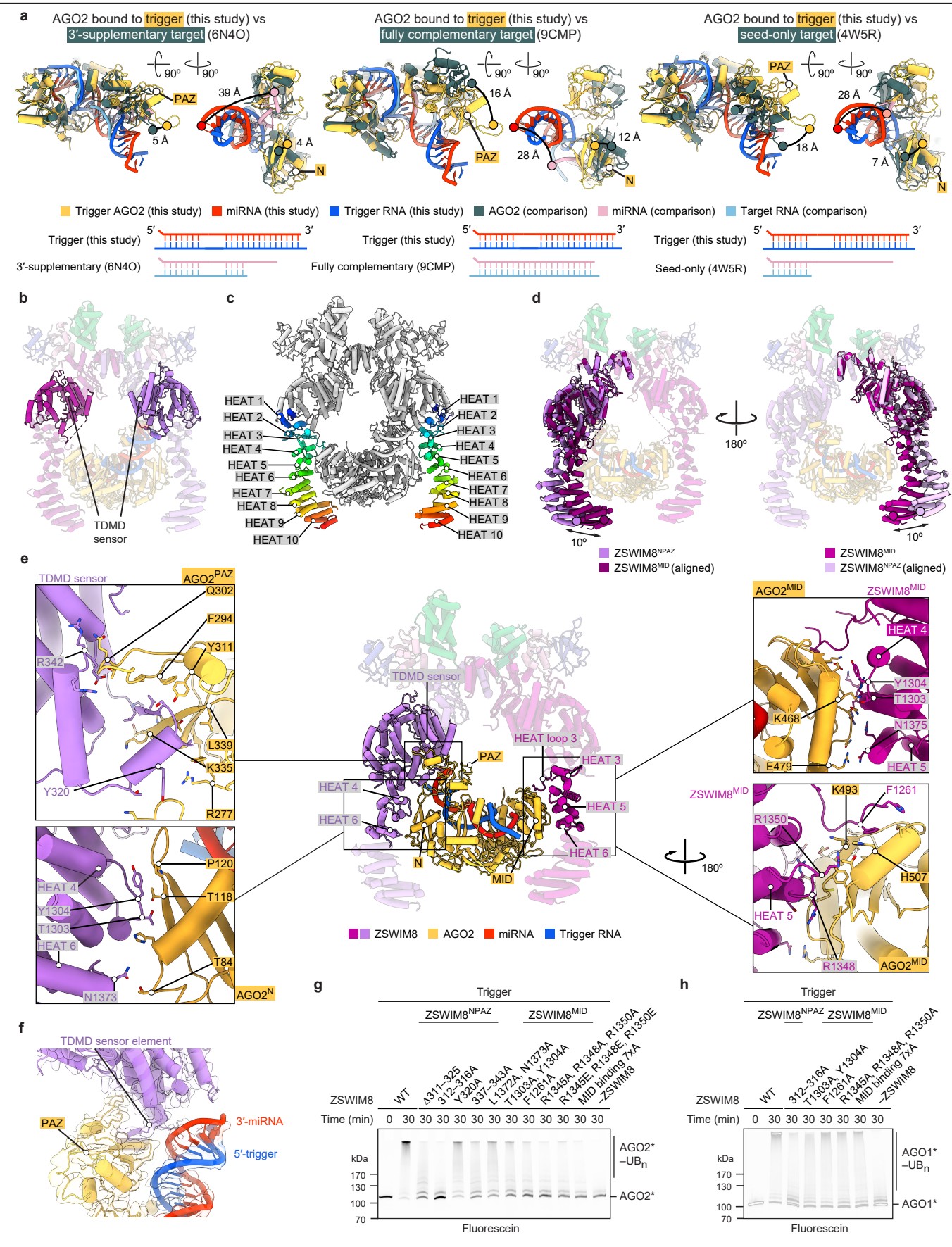

**Extended Data Fig. 5** | See next page for caption.

**Extended Data Fig. 5 | Trigger RNA reshapes the AGO2–miRNA complex into a conformation recognized by ZSWIM8. a**, Cartoon representations comparing the structure of trigger-bound AGO2–miRNA in association with ZSWIM8–CUL3 to previously determined structures of AGO2–miRNA bound to a target RNA with either 3′-supplementary pairing (PDB: 6N4O) (left); fully complementary pairing (PDB: 9CMP) (middle); or pairing to only the seed (PDB: 4W5R) (right). For each comparison, the cartoon representation on the left highlights the movement of the PAZ domain, the cartoon representation on the right highlights the movements of the PAZ and N domains and the miRNA 3′ terminus from a rotated view, and the lower panel diagrams the W–C–F base pairing between the miRNA and the target RNA. **b**, Cartoon representation highlighting the position of the TDMD sensor within the ZSWIM8 dimer. The TDMD sensor spans residues 290–915 but is missing density for a large, apparently unstructured region encompassing residues 500–758. **c**, Cartoon representation highlighting the positions of the HEAT repeats within the ZSWIM8 dimer. HEAT repeats are continuously coloured from N-terminal repeats to C-terminal repeats. **d**, Cartoon representation illustrating structural rearrangements of the two ZSWIM8 protomers. Each ZSWIM8 protomer was aligned with the other protomer using the E3 ligase superdomain and TDMD sensor (residues 1–915). The maximal relative rotation of each protomer was measured at the most C-terminal alpha helix. **e**, Cartoon representation showing interactions of ZSWIM8$^{NPAZ}$ and ZSWIM8$^{MID}$ with the AGO2 PAZ and MID domains, respectively. Middle: Cartoon representation showing an overview of highlighted interactions. Top left: Cartoon representation of ZSWIM8$^{NPAZ}$ interacting with the PAZ domain. The TDMD sensor element forms an intermolecular beta sheet with a portion of the AGO2 PAZ domain (AGO2 residues 335–337). Bottom left: Cartoon representation showing interaction of ZSWIM8$^{NPAZ}$ with the AGO2 N domain. Multiple residues from HEAT repeats 4 and 6 interact with the N domain. Top right: Cartoon representation showing interaction of ZSWIM8$^{MID}$ with the AGO2 MID domain. Residues in HEAT repeats 3, 4, and 5 interact with the MID domain. Bottom right: Same interaction as in top right, rotated by 180°. **f**, Density showing three-way interactions formed by ZSWIM8$^{NPAZ}$, AGO2 PAZ domain, and the miRNA–trigger duplex. Density is derived from the composite map. The underlying cartoon representation is shown. The TDMD sensor element inserts between the PAZ domain and the RNA duplex. **g**, Importance of ZSWIM8 clamp interactions with AGO2. Shown is an in vitro ubiquitylation assay like that in Fig. 1c, except it tested the consequences of mutating ZSWIM8$^{NPAZ}$ and ZSWIM8$^{MID}$ residues that interact with AGO2. Shown is a representative experiment; $n$ = 2 technical replicates. **h**, Evidence that ZSWIM8 clamp interactions important for polyubiquitylation of human AGO2 are also important of polyubiquitylation of human AGO1. Shown is an in vitro ubiquitylation assay like that in panel **g**, but using human AGO1 instead of human AGO2. Shown is a representative experiment; $n$ = 2 technical replicates.

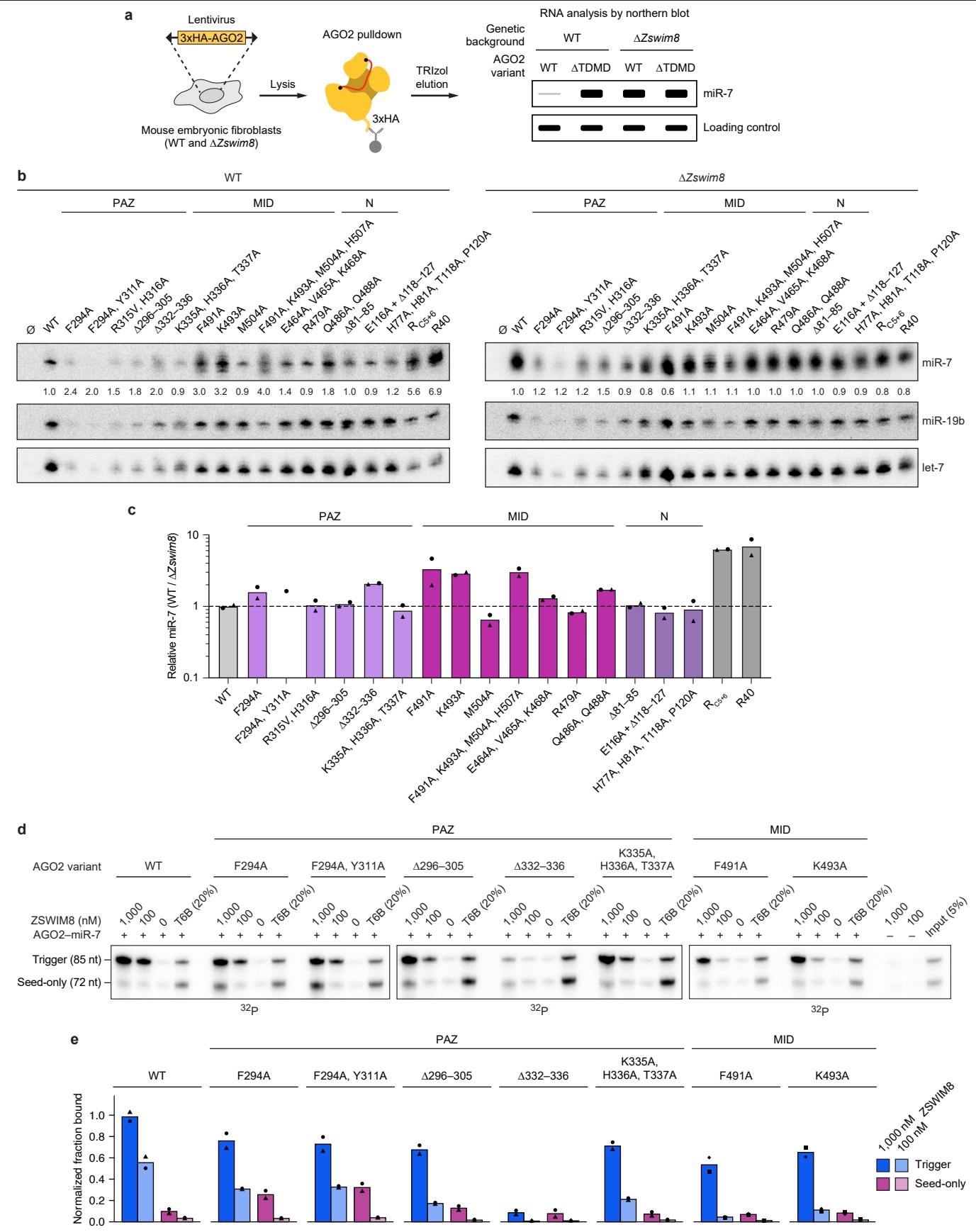

**Extended Data Fig. 6** | See next page for caption.

**Extended Data Fig. 6 | AGO2 residues make important interactions with ZSWIM8. a**, Schematic of the intracellular AGO2 co-IP assay for measuring the levels of miR-7 associated with an AGO2 variant[11]. WT or *Zswim8*-knockout (Δ*Zswim8*) mouse embryonic fibroblasts (MEFs) are transduced with a lentiviral construct expressing an epitope-tagged AGO2 variant of interest (3xHA-AGO2). After selection, cells are harvested and an IP for 3xHA is performed to enrich for RNAs associated with the ectopically expressed AGO2 variant. These co-precipitating RNAs are then analysed by northern blot. The relative amounts of miR-7 associated with each AGO2 variant in WT versus Δ*Zswim8* MEFs reports on the ZSWIM8 sensitivity of the respective AGO2 variant. For example, an AGO2 variant that cannot undergo TDMD (ΔTDMD) is expected to be associated with elevated miR-7 levels in WT cells, and these levels are not expected to be further elevated in Δ*Zswim8* cells, whereas an AGO2 variant that can fully undergo TDMD is expected to resemble WT AGO2, in being associated with a low level of miR-7 in WT cells and an increased level of miR-7 in Δ*Zswim8* cells. **b**, Effect of mutations in AGO2 ZSWIM8-interacting residues on miR-7 levels in cells, assessed as diagrammed in **a**. Shown are representative northern blots measuring levels of miR-7 and control miRNAs (miR-19b, let-7) in IPs from WT and Δ*Zswim8* MEFs expressing the indicated 3xHA-AGO2 variant (Ø, untransduced; $R_{C5+6}$ and R40, previously characterized AGO2 variants with lysine-to-arginine substitutions) (Supplementary Table 2). For each AGO2 variant, the ratio of miR-7:miR-19b and the ratio of miR-7:let-7 was calculated. Shown is the geometric mean of these two ratios normalized to that of WT AGO2. AGO2 levels were detected by western blotting (Supplementary Fig. 11). The samples derive from the same experiment, and blots were processed in parallel. **c**, Quantification of AGO2 co-IP assays. For each AGO2 variant, the geometric mean of the miR-7:miR-19b and miR-7:let-7 ratios in WT MEFs was normalized to that in Δ*Zswim8* MEFs, and this ratio was normalized to the geometric mean of normalized ratios across two WT AGO2 replicates. The bar height represents the geometric mean of two independent measurements observed in two pairs of independently derived WT and Δ*Zswim8* clonal cell lines. *n* = 2 biological replicates. With the F294A, Y311A variant, one of the replicates did not have detectable signal above background, and thus the relative miR-7 levels of this replicate could not be calculated. The quantification for the other replicate is indicated by a black circle. Mutation of the PAZ domain often resulted in decreased levels of control miRNAs, miR-19b and let-7. We considered the possibility that these decreased levels were due to ZSWIM8-dependent turnover of these miRNAs due to a constitutively less occupied PAZ-domain pocket, which might facilitate recognition by ZSWIM8. However, decreased levels of miR-19b and let-7 were also observed in the Δ*Zswim8* background, suggesting that these effects were more likely due to defective formation or stability of the AGO2–miRNA complex, which must have occurred in a mostly ZSWIM8-independent manner. **d**, Importance of AGO2 PAZ- and MID-domain interactions for ZSWIM8 binding. Shown are in vitro co-IP assays as in Fig. 1f, except WT AGO2–miR-7 was replaced with complexes formed using the indicated AGO2 variants. *n* = 2 technical replicates. **e**, Quantification of co-IP assays of panel **d**. Plotted are measurements for radiolabelled target RNA bands associated with each AGO2–miR-7 variant. Band intensities were first background-subtracted using the 0 nM ZSWIM8 sample and normalized to those in the T6B sample. The symbols show data from independent measurements. The bar height represents the mean of these measurements. *n* = 2 technical replicates.

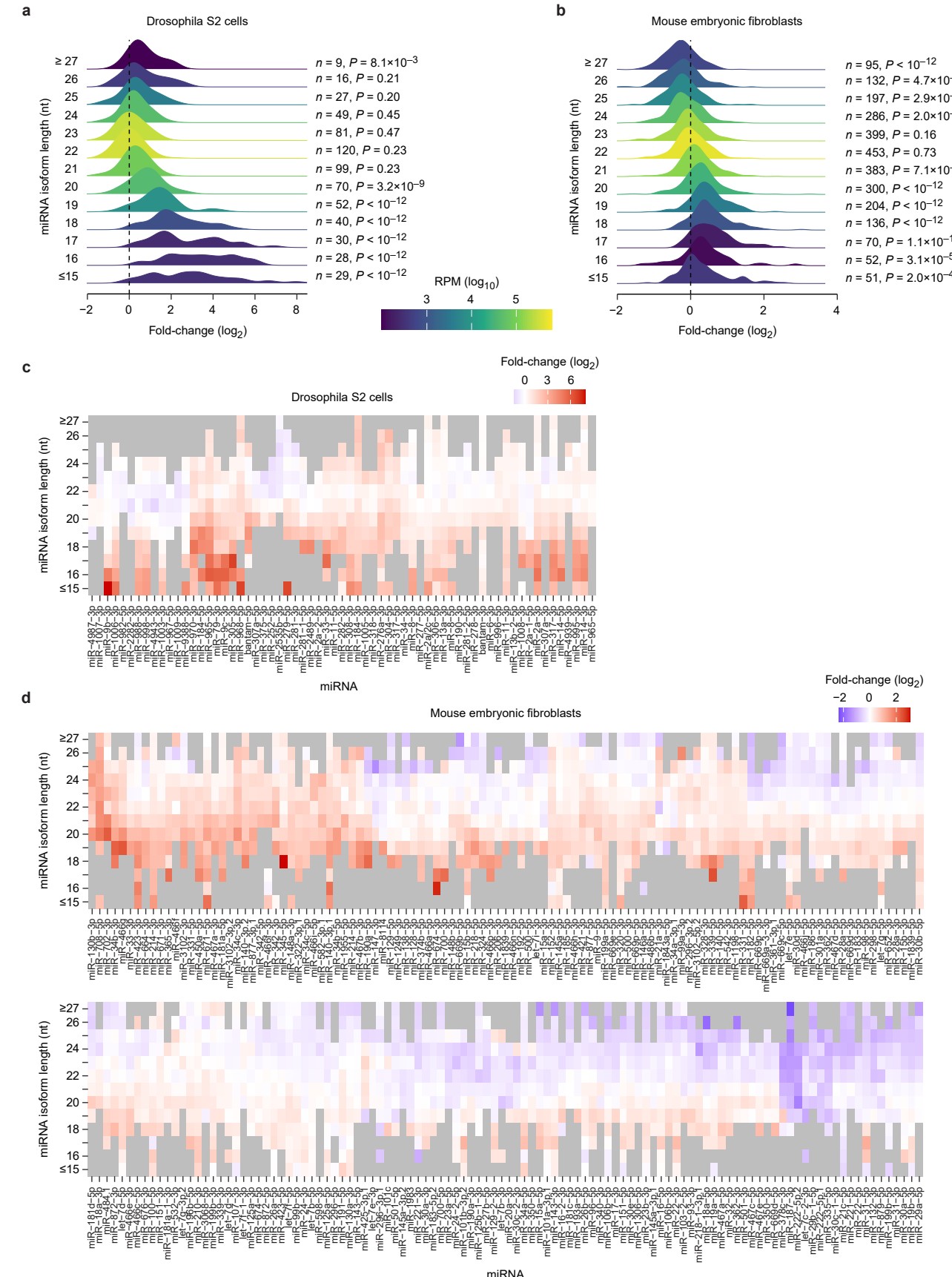

**Extended Data Fig. 7** | See next page for caption.

**Extended Data Fig. 7 | ZSWIM8 destabilizes extensively trimmed miRNAs.**
**a**, Relationship between miRNA isoform length and ZSWIM8 sensitivity in Drosophila cells. The distributions show fold-changes ($\log_2$) in levels of the indicated miRNA length isoforms in *Zswim8*-knockout Drosophila S2 cells, relative to control cells ($n = 3$ biological replicates). This analysis excludes miRNAs with evidence of ZSWIM8 sensitivity of full-length isoforms, for reasons described below. Fold-changes were adjusted to centre the median of the 22-nt distribution at zero, and distributions were coloured by the summed expression in control cells of isoforms composing each bin (key). Isoforms longer than 26 nt or shorter than 16 nt were aggregated into the two terminal bins. For each bin, *n* corresponds to the number of unique miRNAs in that bin. Each distribution of $\log_2$ fold-changes was compared with the distribution of $\log_2$ fold-changes generated by summing the counts for all lengths of a given miRNA using a two-sided Kolmogorov-Smirnov test. The resulting Benjamini-Hochberg-adjusted *P* values and the numbers of different miRNAs contributing to each distribution are listed. The greater fold-changes observed for shorter isoforms support the idea that shorter isoforms are broadly susceptible to ZSWIM8-mediated degradation. One consideration potentially confounding this interpretation is that some TDMD sites also promote the removal or the untemplated addition of nucleotides to the 3′ end of miRNAs, which is presumably due to liberation of the miRNA 3′ end from AGO upon extensive 3′ pairing to a trigger RNA. In the absence of ZSWIM8 or its orthologues, these trimmed or tailed isoforms can accumulate. To maintain focus on a potential effect of miRNA length on ZSWIM8 sensitivity—rather than changes in miRNA length downstream of trigger binding—we sought to exclude from the analysis miRNAs with evidence for classical ZSWIM8 sensitivity. To this end, we summed the reads associated with all lengths of a given miRNA, calculated fold-changes in *Zswim8*-knockout versus control cells for each miRNA, and excluded all putative ZSWIM8-sensitive miRNAs (those with $\log_2$ fold-change > 0 and $P_{adj} < 0.05$, those having undergone an increase significantly larger than that of their passenger strand, and those previously annotated as ZSWIM8-sensitive). ZSWIM8-dependent fold-changes for all isoforms of the remaining, insensitive miRNAs were calculated. Analogous plots for miRNAs whose levels are significantly or potentially affected by ZSWIM8 are shown in Supplementary Fig. 15. **b**, Relationship between miRNA isoform length and ZSWIM8 sensitivity in murine cells. The distributions show fold-changes ($\log_2$) in miRNA levels in *Zswim8*-knockout mouse embryonic fibroblasts (MEFs), relative to control cells ($n = 3$ biological replicates). This panel is as in **a**, except analysis is of data from MEFs. **c**, Heatmap of fold-changes ($\log_2$) of miRNA molecules of the indicated lengths in *Zswim8*-knockout versus control S2 cells for all miRNAs shown in **a** that have at least four isoforms of different lengths, ordered by similarity based on Euclidean distance. For miRNAs with multiple isoforms represented in the terminal bins (for instance, miRNAs with 13-nt, 14-nt, and 15-nt isoforms aggregated into the ≤15-nt bin), the mean of the constituent $\log_2$ fold-changes was used as the $\log_2$ fold-change for the terminal bin. **d**, Heatmap of fold-changes ($\log_2$) of miRNA molecules of the indicated lengths in *Zswim8*-knockout versus control MEFs for all miRNAs shown in **b** that have at least six isoforms of different lengths, plotted as described in **c**. In both cell types, shorter isoforms (<22 nt) accumulated upon loss of ZSWIM8 to a greater extent than the dominant miRNA isoforms, which were typically 22 nt. In Drosophila S2 cells and MEFs, <19-nt guides and 17–19-nt guides, respectively, underwent the highest mean ZSWIM8-associated degradation. Of note, in vitro measurements of miRNA 3′ end accessibility to nucleases indicate that miRNAs typically require 18–19 nt to be stably bound by the PAZ domain of human AGO2 and AGO3, depending on the primary sequence of the 3′-terminal nucleotides[56]. Thus, our data are consistent with the idea that short (<20 nt) guides whose 3′-terminal nucleotides are unable or less likely to occupy the PAZ binding pocket in AGO are recognized by ZSWIM8 through an interaction between ZSWIM8 and PAZ, resulting in miRNA degradation. However, despite stringent removal of putative ZSWIM8 substrates from the analyses, we cannot fully rule out the possibility that the preferential accumulation of short guides in *Zswim8* knockouts results from trimming of weakly ZSWIM8-sensitive miRNAs that have extensive 3′ pairing to unknown trigger RNAs. Similarly, if the extent of trimming or tailing increases over the lifetime of a miRNA, miRNAs even mildly destabilized by ZSWIM8 could accumulate more trimmed and tailed isoforms in the absence of ZSWIM8.

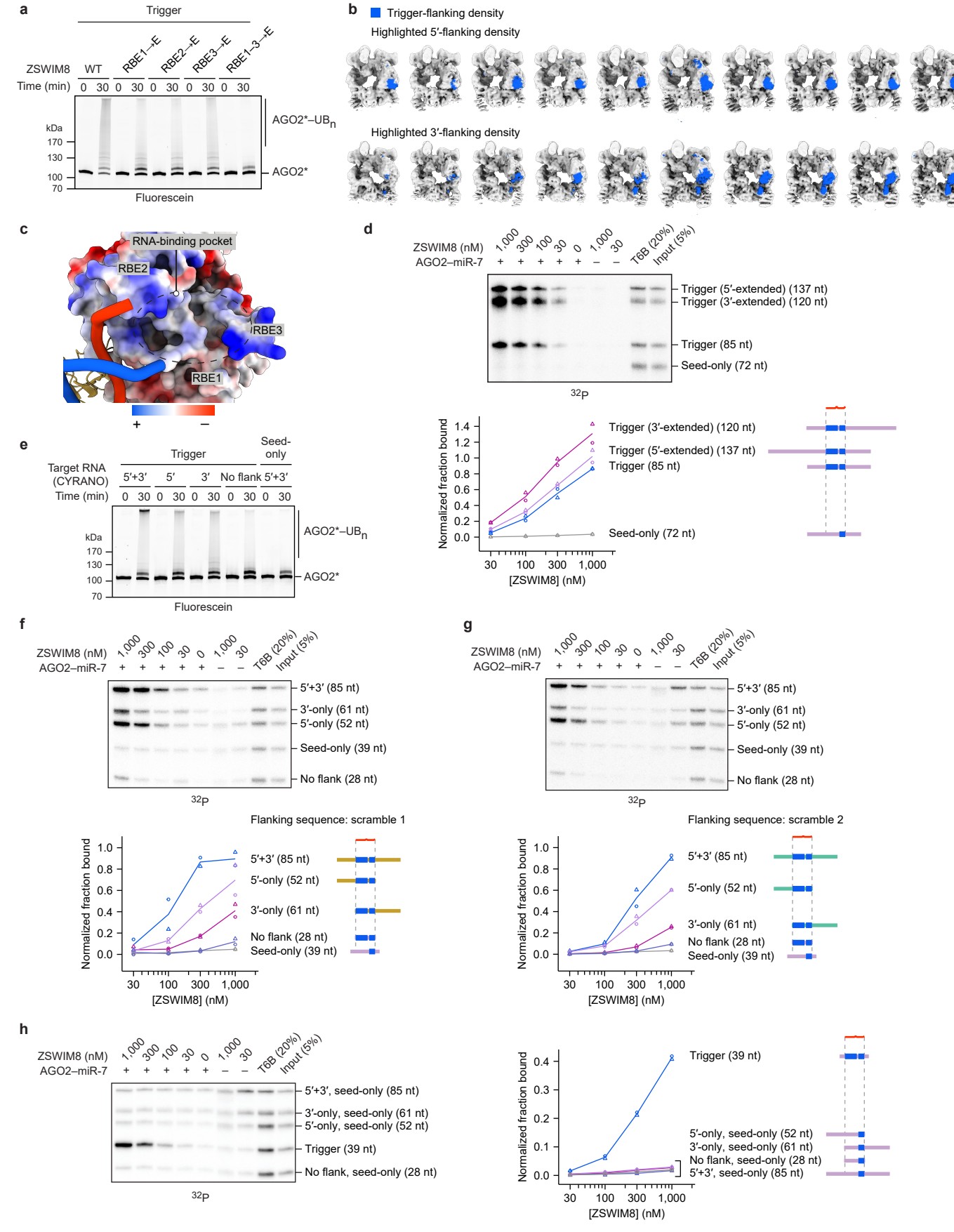

**Extended Data Fig. 8** | See next page for caption.

**Extended Data Fig. 8 | Flanking trigger RNA interacts with ZSWIM8.**
**a**, The importance of RBEs for AGO2 polyubiquitylation. Shown are in vitro ubiquitylation assays like that in Fig. 1c, except ZSWIM8 was mutated at the indicated RBE or at all three RBEs. Residues in RBEs were mutated to glutamates. Shown is a representative experiment; $n = 2$ technical replicates. **b**, Cryo-EM densities obtained by 3D classification showing different extents of additional density assigned as either 5′ or 3′ flanking trigger RNA (top and bottom, respectively). Densities assigned as RNA are shown in blue. The density assigned as 3′ flanking RNA showed considerable variability in its volume and position, indicating structural heterogeneity. **c**, Charge-driven interactions of ZSWIM8 RBEs with RNA. Shown is a surface rendering of ZSWIM8[NPAZ] showing the calculated Coulombic potential of RBEs 1–3 positioned to interact with the miRNA–trigger duplex and flanking trigger RNA. Coulombic potential is shown in the range of –10–10 kcal/(mol·$e$). **d**, Testing the effect of further extending the trigger flanking regions on ZSWIM8 binding to AGO2–miR-7–CYRANO. This panel is as in Fig. 1f, except the 85-nt trigger RNA was extended in either direction by additional CYRANO sequence. In addition, a higher concentration of heparin (10 µg/mL) was included. $n = 2$ technical replicates. **e**, Contributions of 5′ and 3′ trigger flanking sequences to AGO2 polyubiquitylation. Shown are in vitro ubiquitylation assays like that in Fig. 1c, except the target RNA had CYRANO sequence flanking the trigger site at either both ends, one end, or neither end. CYRANO trigger 5′ + 3′ (85 nt), 5′-only (52 nt), 3′-only (61 nt), no flank (28 nt), and seed-only (39 nt) target RNAs were used (Supplementary Table 1). Shown is a representative experiment; $n = 2$ technical replicates. **f**, Testing sequence specificity of the contributions of trigger flanking sequences to ZSWIM8 binding to AGO2–miR-7–CYRANO. This panel is as in Fig. 4e, except the flanking regions derived from CYRANO were replaced with scrambled sequences that maintained the nucleotide composition of the original CYRANO sequence. In addition, heparin was not included in the reaction. $n = 2$ technical replicates. **g**, Testing sequence specificity of the contributions of trigger flanking sequences to ZSWIM8 binding to AGO2–miR-7–CYRANO. This panel is as in **f** but with different scrambled sequences (scramble 2). $n = 2$ technical replicates. **h**, Testing the contributions of trigger flanking sequences to ZSWIM8 binding to AGO2-miR-7–CYRANO[Seed-only]. This panel is as in Fig. 4e, except most target RNAs contained seed-only pairing instead of trigger pairing, and heparin was not included in the reaction. $n = 2$ technical replicates.

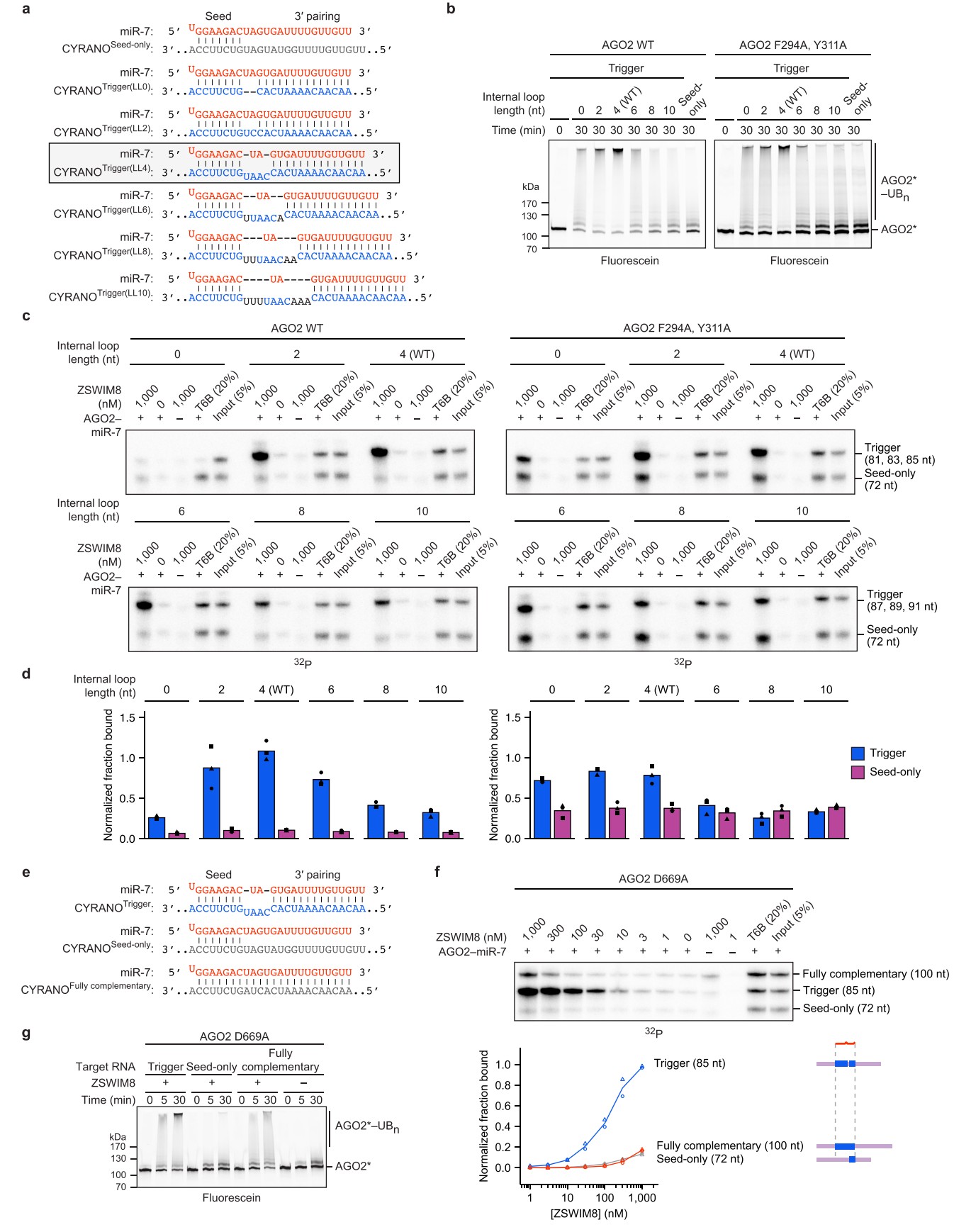

**Extended Data Fig. 9** | See next page for caption.

**Extended Data Fig. 9 | ZSWIM8 recognizes trigger-specific RNA trajectory.**
**a**, Diagrams showing pairing between miR-7 and the miRNA-binding regions of CYRANO$^{Trigger}$ with varying lengths of the internal loop (LL, loop length). Vertical lines represent W–C–F base pairing. CYRANO$^{Trigger(LL4)}$ corresponds to the WT internal loop length and is highlighted by a grey box. **b**, Effect of varying trigger internal loop length on AGO2 polyubiquitylation. Shown is an in vitro ubiquitylation assay like that in Fig. 1c, except it used either WT AGO2 (left panel) or a variant with mutations in the PAZ-domain pocket (F294A, Y311A) (right panel), and it used target RNAs with varying internal loops lengths (panel **a**) (Supplementary Table 1). Shown is a representative experiment; $n = 2$ technical replicates. **c**, Effect of varying trigger internal loop length on ZSWIM8 binding to AGO2–miR-7 complexes. Shown is an in vitro co-IP assay like that in Fig. 1f, except it used either WT AGO2 (left panel) or a variant with mutations in the PAZ-domain pocket (F294A, Y311A) (right panel), and it used target RNAs with varying internal loop lengths (panel **a**) (Supplementary Table 1). **d**, Quantification of co-IP assays of panel **c**. Plotted are measurements for radiolabelled target RNA bands associated with each AGO2–miR-7–target complex (left panel, WT AGO2; right panel, AGO2 F294A, Y311A). Band intensities were first background-subtracted using the 0 nM ZSWIM8 sample and normalized to those in the T6B sample. The symbols show data from independent measurements. The bar height represents the mean of these measurements. $n = 3$ technical replicates. **e**, Diagrams showing pairing between miR-7 and the miRNA-binding regions of CYRANO$^{Trigger}$, CYRANO$^{Seed-only}$, and CYRANO$^{Fully\ complementary}$ target RNAs used for in vitro ubiquitylation and co-IP assays. Vertical lines represent W–C–F base pairing. **f**, Effect of fully complementary target pairing on ZSWIM8 binding to AGO2–miR-7 complexes. Shown is an in vitro co-IP assay like that in Fig. 1f, except it used an active-site AGO2 variant (D669A) and included a 100-nt radiolabelled target RNA with full complementarity to miR-7 (panel **e**) (Supplementary Table 1). The symbols show data from independent measurements. $n = 2$ technical replicates. **g**, Effect of fully complementary target pairing on AGO2 polyubiquitylation. Shown is an in vitro ubiquitylation assay like that in Fig. 1c, but also testing the effects of fully complementary pairing to miR-7 (panel **e**) using an active-site AGO2 variant (D669A). CYRANO trigger (120 nt), seed-only (135 nt), and fully complementary (100 nt) target RNAs were used (Supplementary Table 1). Shown is a representative experiment; $n = 2$ technical replicates.

**Extended Data Table 1 | Cryo-EM data collection, refinement and validation statistics**

| | Map A ZSWIM8-CUL3 complex bound to AGO2-miR-7-CYRANO, (EMDB-54348) | Map B Locally refined interactions of ZSWIM8-CUL3 complex bound to AGO2-miR-7-CYRANO (EMDB-54349) | Map C focused map of ZSWIM8-CUL3 complex bound to AGO2-miR-7-CYRANO (EMDB-54351) | Map D Locally refined map of ZSWIM8-CUL3 complex bound to AGO2-miR-7-CYRANO (EMDB-54350) | Map E Composite map of ZSWIM8-CUL3 complex bound to AGO2-miR-7-CYRANO (EMDB-54352) (PDB-9RWZ) |
|---|---|---|---|---|---|
| **Data collection and processing** | | | | | |
| Magnification | 105,000 | 105,000 | 105,000 | 105,000 | 105,000 |
| Voltage (kV) | 300 | 300 | 300 | 300 | 300 |
| Electron exposure (e–/Å$^2$) | 58 | 58 | 58 | 58 | 58 |
| Defocus range (μm) | -0.5 – -2.0 | -0.5 – -2.0 | -0.5 – -2.0 | -0.5 – -2.0 | -0.5 – -2.0 |
| Pixel size (Å) | 0.8512 | 0.8512 | 0.8512 | 0.8512 | 0.8512 |
| Symmetry imposed | C1 | C1 | C1 | C1 | C1 |
| Initial particle images (no.) | 5,812,715 | 5,812,715 | 5,812,715 | 5,812,715 | 5,812,715 |
| Final particle images (no.) | 234,181 | 234,181 | 234,181 | 234,181 | 234,181 |
| Map resolution (Å) FSC threshold | 3.1 | 3.1 | 3.2 | 3.2 | 3.1 |
| Map resolution range (Å) | 2.8-3.3 | 2.7-3.2 | 2.8-3.3 | 2.8-3.4 | 2.8-3.3 |
| | | | | | |
| **Refinement** | | | | | |
| Initial model used (PDB code) | | | | | Alphafold3, AGO2 (6NIT), CUL3 (5NLB), ELOB/C (1LM8) |
| Model resolution (Å) FSC threshold | | | | | 3.1 |
| Model resolution range (Å) | | | | | |
| Model composition | | | | | |
| Non-hydrogen atoms | | | | | 28,597 |
| Protein residues | | | | | 3,609 |
| Nucleotides | | | | | 50 |
| Ligands | | | | | 2 Zn |
| *B* factors (Å$^2$) | | | | | |
| Protein | | | | | 87.42 |
| Nucleotides | | | | | 103.12 |
| Ligand | | | | | 97.48 |
| R.m.s. deviations | | | | | |
| Bond lengths (Å) | | | | | 0.007 |
| Bond angles (°) | | | | | 0.603 |
| Validation | | | | | |
| MolProbity score | | | | | 2.02 |
| Clashscore | | | | | 5.75 |
| Poor rotamers (%) | | | | | 2.10 |
| Ramachandran plot | | | | | |
| Favored (%) | | | | | 92.77 |
| Allowed (%) | | | | | 7.14 |
| Disallowed (%) | | | | | 0.08 |

| | |
|---|---|

# Reporting Summary

## Statistics

For all statistical analyses, confirm that the following items are present in the figure legend, table legend, main text, or Methods section.

| n/a | Confirmed | |
|---|---|---|
| ☐ | ☒ | The exact sample size (*n*) for each experimental group/condition, given as a discrete number and unit of measurement |
| ☐ | ☒ | A statement on whether measurements were taken from distinct samples or whether the same sample was measured repeatedly |
| ☐ | ☒ | The statistical test(s) used AND whether they are one- or two-sided<br>*Only common tests should be described solely by name; describe more complex techniques in the Methods section.* |
| ☒ | ☐ | A description of all covariates tested |
| ☐ | ☒ | A description of any assumptions or corrections, such as tests of normality and adjustment for multiple comparisons |
| ☐ | ☒ | A full description of the statistical parameters including central tendency (e.g. means) or other basic estimates (e.g. regression coefficient) AND variation (e.g. standard deviation) or associated estimates of uncertainty (e.g. confidence intervals) |
| ☐ | ☒ | For null hypothesis testing, the test statistic (e.g. *F*, *t*, *r*) with confidence intervals, effect sizes, degrees of freedom and *P* value noted<br>*Give P values as exact values whenever suitable.* |
| ☒ | ☐ | For Bayesian analysis, information on the choice of priors and Markov chain Monte Carlo settings |
| ☒ | ☐ | For hierarchical and complex designs, identification of the appropriate level for tests and full reporting of outcomes |
| ☒ | ☐ | Estimates of effect sizes (e.g. Cohen's *d*, Pearson's *r*), indicating how they were calculated |

*Our web collection on statistics for biologists contains articles on many of the points above.*

## Software and code

Policy information about availability of computer code

| Data collection | Cryo-EM: SerialEM (v4.1); Flow cytometry: BD FACSDiva (v9.0); Gel and blot imaging: Amersham Typhoon, Typhoon 9410, and LI-COR Odyssey CLx; sRNA-seq: Illumina NovaSeq 6000; Bio-layer interferometry: Octet Data Acquisition HT (v13.0.1) |
|---|---|
| Data analysis | Cryo-EM structure analysis: CryoSparc (v6.4.2), COOT (0.9.8.95), Phenix (1.21.1), AlphaFold3, and ChimeraX (v1.8–1.9); Flow cytometry analysis: FlowJo (v10.10.0); Gel band quantification: ImageQuant TL (v10.2) and LI-COR ImageStudio (v6.1.0.79); Bio-layer interferometry analysis: Octet Data Analysis HT (v13.0.1); All statistical analyses: GraphPad Prism (v10.4.0); sRNA-seq analysis: cutadapt (v4.8), FASTX Toolkit (v0.0.14), and DESeq2 (v1.38.3). Original code for the analysis of sRNA-seq data is available publicly at https://github.com/lwblodgett/ZSWIM8_sensitivity_of_miRNA_isoforms.git (copy archived at Zenodo: https://doi.org/10.5281/zenodo.18265217). |

For manuscripts utilizing custom algorithms or software that are central to the research but not yet described in published literature, software must be made available to editors and reviewers. We strongly encourage code deposition in a community repository (e.g. GitHub). See the Nature Portfolio guidelines for submitting code & software for further information.

## Data

Policy information about availability of data

All manuscripts must include a data availability statement. This statement should provide the following information, where applicable:

- Accession codes, unique identifiers, or web links for publicly available datasets
- A description of any restrictions on data availability
- For clinical datasets or third party data, please ensure that the statement adheres to our policy

The structural data will be made publicly available from the PDB and EMDB upon manuscript publication. The atomic coordinates have been deposited in the PDB with accession code 9RWZ, and electron microscopy maps deposited with the Electron Microscopy Data Bank with codes EMD-54348, EMD-54349, EMD-54350, EMD-54351, and EMD-54352. Sequencing data has been deposited in the Gene Expression Omnibus with accession code GSE303177 and will be made publicly available upon manuscript publication. Uncropped in-gel fluorescence images, the workflow for cryo-EM structure generation, and gating strategies for flow cytometry experiments are provided in the Supplementary Figures.

## Research involving human participants, their data, or biological material

Policy information about studies with human participants or human data. See also policy information about sex, gender (identity/presentation), and sexual orientation and race, ethnicity and racism.

| | |
|---|---|
| Reporting on sex and gender | This study does not involve human participants, their data, or their biological material. |
| Reporting on race, ethnicity, or other socially relevant groupings | This study does not involve human participants, their data, or their biological material. |
| Population characteristics | This study does not involve human participants, their data, or their biological material. |
| Recruitment | This study does not involve human participants, their data, or their biological material. |
| Ethics oversight | This study does not involve human participants, their data, or their biological material. |

Note that full information on the approval of the study protocol must also be provided in the manuscript.

# Field-specific reporting

Please select the one below that is the best fit for your research. If you are not sure, read the appropriate sections before making your selection.

☒ Life sciences  ☐ Behavioural & social sciences  ☐ Ecological, evolutionary & environmental sciences

For a reference copy of the document with all sections, see nature.com/documents/nr-reporting-summary-flat.pdf

# Life sciences study design

All studies must disclose on these points even when the disclosure is negative.

| | |
|---|---|
| Sample size | No statistical methods were used to predetermine sample size. Sample sizes were chosen based on pilot experiments to ensure clear and reliable interpretation of the results. The sample sizes used in this study are consistent with standard practices in the field (PMID: 32661162, 33536622). The exact sample size is indicated in each figure legend. |
| Data exclusions | No data were excluded. |
| Replication | Experiments were replicated at least twice with similar results. Representative images are shown where appropriate. |
| Randomization | Not applicable; there was no subjective rating of data involved in our study. |
| Blinding | Not applicable; there was no subjective rating of data involved in our study. |

# Reporting for specific materials, systems and methods

We require information from authors about some types of materials, experimental systems and methods used in many studies. Here, indicate whether each material, system or method listed is relevant to your study. If you are not sure if a list item applies to your research, read the appropriate section before selecting a response.

## Materials & experimental systems

| n/a | Involved in the study |
|-----|------------------------|
| ☐ | ☒ Antibodies |
| ☐ | ☒ Eukaryotic cell lines |
| ☒ | ☐ Palaeontology and archaeology |
| ☒ | ☐ Animals and other organisms |
| ☒ | ☐ Clinical data |
| ☒ | ☐ Dual use research of concern |
| ☒ | ☐ Plants |

## Methods

| n/a | Involved in the study |
|-----|------------------------|
| ☒ | ☐ ChIP-seq |
| ☐ | ☒ Flow cytometry |
| ☒ | ☐ MRI-based neuroimaging |

# Antibodies

| | |
|---|---|
| Antibodies used | Primary antibodies: rabbit anti-ZSWIM8 (1:400; Invitrogen, PA5-59492), rabbit anti-HA (1:5,000; Cell Signaling Technology, C29F4, 3724), mouse anti-GAPDH (1:2,000; Invitrogen, GA1R, MA5-15738). Secondary antibodies: IRDye 680RD goat anti-rabbit (1:10,000; LI-COR, 926-68071), IRDye 800CW goat anti-mouse (1:10,000, LI-COR, 926-32210). |
| Validation | anti-ZSWIM8: https://www.thermofisher.com/antibody/product/ZSWIM8-Antibody-Polyclonal/PA5-59492 anti-HA: https://www.cellsignal.com/products/primary-antibodies/ha-tag-c29f4-rabbit-mab/3724?srsltid=AfmBOopj1hBE5_sImYHbedpnQ-KkBtzFCpzgMNo4rj3eqg7Z7JtCB9YW anti-GAPDH: https://www.thermofisher.com/antibody/product/GAPDH-Loading-Control-Antibody-clone-GA1R-Monoclonal/MA5-15738 |

# Eukaryotic cell lines

Policy information about cell lines and Sex and Gender in Research

| | |
|---|---|
| Cell line source(s) | Sf9 cells were were obtained from Thermo Fisher Scientific (11496015). High Five cells (BTI-TN-5B1-4) were obtained from Thermo Fisher Scientific (B85502). Expi293F cells were obtained from Thermo Fisher Scientific (A14527). K562 cells harboring a miR-7-sensitive GFP reporter were a gift from Joshua Mendell, and were originally obtained from American Type Culture Collection (ATCC). MEF, S2, and HEK293FT cells are Bartel lab stocks. Generation of specific cell lines was performed as described in the methods. |
| Authentication | Cell lines were not authenticated. |
| Mycoplasma contamination | Cell lines tested negative for mycoplasma contamination upon arrival to the lab. |
| Commonly misidentified lines (See ICLAC register) | No commonly misidentified cell lines were used in this study. |

# Plants

| | |
|---|---|
| Seed stocks | No plant material was used in this study. |
| Novel plant genotypes | No plant material was used in this study. |
| Authentication | No plant material was used in this study. |

# Flow Cytometry

## Plots

Confirm that:

☒ The axis labels state the marker and fluorochrome used (e.g. CD4-FITC).

☒ The axis scales are clearly visible. Include numbers along axes only for bottom left plot of group (a 'group' is an analysis of identical markers).

☒ All plots are contour plots with outliers or pseudocolor plots.

☒ A numerical value for number of cells or percentage (with statistics) is provided.

## Methodology

| | |
|---|---|
| Sample preparation | Samples were prepared as described in the methods. Upon harvesting, cells were concentrated to 2 million cells/mL in media and subjected to flow cytometry. |
| Instrument | BD LSR Fortessa |
| Software | BD FACSDiva (v9.0); FlowJo (v10.10.0) |
| Cell population abundance | For each sample, 20,000 live cells were analyzed for GFP fluorescence in order to obtain a sufficient representation of each sample population. |
| Gating strategy | Initial gating steps included identification of live cells (SSC-A/FSC-A) followed by identification of single cells (FSC-H/FSC-A). This population of cells was analyzed by plotting the GFP fluorescence as a histogram. The gating strategy is shown in Supplementary Figure 7. |

☒ Tick this box to confirm that a figure exemplifying the gating strategy is provided in the Supplementary Information.

