## [Peer Review File · Nature]

The E3 ubiquitin ligase mechanism specifying targeted microRNA degradation

Corresponding Author: Dr David Bartel

Version 0:

Reviewer comments:

Referee #1

(Remarks to the Author)

This is a beautiful study on the mechanism of target directed microRNA degradation (TDMD). In this process Argonaut binds miRNA and a trigger RNA and the combination of those three is recognized by an adaptor protein ZSWIM8. This adaptor protein would be expected to collaborate with CUL2/CUL5 CRLs, but in this instance it works with CUL3. A cryo-EM study is presented that explains the 4-way specificity of the complex as well as the need for CUL3, all backed up by mutant analysis and biochemical and cellular assays. All in all, this is very high-class work, really illuminating an unusual regulatory pathway, helping the field forward.

There are only a few minor comments

- The introduction talks a lot about mouse and Drosophila but fails to mention human. As the main text is also not clear on this, one needs to find the materials and methods to realize the work is on human proteins. It would be helpful to mention this in a few places.
- The cryo-EM structure is introduced too briefly to be understood, it would be good to use a few sentences to explain how 'The structure' was obtained (combination of focused maps, with mobility analysis, to be discussed later) and which aspects will be discussed later e.g. the 'flanking' RNA regions are mentioned but only explained pages later
- An interesting question, triggered in part by the discussion, is whether the rearranged interaction changes k_{cat} or K_M of the E3 ligase. It seems the authors favor binding as the main effect, but it would be useful to point out in discussion that effects on catalysis are also possible
- Fig 1A: it is not at all clear how long Cul3 is, what part is in the experiment and what part is visible; please replace the schematics in Figure 1 to reflect the full-length proteins, and indicate what was included in the experiment and what was visible in some sort of color/pattern scheme. Also indicate precise boundaries for the constructs here or in materials and methods.
- Extended Fig 3I: what is the extra band at 70 kD?
- Extended Fig 4D: the red region is called CUL3 box in the legend, why not add this to the figure; it would also be helpful to color this region red in Fig 4E
- Extended Fig 7D: Whereas one of the double mutants is affected, there does not seem to be a detrimental effect of the R315V,H316A double mutant, whereas the main text claims they are both detrimental. This needs to be clarified better in the text.

Referee #2

(Remarks to the Author)

In the submitted manuscript, Farnung and colleagues describe the structural basis for target-directed microRNA degradation (TDMD). They first reconstitute the process of TDMD in vitro, demonstrating that a model TDMD-triggering RNA (Cyrano) can specifically direct ubiquitylation of Argonaute proteins by an E3 ligase of which ZSWIM8 is the substrate recognition module. The TDMD trigger requires extensive complementarity to the miRNA to elicit ubiquitylation since a seed-only binding site does not do so. The authors confirm the unusual composition of the ZSWIM8 E3 complex that was previously

implied by genetic experiments (coupling Elongin B/C with CUL3 and ARIH1). The authors establish an assay that tests recruitment of ZSWIM8 by various AGO-miRNA-target/trigger ternary complexes, demonstrating that the complex containing the Cyrano trigger binding site more efficiently recruits ZSWIM8 than a seed-only complement. Building upon this, the authors resolve the structure of a full "TDMD complex" in which the AGO2-miRNA-trigger complex is bound by a clamp of two ZSWIM8 protomers, each itself bound to one unit of ELOB, ELOC, and CUL3. The structure provides many surprising insights into the mode of interaction of the AGO2-miRNA-trigger complex with the ubiquitin ligase. First, the dimerization of ZSWIM8 is unusual, forming an irreversible intermolecular knot between the monomers; nearby domains also form the binding platforms for CUL3 and ELOB/C. Second, the ZSWIM8 protomers each interact with opposite surfaces of the AGO, one along the MID domain and the other along the PAZ/N domains. Third, specific contacts sense an "empty" PAZ domain that results from release the miRNA 3' terminus. Fourth, ZSWIM8 forms nonspecific interactions with the regions of the trigger RNA extending from AGO.

The structure is very impressive and answers many questions regarding the mechanism of TDMD. The biochemical assay measuring recruitment of ZSWIM8 by various RNA substrates is very informative, and the use T6B as a control for differences in target affinities is elegant. The experiments testing the requirement for dimerization of ZSWIM8 are also particularly cleverly designed and easy to interpret.

Some of the experiments testing importance of specific interfaces would be more impactful with additional assays performed as suggested below. Some of the writing tends to overstate the certainty of the role of RNA trajectory.

- The assay for testing the function of AGO2 residues in Extended Data Figure 7b seems more difficult to interpret than other assays in the paper. Introducing the variants into cells expressing endogenous AGOs and normalizing to other IPed miRNAs may have caveats. For instance, if a constitutively empty PAZ pocket reduces ZSWIM8's ability to discriminate between a trigger and a seed-only complex, as suggested by Figure 3F, then non-TDMD substrates like miR-19b and let-7 could become TDMD substrates when loaded in PAZ mutant AGO, confounding interpretation of the assay as it's currently normalized. (Assuming that non-PAZ AGO mutants would not affect ZSWIM8's discrimination between trigger and seed-only could also be worrisome and slightly circular.) The in vitro co-IP assay (Extended Data 7d) does not carry the same caveats, so it would be helpful if additional AGO mutants could be tested by this assay: particularly the mutants expected to disrupt interaction with ZSWIM8 F1261 (AGO2 F491A and K493A).

- On page 8, "With the goal of impairing miRNA 3' end binding to the PAZ pocket, PAZ residues that interact with the terminal miRNA nucleotide at either the nucleobase (F294A, Y311A) or its backbone phosphate (R315V, H316A) were mutated 14,72 (Figure 3e). These substitutions caused increased binding to the seed-only-bound AGO2-miR-7 (Extended Data Figure 7d,e), consistent with an unoccupied PAZ pocket comprising a feature selectively recognized by ZSWIM8." R315V, H316A does not appear to significantly increase binding to the seed-only complex. What is the interpretation of the difference in effects between F294A, Y311A and R315V, H316A?

- ZSWIM8 expression levels (as shown in Supplementary Figure 6) should be quantified.

- Page 9, "Therefore, despite presenting an unoccupied PAZ pocket, AGO-miRNA bound to a fully complementary target is a suboptimal partner for ZSWIM8, which we attribute to its misaligned RNA." This statement is at odds with the illustration (as in Extended Data Figure 5C) of a steric clash of ZSWIM8 with the N domain and a gap in the interaction with the PAZ domain for AGO2 bound to a fully complementary target. Why then is the suboptimal TDMD by the fully complementary trigger attributed only to misaligned RNA?

In general, the role of the RNA trajectory currently seems somewhat overstated and undertested. While I agree that the contacts of RBE1-3 with the trigger flanks appear to enhance the overall interaction, I do not see direct tests of the role of the P2 duplex in promoting those interactions. The relevant experiment would need to disentangle the role of the P2 duplex in PAZ pocket clearing from its role in directing RNA trajectory. Does the retained ZSWIM8-mediated discrimination of the trigger complex vs the seed-only complex in the PAZ pocket mutants (F294A, Y311A and R315V, H316A) reflect the role of "RNA trajectory"? (By this logic, perhaps the complexes containing a 14-mer miRNA are not well discriminated because the trigger and seed-only sites differ in neither PAZ clearance nor RNA trajectory due to the meager 4-bp P2 duplex in this context?) Could this be tested by modulating the size of the central bulge? For instance, if extending or shrinking the central bulge on the trigger side is expected to modify the trajectory of the flanking RNA, then does this further reduce ZSWIM8 discrimination between the trigger complex and the seed-only complex in the context of the PAZ pocket mutants? (And what is the effect in the context of wild type AGO2?) Performing these experiments by in vitro co-IP in the context of both wild type and PAZ pocket mutant AGO2 could help lend weight to the RNA trajectory argument. Alternatively, weakening or removing the language about the unique RNA trajectory in the results and conclusions sections would be appropriate.

- It is a little difficult to understand how the current structure relates to the previous TDMD structure of just AGO-miRNA-trigger (e.g. 6NIT as in Figure 3B). It seems that perhaps ZSWIM8PAZ wedges in between the P2 duplex and the PAZ, slightly increasing the distance between the RNA and the PAZ and stabilizing their positions. Is that the case? Could you provide a video showing the morphed transition from the supplementary pairing to the previous TDMD structure to the current ZSWIM8-bound structure? A zoomed out view as well as a zoomed in view of the PAZ pocket would be helpful.

- Throughout the paper, proper statistical tests should be performed. Examples include (but are not limited to) Figure 3C and Figure 4C.

Minor points

- Please discuss/show conservation of key residues/regions of ZSWIM8 (as is done for AGOs in Supplementary Figure 12).
- In supplementary Figure 12, positions R315 and H316 should be shown since these are key positions tested in Extended Data 7D.

Referee #3

(Remarks to the Author)

I co-reviewed this manuscript with one of the reviewers who provided the listed reports.

Referee #4

(Remarks to the Author)

In this work, the David group and collaborators advance their pioneering discovery by revealing the structural and mechanistic basis of ZSWIM8-mediated TDMD, providing a compelling explanation for the rapid turnover of extensively paired miRNAs. Biochemically, they establish the substrate selectivity of ZSWIM8 toward the trigger-bound AGO complex, in contrast to the canonical mRNA-bound AGO complex, and demonstrate that this activity is enhanced by the trigger-flanking sequence. Through structural analysis, they identify a unique PAZ conformation, an unoccupied PAZ pocket, and a perturbed RNA duplex trajectory, which collectively determine ZSWIM8's selectivity. This paper, written in clear logic with high-quality experimental data, represents a major conceptual advance in our understanding of microRNA. However, I believe the manuscript could be strengthened by addressing the following questions.

1. As observed in other Argonaute structures, the PAZ domain is relatively flexible compared to the other domains. This raises questions about the causality between PAZ stabilization and ZSWIM8 recognition. To address this, I suggest presenting the structure of the AGO2-miR-7-trigger complex to compare the structural differences between the ZSWIM8-bound AGO2-miR-7-trigger complex and the ZSWIM8-free AGO2-miR-7-trigger complex. Additionally, it would be helpful to roughly estimate the binding affinity between ZSWIM8 and the AGO complex based on the current binding assays, and to indicate the percentage of free AGO versus bound AGO in the cryo-EM dataset. Also, does the PAZ domain form reliable interactions with the trigger-miR-7 RNA duplex?

2. The authors propose a fascinating model involving a perturbed RNA trajectory, which distinguishes the miRNA-trigger RNA duplex from a perfectly matched RNA duplex. To further support this model, additional controls should be added in Figure 1e to demonstrate that ZSWIM8 selectivity does not extend to the perfectly matched duplex. Moreover, for other miRNA-trigger pairs, can this middle-positioned imperfection and the perturbed RNA trajectory be computationally observed?

3. The poly-ubiquitination process requires more detailed description. What are the specific ubiquitination sites on Ago2? Are these ubiquitination sites related to the spatial organization of the ZSWIM8-ELOB-ELOC-AGO complex?

4. Is this multivalent interaction mode conserved across other Ago proteins (such as Ago1, Ago3, or Ago4)? Are the interacting residues conserved in other E3-AGO pairs?

Version 1:

Reviewer comments:

Referee #1

(Remarks to the Author)

The authors have addressed all issues satisfactorily and can be congratulated on a great paper

Referee #2

(Remarks to the Author)

My comments were sufficiently addressed. The new internal loop length experiments are very interesting. Clarity is improved on small points throughout. Proper statistical tests have been incorporated.

Referee #3

(Remarks to the Author)

I co-reviewed this manuscript with one of the reviewers who provided the listed reports.

Referee #4

(Remarks to the Author)

In the revised manuscript, the authors have appropriately addressed the questions. I have minor suggestions to further strengthen the manuscript. I apologize for not explicitly stating my questions regarding the middle-positioned imperfection and the perturbed RNA trajectory in other miRNA-trigger pairs. This reviewer agrees that it can be difficult to faithfully predict RNP structures using current prediction tools. With that said, I am wondering whether, at the sequence level, other miRNA-trigger pairs share the same kind of middle imperfection (e.g., a bulge) as observed in the CYRANO-miR-7 pair. It would be helpful to provide diagrams of other miRNA-trigger pairs, similar to those shown in Fig. 1b. In addition, it would be helpful to present the mass spec data together with the lysine positions in Supplementary Fig. 2. Finally, I suggest that the authors eliminate the Ramachandran and side-chain outliers in PDB entry 9RWZ.

We are pleased by the overall enthusiasm for our study. We thank the referees for taking the time to review our paper and for their constructive feedback, which has helped us to improve our manuscript. We repeat all referee comments in black type, and insert our responses in blue.

In addition to the improvements made as we addressed the points of the reviewers, we have also improved our ZSWIM8 co-IP results. We have replaced many of our ZSWIM8 *in vitro* co-IP assays with versions in which we have added heparin, which reduces non-specific binding of longer RNAs to the beads. The addition of heparin provided a cleaner signal over background in our assays, and it did not change any conclusions that we had made from these assays in our original submission. These updated assays are shown in Figures 1f, 3g, and 4e, and Extended Data Figures 7d, 10c, and 10f.

Referee #1 (Remarks to the Author):

This is a beautiful study on the mechanism of target directed microRNA degradation (TDMD). In this process Argonaut binds miRNA and a trigger RNA and the combination of those three is recognized by an adaptor protein ZSWIM8. This adaptor protein would be expected to collaborate with CUL2/CUL5 CRLs, but in this instance it works with CUL3. A cryo-EM study is presented that explains the 4-way specificity of the complex as well as the need for CUL3, all backed up by mutant analysis and biochemical and cellular assays. All in all, this is very high-class work, really illuminating an unusual regulatory pathway, helping the field forward.

We are pleased by the reviewer's enthusiasm for our study.

There are only a few minor comments

- The introduction talks a lot about mouse and Drosophila but fails to mention human. As the main text is also not clear on this, one needs to find the materials and methods to realize the work is on human proteins. It would be helpful to mention this in a few places.

We have revised the introduction to highlight the implied scope of TDMD in humans, pointing out that over 50 ZSWIM8-sensitive miRNAs have been identified in human cell lines. In addition, we revised the structure section to emphasize that our study investigates human proteins.

- The cryo-EM structure is introduced too briefly to be understood, it would be good to use a few sentences to explain how 'The structure' was obtained (combination of focused maps, with mobility analysis, to be discussed later) and which aspects will be discussed later e.g. the 'flanking' RNA regions are mentioned but only explained pages later

When introducing the structure, we have added more information explaining how the structure was obtained and analyzed.

- An interesting question, triggered in part by the discussion, is whether the rearranged interaction changes k_{cat} or k_M of the E3 ligase. It seems the authors favor binding as the main effect, but it would be useful to point out in discussion that effects on catalysis are also possible.

Effects on catalysis could indeed be playing a role in TDMD selectivity. We now state that beyond preferential binding by ZSWIM8, the trigger RNA could influence catalytic efficiency in other ways, such as by orienting the AGO protein to place its lysine residues in a position more suitable for ubiquitylation, and/or by increasing processivity of polyubiquitylation.

- Fig 1A: it is not at all clear how long Cul3 is, what part is in the experiment and what part is visible; please replace the schematics in Figure 1 to reflect the full-length proteins, and indicate what was included in the experiment and what was visible in some sort of color/pattern scheme. Also indicate precise boundaries for the constructs here or in materials and methods.

We have updated Figure 2a to indicate the boundaries of the CUL3 N-terminal domain (NTD) within a schematic of full-length CUL3. A more detailed description has also been added to the legend of Figure 2, stating the length of CUL3 used to obtain the structure and specifying the residues that were modelled in our structure. In addition, the precise truncation and associated solubilizing mutations of CUL3 NTD have been added to the methods.

- Extended Fig 3I: what is the extra band at 70 kD?

The band corresponds to the thioester-linked ARIH1~UB* intermediate, from which the fluorescent UB* is transferred to the substrate. We have added a label to the gel and an explanation in the legend for Extended Data Figure 3I.

- Extended Fig 4D: the red region is called CUL3 box in the legend, why not add this to the figure; it would also be helpful to color this region red in Fig 4E

We have updated Extended Data Figure 4d and e, labeling the CUL3-box and coloring this region in red.

- Extended Fig 7D: Whereas one of the double mutants is affected, there does not seem to be a detrimental effect of the R315V,H316A double mutant, whereas the main text claims they are both detrimental. This needs to be clarified better in the text.

We thank the reviewer (and Reviewer #2) for pointing this out. F294 and Y311 are central to the pocket binding the 3' end of the miRNA. In contrast, R315 and H316 are not central to the PAZ pocket, nor do they directly contact the 3' base. Their potential role in interacting with the phosphodiester backbone of the miRNA also remains unclear. Therefore, we removed the data and text regarding the R315V, H316A mutant.

Referee #2 (Remarks to the Author):

In the submitted manuscript, Farnung and colleagues describe the structural basis for target-directed microRNA degradation (TDMD). They first reconstitute the process of TDMD in vitro, demonstrating that a model TDMD-triggering RNA (Cyrano) can specifically direct ubiquitylation of Argonaute proteins by an E3 ligase of which ZSWIM8 is the substrate recognition module. The TDMD trigger requires extensive complementarity to the miRNA to elicit ubiquitylation since a seed-only binding site does not do so. The authors confirm the unusual composition of the ZSWIM8 E3 complex that was previously implied by genetic experiments (coupling Elongin B/C with CUL3 and ARIH1). The authors establish an assay that tests recruitment of ZSWIM8 by various AGO-miRNA-target/trigger ternary complexes, demonstrating that the complex containing the Cyrano trigger binding site more efficiently recruits ZSWIM8 than a seed-only complement. Building upon this, the authors resolve the structure of a full "TDMD complex" in which the AGO2-miRNA-trigger complex is bound by a clamp of two ZSWIM8 protomers, each itself bound to one unit of ELOB, ELOC, and CUL3. The structure provides many surprising insights into the mode of interaction of the AGO2-miRNA-trigger complex with the ubiquitin ligase. First, the dimerization of ZSWIM8 is unusual, forming an irreversible intermolecular knot between the monomers; nearby domains also form the binding platforms for CUL3 and ELOB/C. Second, the ZSWIM8 protomers each interact with opposite surfaces of the AGO, one along the MID domain and the other along the PAZ/N domains. Third, specific contacts sense an "empty" PAZ domain that results from release the miRNA 3' terminus. Fourth, ZSWIM8 forms nonspecific interactions with the regions of the trigger RNA extending from AGO.

The structure is very impressive and answers many questions regarding the mechanism of TDMD. The biochemical assay measuring recruitment of ZSWIM8 by various RNA substrates is very informative, and the use T6B as a control for differences in target affinities is elegant. The experiments testing the requirement for dimerization of ZSWIM8 are also particularly cleverly designed and easy to interpret.

We are pleased by the reviewer's enthusiasm for our study.

Some of the experiments testing importance of specific interfaces would be more impactful with additional assays performed as suggested below. Some of the writing tends to overstate the certainty of the role of RNA trajectory.

- The assay for testing the function of AGO2 residues in Extended Data Figure 7b seems more difficult to interpret than other assays in the paper. Introducing the variants into cells expressing endogenous AGOs and normalizing to other IPed miRNAs may have caveats. For instance, if a constitutively empty PAZ pocket reduces ZSWIM8's ability to discriminate between a trigger and a seed-only complex, as suggested by Figure 3F, then non-TDMD substrates like miR-19b and let-7 could become TDMD substrates when loaded in PAZ mutant AGO, confounding interpretation of the assay as it's currently normalized. (Assuming that non-PAZ AGO mutants would not affect ZSWIM8's discrimination between trigger and seed-only could also be worrisome and slightly circular.) The in vitro co-IP assay (Extended Data 7d) does not carry the same caveats, so it would be helpful if additional AGO mutants could be tested by this assay: particularly the mutants expected to disrupt interaction with ZSWIM8 F1261 (AGO2 F491A and K493A).

As suggested, we have tested the F491A and the K493A AGO2 variants in the *in vitro* ZSWIM8 co-IP assay. Consistent with the conclusions of the intracellular AGO2 co-IP assay, we observed a reduction in ZSWIM8 co-IP *in vitro* for both AGO2 variants when bound to the trigger RNA. We have added these results to Extended Data Figure 7d and e.

Regarding the larger issue of interpreting results of the intracellular AGO2 co-IP assay (Extended Data Figure 7b,c), we appreciate the comments regarding potentially confounding effects on miRNA levels upon mutating the AGO2 PAZ domain. Although we cannot definitively exclude a minor contribution of non-specific miRNA turnover through the action of ZSWIM8 upon mutation of AGO2, we would like to point out that reduced levels of miR-19b and let-7 associated with AGO2 PAZ-domain variants were also observed in $\Delta Zswim8$ cells. In the $\Delta Zswim8$ background, miRNAs do not undergo TDMD, and thus reductions in miR-7, miR-19b, and let-7 levels in this background did not appear to be due to TDMD. Instead, we attribute reductions in levels of these miRNAs to defects in the formation or stability of the AGO2-miRNA complex, which must have occurred in a ZSWIM8-independent manner. Therefore, with this consideration of the relative miR-7, miR-19b, and let-7 levels in $\Delta Zswim8$ cells, our intracellular AGO2 co-IP assay is informative. Compared to the *in vitro* ZSWIM8 co-IP assay, the intracellular AGO2 co-IP assay is also more convenient for examining the effects of AGO substitutions because it does not require prior purification of the mutant AGO-miRNA complex. We have clarified these issues in the revised text and figure legend.

- On page 8, "With the goal of impairing miRNA 3' end binding to the PAZ pocket, PAZ residues that interact with the terminal miRNA nucleotide at either the nucleobase (F294A, Y311A) or its backbone phosphate (R315V, H316A) were mutated (Figure 3e). These substitutions caused increased binding to the seed-only-bound AGO2-miR-7 (Extended Data Figure 7d,e), consistent with an unoccupied PAZ pocket comprising a feature selectively recognized by ZSWIM8." R315V, H316A does not appear to significantly increase binding to the seed-only complex. What is the interpretation of the difference in effects between F294A, Y311A and R315V, H316A?

We thank the reviewer (and Reviewer #1) for pointing this out. F294 and Y311 are central to the pocket binding the 3' end of the miRNA. In contrast, R315 and H316 are not central to the PAZ pocket, nor do they directly contact the 3' base. Their potential role in interacting with the phosphodiester backbone of the miRNA also remains unclear. Therefore, we removed the data and text regarding the R315V, H316A mutant.

- ZSWIM8 expression levels (as shown in Supplementary Figure 6) should be quantified.

As requested, we have updated this figure (now Supplementary Figure 7) to show the quantification of ZSWIM8 expression levels. Likewise, we have updated now Supplementary Figure 10 to show the quantification of AGO2 expression levels.

- Page 9, "Therefore, despite presenting an unoccupied PAZ pocket, AGO-miRNA bound to a fully complementary target is a suboptimal partner for ZSWIM8, which we attribute to its misaligned RNA." This statement is at odds with the illustration (as in Extended Data Figure 5C) of a steric clash of ZSWIM8 with the N domain and a gap in the interaction with the PAZ domain for AGO2 bound to a fully complementary target. Why then is the suboptimal TDMD by the fully complementary trigger attributed only to misaligned RNA?

As the reviewer correctly points out, we cannot unambiguously state that the suboptimal binding observed with the fully complementary CYRANO RNA is solely attributable to the RNA trajectory. The contributions of PAZ- and N-domain orientation to this selectivity are more difficult to interpret, as single-molecule FRET analyses (Willkomm, *Nat. Comm.* 2022) have reported increased flexibility in the PAZ and N domains upon binding of fully complementary RNA. This increased flexibility may at least in part resolve the steric clashes/gaps observed in the AGO structures bound to fully complementary RNA. The sentence quoted by the referee has been deleted in our revision, and more generally, we have further investigated the potential role of RNA trajectory, as described in our response to the next point.

In general, the role of the RNA trajectory currently seems somewhat overstated and undertested. While I agree that the contacts of RBE1-3 with the trigger flanks appear to enhance the overall interaction, I do not see direct tests of the role of the P2 duplex in promoting those interactions. The relevant experiment would need to disentangle the role of the P2 duplex in PAZ pocket clearing from its role in directing RNA trajectory. Does the retained ZSWIM8-mediated discrimination of the trigger complex vs the seed-only complex in the PAZ pocket mutants (F294A, Y311A and R315V, H316A) reflect the role of "RNA trajectory"? (By this logic, perhaps the complexes containing a 14-mer miRNA are not well discriminated because the trigger and seed-only sites differ in neither PAZ clearance nor RNA trajectory due to the meager 4-bp P2 duplex in this context?) Could this be tested by modulating the size of the

central bulge? For instance, if extending or shrinking the central bulge on the trigger side is expected to modify the trajectory of the flanking RNA, then does this further reduce ZSWIM8 discrimination between the trigger complex and the seed-only complex in the context of the PAZ pocket mutants? (And what is the effect in the context of wild type AGO2?) Performing these experiments by *in vitro* co-IP in the context of both wild type and PAZ pocket mutant AGO2 could help lend weight to the RNA trajectory argument. Alternatively, weakening or removing the language about the unique RNA trajectory in the results and conclusions sections would be appropriate.

We appreciate the suggestions to further test the importance of the RNA trajectory in dictating TDMD selectivity. As suggested, we performed *in vitro* co-IP and ubiquitylation assays using trigger RNA variants in which the length of the internal loop was varied. These trigger variants were expected to retain pairing to the 3' region of the miRNA, thus generating an unoccupied PAZ pocket, but were expected to have an altered P2-duplex trajectory. Changing the size of the internal loop led to weaker ZSWIM8 binding and reduced polyubiquitylation of AGO2. These findings support the idea that besides removing the miRNA 3' terminus from the PAZ pocket, the P2 duplex also functions to position the 5' flanking region of the trigger RNA in proximity to the RBEs of the ZSWIM8^{NPAZ} protomer, although it is also possible that the size of the internal loop affected other aspects of the complex, such as the orientation of AGO2 domains and/or the propensity for miRNA–trigger 3' pairing to occur. Upon mutation of the PAZ RNA-binding pocket (F294A, Y311A), trigger variants with longer internal loops induced binding and polyubiquitylation no better than did the seed-only variant, supporting the idea that together, trigger-specific RNA trajectory and binding of the vacated PAZ pocket substantially contribute to ZSWIM8 selectivity for TDMD substrates. We have included these new co-IP and ubiquitylation assays in Extended Data Figure 10a–d, with corresponding updates to the text.

- It is a little difficult to understand how the current structure relates to the previous TDMD structure of just AGO–miRNA–trigger (e.g. 6NIT as in Figure 3B). It seems that perhaps ZSWIM8PAZ wedges in between the P2 duplex and the PAZ, slightly increasing the distance between the RNA and the PAZ and stabilizing their positions. Is that the case? Could you provide a video showing the morphed transition from the supplementary pairing to the previous TDMD structure to the current ZSWIM8-bound structure? A zoomed out view as well as a zoomed in view of the PAZ pocket would be helpful.

We have prepared an additional video (Supplementary Video 4) showing the structural transitions observed between a previously determined structure of an AGO2–miRNA complex bound to a target RNA with 3'–supplementary pairing (PDB: 6N4O), a previously determined structure of an AGO2–miRNA–trigger complex (PDB: 6MDZ), and our AGO2–miR-7–CYRANO structure. These morphed transitions visualize the rotation of the PAZ domain when transitioning from an AGO2–miRNA–trigger complex to a complex bound by ZSWIM8. This rotation decreases the gap between the PAZ domain and miRNA–trigger duplex, and it places the PAZ domain in position to interact with the ZSWIM8 TDMD sensor domain. AGO2 residues 331–337 also rearrange to widen the central pocket of the PAZ domain, which accommodates binding of the ZSWIM8 TDMD sensor element.

- Throughout the paper, proper statistical tests should be performed. Examples include (but are not limited to) Figure 3C and Figure 4C.

As requested, statistical tests have been added to Figures 3C and 4C, and elsewhere, where appropriate (Extended Data Figures 3f, 3n, 4b, 4j, 7c, 7e, 10c in the revised manuscript).

Minor points

- Please discuss/show conservation of key residues/regions of ZSWIM8 (as is done for AGOs in Supplementary Figure 12).

As requested, we have added a multiple-sequence alignment of ZSWIM8 and other ZSWIM family members (Supplementary Figure 13 of the revised manuscript). The alignment shows conservation of ZSWIM8 residues that interact with AGO2, as determined in our cryo-EM structure. Interestingly, many ZSWIM8 residues that form essential interactions with AGO2 (311–343, 1261, 1303–1304, 1341–1350, 1372–1379) are found in insertions unique to ZSWIM8 and are not present in other members of the ZSWIM family. Other regions of the ZSWIM family are highly conserved, with ZSWIM8 being the most divergent member of the family.

- In supplementary Figure 12, positions R315 and H316 should be shown since these are key positions tested in Extended Data 7D.

Due to the ambiguous expectations for the R315V, H316A mutant, we have opted to not consider this mutant in the sequence alignment shown in Supplementary Figure 12.

Referee #3 (Remarks to the Author):

I co-reviewed this manuscript with one of the reviewers who provided the listed reports.

Referee #4 (Remarks to the Author):

In this work, the David group and collaborators advance their pioneering discovery by revealing the structural and mechanistic basis of ZSWIM8-mediated TDMD, providing a compelling explanation for the rapid turnover of extensively paired miRNAs. Biochemically, they establish the substrate selectivity of ZSWIM8 toward the trigger-bound AGO complex, in contrast to the canonical mRNA-bound AGO complex, and demonstrate that this activity is enhanced by the trigger-flanking sequence. Through structural analysis, they identify a unique PAZ conformation, an unoccupied PAZ pocket, and a perturbed RNA duplex trajectory, which collectively determine ZSWIM8's selectivity. This paper, written in clear logic with high-quality experimental data, represents a major conceptual advance in our understanding of microRNA. However, I believe the manuscript could be strengthened by addressing the following questions.

We are pleased by the reviewer's enthusiasm for our study.

1. As observed in other Argonaute structures, the PAZ domain is relatively flexible compared to the other domains. This raises questions about the causality between PAZ stabilization and ZSWIM8 recognition. To address this, I suggest presenting the structure of the AGO2-miR-7-trigger complex to compare the structural differences between the ZSWIM8-bound AGO2-miR-7-trigger complex and the ZSWIM8-free AGO2-miR-7-trigger complex.

We have prepared an additional video (Supplementary Video 4) showing the structural transitions observed between a previously determined structure of an AGO2-miRNA complex bound to a target RNA with 3'-supplementary pairing (PDB: 6N4O), a previously determined structure of an AGO2-miRNA-trigger complex (PDB: 6MDZ), and our AGO2-miR-7-CYRANO structure. These morphed transitions visualize the rotation of the PAZ domain when transitioning from an AGO2-miRNA-trigger complex to a complex bound by ZSWIM8. This rotation decreases the gap between the PAZ domain and miRNA-trigger duplex, and it places the PAZ domain in position to interact with the ZSWIM8 TDMD sensor domain. AGO2 residues 331-337 also rearrange to widen the central pocket of the PAZ domain, which accommodates binding of the ZSWIM8 TDMD sensor element.

Additionally, it would be helpful to roughly estimate the binding affinity between ZSWIM8 and the AGO complex based on the current binding assays, and to indicate the percentage of free AGO versus bound AGO in the cryo-EM dataset.

Results of our *in vitro* ZSWIM8 co-IP assay suggest a conservative estimate of ~50 nM for the binding affinity of interaction of AGO2-miR-7-CYRANO with ZSWIM8 (Figure 1f, 120-nt trigger). However, we caution that this co-IP assay is not an equilibrium measurement and only provides an upper-limit estimate for the K_d value. We did not observe any free AGO2 in our cryo-EM dataset and can therefore not use this dataset to speak to the proportion of bound and unbound AGO2.

Also, does the PAZ domain form reliable interactions with the trigger-miR-7 RNA duplex?

In the TDMD conformation, the miRNA-trigger duplex is not accessible to the miRNA-binding pocket of the PAZ domain. Therefore, our work cannot speak to any interactions between the PAZ-domain pocket and the miRNA-trigger duplex. Residues in the PAZ and N domains appear to contact nucleotides in the distal region of the miR-7-trigger duplex.

2. The authors propose a fascinating model involving a perturbed RNA trajectory, which distinguishes the miRNA-trigger RNA duplex from a perfectly matched RNA duplex. To further support this model, additional controls should be added in Figure 1e to demonstrate that ZSWIM8 selectivity does not extend to the perfectly matched duplex.

Assays with the perfectly matched duplex are shown in Extended Data Figure 10e-g of the revised manuscript. We find that the fully complementary target RNA is only weakly effective in binding and somewhat more effective in polyubiquitylation, but is still less effective than the CYRANO fragment. (The reduced effect on polyubiquitylation compared to binding is attributable to the higher protein concentrations used in our ubiquitylation assay). The lower efficacy of the fully complementary target indicates that a miRNA-free PAZ domain is not sufficient to induce AGO2

binding and polyubiquitylation. We suggest that the fully complementary RNA adopts a conformation that is suboptimal for ZSWIM8 binding, perhaps because the RNA trajectory differs markedly from that of the TDMD complex. This possibility is supported by our new assays examining the effects of changing the size of the internal loop (Extended Data Figure 10a–d of the revised manuscript). On the whole, our results suggest that the entire ensemble of structural changes of both RNA and protein is required for full recognition by ZSWIM8.

Moreover, for other miRNA-trigger pairs, can this middle-positioned imperfection and the perturbed RNA trajectory be computationally observed?

AlphaFold3 (AF3) can generate predictions that include the internal loop and central bend, presumably due to the abundance of AGO structures with a centrally mismatched miRNA–target duplex in the PDB. However, comparison of our AGO2–miR-7–CYRANO structure to the AF3 prediction shows that AF3 fails to predict the geometry of the internal loop and the associated RNA trajectory (**Reviewer Figure 1**). The predicted miRNA conformation aligns to our structure with an RMSD of 13 Å², and the trigger RNA with an RMSD of 15 Å². The 3'-terminus of the miRNA is placed 30 Å away from its location in our structure. Even the internal loop shows distances of 28 Å between predicted and experimental structures. Because AF3 failed to predict the known AGO2–miR-7–CYRANO structure, we did not attempt to apply it for the prediction of ribonucleoprotein structures with other miRNA–trigger pairs. This being said, the TDMD conformations reported in Sheu-Gruttadauria, *Mol. Cell.* 2019, show that a similar RNA trajectory is achieved by a few other miRNA–trigger pairs. The poor performance in predicting the AGO2–miR-7–CYRANO structure was also observed for other state-of-the-art computational methods such as RoseTTAFoldNA.

Reviewer Figure 1. Cartoon representations comparing the AlphaFold3 prediction of the AGO2–miR-7–CYRANO complex to the cryo-EM structure from this study. For the AlphaFold3 prediction, the trigger RNA nucleotides visible in our cryo-EM map were included as input. Distances were calculated between phosphorus atoms in the RNA backbone.

3. The poly-ubiquitination process requires more detailed description. What are the specific ubiquitination sites on AGO2? Are these ubiquitination sites related to the spatial organization of the ZSWIM8-ELOB-ELOC-AGO complex?

As requested, we mapped the ubiquitylation sites on AGO2. Seven lysine residues were identified (39, 241, 248, 425, 655, 696, 844). These lysines are exposed on the surface, distributed across both sides of AGO2, and at positions that would not be structurally occluded by ZSWIM8. This arrangement of modified lysines aligns with the dimeric nature of ZSWIM8, which, together with a flexible, neddylated CUL3–RBX1, can position ARIH1 on both sides of AGO2 for ubiquitin transfer. These findings also correlate with previous molecular analyses, which indicate that

multiple lysine residues must be substituted to abrogate TDMD (Shi, *Science* 2020). These new results are shown in Supplementary Figure 2 of our revised manuscript.

4. Is this multivalent interaction mode conserved across other Ago proteins (such as Ago1, Ago3, or Ago4)? Are the interacting residues conserved in other E3-AGO pairs?

We now include multiple-sequence alignments of AGO2 and ZSWIM8 homologs in Supplementary Figures 12 and 13, respectively. Residues shown to interact are conserved across AGO paralogs in humans, as well as across AGO and ZSWIM8 orthologs in other metazoan species. Moreover, ZSWIM8 mutants shown to be defective in human AGO2 polyubiquitylation were also defective in human AGO1 polyubiquitylation (Extended Data Figure 6g of the revised manuscript), further supporting the conservation of ZSWIM8–AGO interactions.

1. Please reduce the overall length of the article as detailed below (see 'LENGTH'). This will probably only require mild shortening of the manuscript text considering that you only have 4 figures. I suggest you aim for 5,000 words in the main text (currently 5,513)

We have reduced the length of the main text to 4,999 words.

2. We require that the cryo-EM reporting table is included in the Extended Data. The table is currently included as Supplementary Table 2. Please note that we only allow up to 10 Extended Data items (tables and figures), so please consider removing one ED figure to stay within that limit (e.g. by merging two, or moving one to SI). Like figures, Extended Data tables also have to be submitted in .jpg, .tif or .eps format. See 'EXTENDED DATA' and 'DATA DEPOSITION' below for detailed guidance.

We have included the cryo-EM reporting table as Extended Data Table 1. To accommodate the additional Extended Data item we have rearranged the previous Extended Data Figures 5 and 6 into the current Extended Data Figure 5 and Supplementary Figure 10.

3. Please reorganise your article file. For reference, the Word file should be in the order: title & front matter, summary, main text, references, methods, data (&code) availability statement, additional references (with continuous numbering), acknowledgements, author contributions, additional information, figure legends, extended data figure legends.

The manuscript has been updated accordingly.

4. Please reduce the length of the title to 75 characters (with spaces) or less, so that it fits on two lines in the final layout.

We have updated the title to reduce its length to 74 characters:

The E3 ubiquitin ligase mechanism specifying targeted microRNA degradation

5. Please provide the manuscript in .docx format. Currently it is in PDF format.

We have submitted the updated manuscript file as a word document.

6. Please reduce the Abstract to 230 words or less. Currently there are 300 words.

We have reduced the length of the abstract to 229 words.

7. The number of references should generally not exceed 60. Currently you have 85 references

We have reduced the number of main text references to 60.

8. Please create a separate reference list for any methods references, making sure that the numbering continues from the main text references.

The reference formatting has been updated accordingly.

9. Please note that methods section is provided as a separate file. This needs to be included with the main text (see point 3).

The methods have been added to the main manuscript file.

10. Please remove the main figures from the article file and re-supply them individually in an acceptable format such as EPS, AI, PS, PDF, PPT, PSD or XLS (for graphs) with editable vector files.

The figure files have been provided separately as .ai files.

11. Please ensure that the text size in all figures is at least 5 pt Arial.

All text in the figures is at least 5 pt.

12. Please remove the Extended data figures from the article file and re-supply them individually in EPS, JPEG or TIF format.

The figure files have been provided separately as .eps files.

13. Please re-supply the figures in an acceptable format such as EPS, AI, PS, PDF, PPT, CDR, PSD or XLS (for graphs)

All figure files have been provided separately as either .ai or .eps files.

14. Please note that the legends for the main figures should not exceed 300 words. If it is not possible to reduce the length accordingly, please ensure that the final legends are as close as possible to 300 words.

We have reduced the length of the legends to be as close to 300 words as possible. The legends for the four figures are 319, 160, 320, and 295 words, respectively.

16. Please reduce subheadings to 40 characters (with spaces) or less.

Subheadings have been updated to be 40 characters or less.

17. Please provide a supplementary information guide as a separate word document.

A supplementary information guide covering Supplementary Figures, Tables and Videos has been submitted.

18. Please provide titles/legends for the supplementary videos.

Titles and legends for Supplementary Videos have been submitted as part of the supplementary information guide.

19. There are potential third party rights issues in the figures. Please check the sources of all illustrations and clarify whether permissions are needed to adapt or reproduce them. Please make sure to include the relevant details in third party rights table when you resubmit (more information below). If Biorender or a similar software has been used, please also ensure to provide relevant licenses. In particular please check: Figures 1a, 3(f,e), 4(a,f,g), Extended data Figures 1a, 3(d,i,l-n), 4d, 7a.

All cartoons were drawn by the authors, without the aid of BioRender or a similar software, and so there are no third party issues.

20. Please make sure to provide a third party rights table (more information below) when you resubmit. If Biorender or similar software has been used, please also ensure to provide relevant licenses.

See answer to point 19.

21. Flagging that the figure panels contain a table.

Figure panel 4g is not a table but a cartoon illustrating the structural comparisons of multiple AGO structures. The panel contains no data but instead summarizes the authors' interpretations of structural analyses.

22. A combined "Data and code availability" statement has been provided in the manuscript. Please provide separate 'Data availability' and 'Code availability' statements.

We have updated the manuscript to include separate "Data availability" and "Code availability" statements.

23. The data deposited to the PDB with dataset identifier 9RWZ and EMDB with the dataset identifiers EMD-54348, EMD-54351, EMD-54350, EMD-54349, EMD-54350, EMD-54351 and

EMD-54352 are currently not available. Please ensure their timely public release. Likewise, other datasets and code (sequencing data in GEO, GitHub) should be made available.

We will ensure the timely release of the PDB, EMDB, and GEO submissions, as well as our code, once we have received the proofs and publication date of our manuscript.

24. The Competing Interests statement needs to encompass all authors (i.e. add a sentence like "All other authors declare no competing interests.") Also, for patents, relevant authors should be listed and the patent number stated, e.g. "X.X and Y.Y are named as inventors on a pending patent application for ZZZ (insert patent number)."

We have updated the Competing Interests statement accordingly.

25. For any Supplementary Figures, please check and confirm that:

* If data is presented as bar charts, individual data points are shown using overlaid dot plots.

* The n number (i.e. the sample size used to derive statistics) is provided and defined as a precise value (not a range), using the wording "n=X samples/cells/independent experiments" etc. where applicable.

* Any chart axis, error bars, scale bars, symbols and colour scales are defined.

* Any statistical tests used for data analysis are specified and exact p-values are provided either on the figures themselves, in the legend or in the Source Data file.

* Wherever representative data such as micrographs are shown, the legend indicates how many times the experiment was repeated with the same results.

The Supplementary Figures have been updated accordingly.

Reviewer Comments:

Referee #4 (Remarks to the Author):

In the revised manuscript, the authors have appropriately addressed the questions. I have minor suggestions to further strengthen the manuscript. I apologize for not explicitly stating my questions regarding the middle-positioned imperfection and the perturbed RNA trajectory in other miRNA–trigger pairs. This reviewer agrees that it can be difficult to faithfully predict

RNP structures using current prediction tools. With that said, I am wondering whether, at the sequence level, other miRNA–trigger pairs share the same kind of middle imperfection (e.g., a bulge) as observed in the CYRANO–miR-7 pair. It would be helpful to provide diagrams of other miRNA–trigger pairs, similar to those shown in Fig. 1b. In addition, it would be helpful to present the mass spec data together with the lysine positions in Supplementary Fig. 2. Finally, I suggest that the authors eliminate the Ramachandran and side-chain outliers in PDB entry 9RWZ.

Diagrams of other miRNA–trigger pairs have been compiled in two recent reviews, both of which highlight the common feature of an internal loop, though with varying loop architectures (Table 1 in Buhagiar & Kleaveland, 2024, and Figure 3a in Hiers et al., 2024). Instead of reiterating these diagrams (which take a lot of space) in our paper, we have opted to refer the readers to these reviews.

We have updated Supplementary Figure 2 (now Supplementary Figure 3 in the revised manuscript) to include exemplary mass spectrometry data of the identified ubiquitylation sites.

We have reduced the number of Ramachandran and rotamer outliers and have submitted this updated model to the PDB. Please see the updated validation report submitted with our revised manuscript.